

# A new giant sauropod, *Australotitan cooperensis* gen. et sp. nov., from the mid-Cretaceous of Australia

Scott A. Hocknull[1,2], Melville Wilkinson[3], Rochelle A. Lawrence[1], Vladislav Konstantinov[4], Stuart Mackenzie[3] and Robyn Mackenzie[3]

[1] Geosciences, Queensland Museum, Hendra, Brisbane City, Australia
[2] Biosciences, University of Melbourne, Melbourne, Victoria, Australia
[3] Eromanga Natural History Museum, Eromanga, Queensland, Australia
[4] Unaffiliated, Oktyabrskaya, Ryazan, Russian Federation

Corresponding author
Scott A. Hocknull,
scott.hocknull@qm.qld.gov.au

## ABSTRACT

A new giant sauropod, *Australotitan cooperensis* gen. et sp. nov., represents the first record of dinosaurs from the southern-central Winton Formation of the Eromanga Basin, Australia. We estimate the type locality to be 270–300 m from the base of the Winton Formation and compare this to the semi-contemporaneous sauropod taxa, *Diamantinasaurus matildae* Hocknull et al., 2009, *Wintonotitan wattsi* Hocknull et al., 2009 and *Savannasaurus elliottorum* Poropat et al., 2016. The new titanosaurian is the largest dinosaur from Australia as represented by osteological remains and based on limb-size comparisons it reached a size similar to that of the giant titanosaurians from South America. Using 3-D surface scan models we compare features of the appendicular skeleton that differentiate *Australotitan cooperensis* gen. et sp. nov. as a new taxon. A key limitation to the study of sauropods is the inability to easily and directly compare specimens. Therefore, 3-D cybertypes have become a more standard way to undertake direct comparative assessments. Uncoloured, low resolution, and uncharacterized 3-D surface models can lead to misinterpretations, in particular identification of pre-, syn- and post-depositional distortion. We propose a method for identifying, documenting and illustrating these distortions directly onto the 3-D geometric surface of the models using a colour reference scheme. This new method is repeatable for researchers when observing and documenting specimens including taphonomic alterations and geometric differences. A detailed comparative and preliminary computational phylogenetic assessment supports a shared ancestry for all four Winton Formation taxa, albeit with limited statistical support. Palaeobiogeographical interpretations from these resultant phylogenetic hypotheses remain equivocal due to contrary Asian and South American relationships with the Australian taxa. Temporal and palaeoenvironmental differences between the northern and southern-central sauropod locations are considered to explain the taxonomic and morphological diversity of sauropods from the Winton Formation. Interpretations for this diversity are explored, including an eco-morphocline and/or chronocline across newly developed terrestrial environments as the basin fills.

## INTRODUCTION

Australian dinosaur palaeontology has experienced somewhat of a resurgence of research over the last decade or so with several new taxa recorded from Cretaceous-aged localities across Australia, including *Wintonotitan wattsi*, *Diamantinasaurus matildae*, *Australovenator wintonensis* (*Hocknull et al., 2009*) and *Savannasaurus elliottorum* (*Poropat et al., 2016*) from Winton, Queensland; *Kunburrasaurus ieversi* (*Leahey et al., 2015*) from Richmond, Queensland; *Weewarrasaurus pobeni* (*Bell et al., 2018*) and *Fostoria dhimbangunmal* (*Bell et al., 2019a*) from Lightning Ridge, New South Wales; *Diluvicursor pickeringi* (*Herne et al., 2018*) and *Galleonosaurus dorisae* (*Herne et al., 2019*) from coastal Victoria; and six new ichnotaxa from Broome, Western Australia (*Salisbury et al., 2016*).

This increased naming of new taxa has mostly occurred due to more intensive study of previously described specimens and already established fossil collections, alongside a moderate increase in new discoveries from known fossil fields. Although a new 'wave' of research focus on Australian dinosaurs is underway, large regions of prospect for Cretaceous-aged fauna remain. Developing this potential both in terms of fauna and their geochronological context is crucial to better understand the palaeobiogeography and biochronology of the Cretaceous-aged terrestrial faunal assemblages.

In the Winton Formation the dinosaurian fossil record is concentrated to a small number of sites near Winton and Isisford, located in the northern portion of the Eromanga Basin (Figs. 1 & 2A). This concentrated research effort is in spite of vast areas of mapped Winton Formation occurring throughout the central, southern and western Eromanga Basin, including much of western Queensland (QLD), large areas of interior and north-eastern South Australia (SA), south-eastern Northern Territory (NT) and north-western New South Wales (NSW) (Figs. 1 & 2A). These poorly developed regions comprise an area of approximately two thirds of the Eromanga Basin, but have currently only yielded isolated vertebrate faunal remains (Table 1). As such, major palaeobiogeographic gaps occur in our knowledge of these mid- to Late Cretaceous faunas, paralleling the vast gaps occurring in other high profile Australian vertebrate fossil records, such as the Quaternary megafauna (*Hocknull et al., 2020*).

New fossil sites from the southwest Queensland portion of the Winton Formation, near the townships of Eromanga and Quilpie have recorded floral, faunal and ichnofossils, including the remains of sauropod dinosaurs (*Hocknull et al., 2019*) (Fig. 2A & 2B). Dinosaurian vertebrate fossils were first discovered in this area in 2004 by property owners of Plevna Downs Station. Subsequent excavations undertaken by Queensland Museum from 2006, and then between the newly established Eromanga Natural History Museum and Queensland Museum, have recovered vertebrate fossil remains that include the fossils described here. The new specimens described are lodged in the Eromanga Natural History

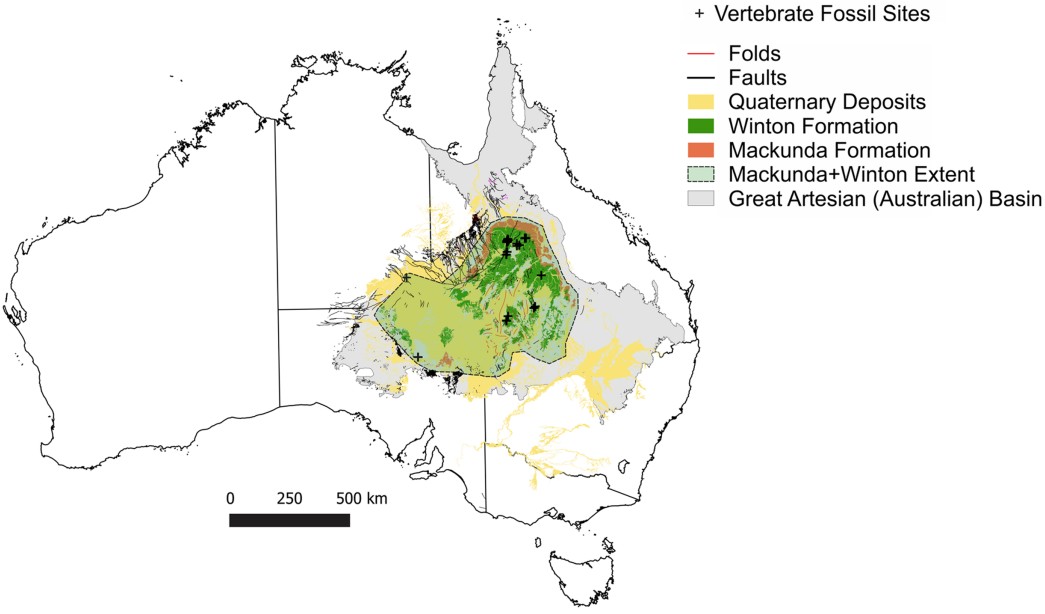

**Figure 1 Vertebrate fossil sites of the Winton Formation (Eromanga Basin).** Geographical map data from http://pid.geoscience.gov.au/dataset/ga/61754 used under CC-BY 4.0 AU. Geological datasets, including the distribution and interpretation of the Quaternary, Winton and Mackunda Formations and their associated and interpreted structures were combined using QGIS 3.14.1 software (http://qgis.org) with data retrieved for; Northern Territory from STRIKE (http://strike.nt.gov.au/wss.html) under CC-BY 4.0; South Australia from SARIG (https://map.sarig.sa.gov.au) under CC-BY 3.0 AU; Queensland from QGlobe (http://qldglobe.information.qld.gov.au) under CC-BY 4.0; New South Wales and overall Eromanga Basin structure retrieved from *Raymond et al. (2012)* (http://ga.gov.au) used under CC-BY 3.0 AU. Great Artesian (Australian) (*Ransley & Smerdon, 2012*).

Museum, a not-for-profit museum with a publicly accessible palaeontological collection that represents vertebrate fossils from the southwest region of Queensland.

We describe a new taxon based on associated sauropod limb and girdle elements along with isolated remains referable to this new taxon. We compare these new finds with other sauropods world-wide sharing similar geological age and body-size, but we pay particular attention to comparisons with the previously described taxa from the northern Winton Formation; *Wintonotitan wattsi* Hocknull et al., 2009, *Diamantinasaurus matildae* Hocknull et al., 2009 and *Savannasaurus elliottorum* Poropat et al., 2016. We do not undertake comparisons to the only other Australian Cretaceous sauropod, *Austrosaurus mckillopi* Longman 1933, because it does not preserve comparable appendicular remains. The new taxon represents the largest dinosaur so far found in Australia represented by osteological remains.

Institutional Abbreviations. AODF (Australian Age of Dinosaurs Museum of Natural History Fossil), AODL (Australian Age of Dinosaurs Museum of Natural History Locality) EMF (Eromanga Natural History Museum Fossil), EML (Eromanga Natural History Museum Locality), QMF (Queensland Museum Fossil), QML (Queensland Museum Locality).

Peer J

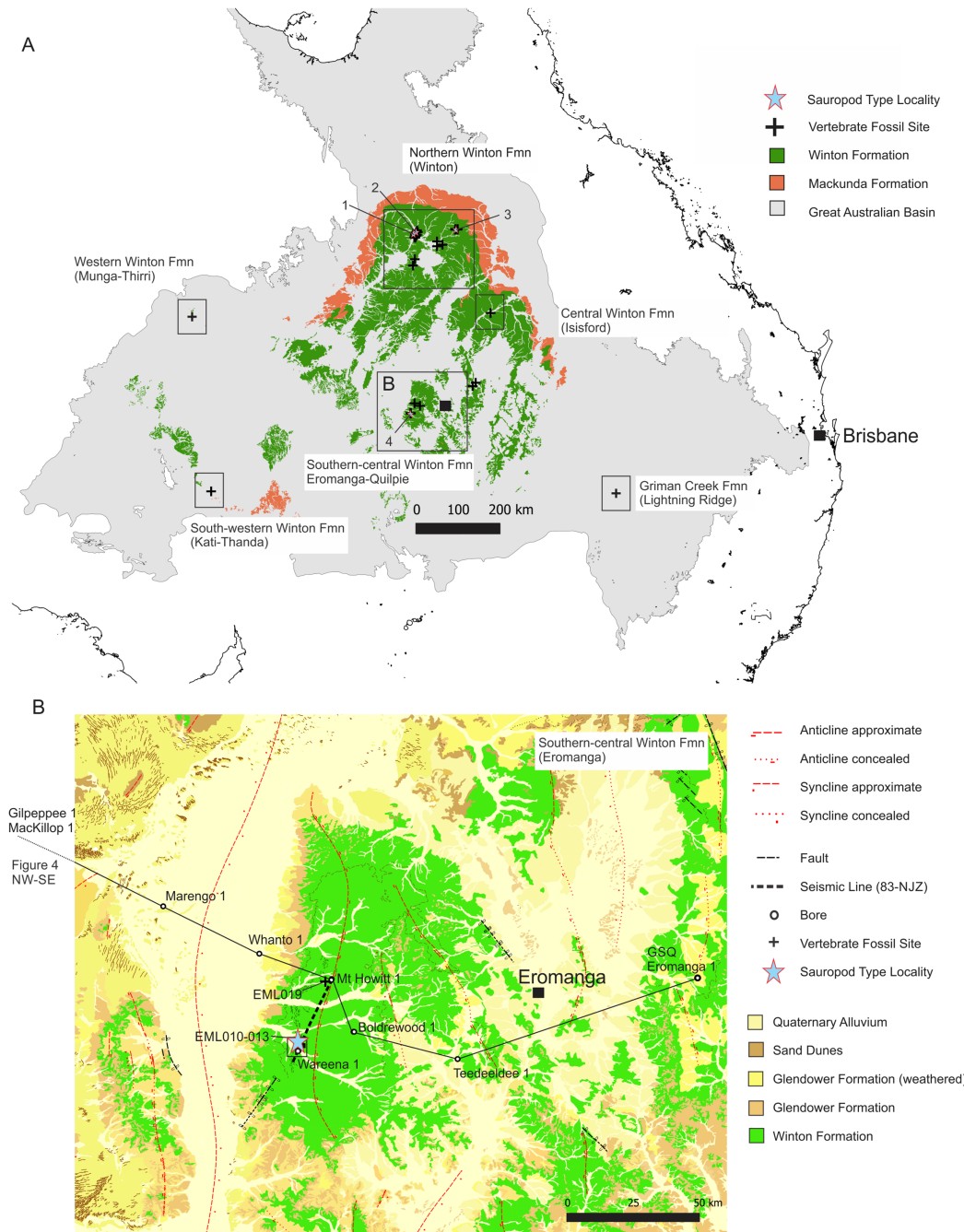

**Figure 2 Distribution of vertebrate fossil sites within the Winton Formation with regionally mapped geology and geological structures relating to the fossil sites described here.** (A) The Winton Formation is here divided into five provinces of known vertebrate fossil sites, including a northern (Winton-Opalton region), central-eastern (Isisford), southern-central (Eromanga-Quilpie region), south-western (Kati Thunda/Lake Eyre) and western (Munga-Thirri/Simpson Desert). South-eastern semi-contemporaneous Griman Creek Formation (Lightning Ridge). Sauropod Type Localities, 1. QML313 (*Wintonotitan wattsi*); 2. AODL085 (*Diamantinasaurus matildae*), 3. AODL082 (*Savannasaurus elliottorum*) and 4. EML010 (*Australotitan cooperensis* gen. et sp. nov.). (B) New vertebrate fossil sites of the southern-central Winton Formation described here including the type locality for *A. cooperensis* gen. et sp. nov. (EML011). Cross-sectional line (NW-SE) shown in Fig. 4A. Seismic line (83-NJZ) cross-sectional

**Figure 2** (continued)

interpretation shown in Fig. 4B. Geographical map data from http://pid.geoscience.gov.au/dataset/ga/61754 used under CC-BY 4.0 AU. Geological datasets, including the distribution and interpretation of the Quaternary, Alluvium, Sand Dunes, Glendower, Winton and Mackunda Formations and their associated and interpreted structures were combined using QGIS 3.14.1 software (http://qgis.org) with data retrieved for; Northern Territory from STRIKE (http://strike.nt.gov.au/wss.html) under CC-BY 4.0; South Australia from SARIG (https://map.sarig.sa.gov.au) under CC-BY 3.0 AU; Queensland from QGlobe (http://qldglobe.information.qld.gov.au) under CC-BY 4.0; New South Wales and overall Eromanga Basin structure retrieved from *Raymond et al. (2012)* (http://ga.gov.au) used under CC-BY 3.0 AU. Great Artesian (Australian) Basin (*Ransley & Smerdon, 2012*). Detailed southern-central geological structures, bores, wells and seismic data retrieved from Qglobe (http://qldglobe.information.qld.gov.au) under CC-BY 4.0.                               

## GEOLOGICAL SETTINGS

The new dinosaur sites reported here are located within the central Eromanga Basin as part of the southern-central Winton Formation. The sites occur 80–90 kilometres (km) west of the township of Eromanga on Plevna Downs Station (Fig. 2B). These new sites are approximately 500–600 km south of the Winton district, which represents the locations for all currently named dinosaurian taxa from the Winton Formation (*Hocknull et al., 2009*; *Poropat et al., 2016*) (Fig. 2A). Approximately 300 km to the north-east of Eromanga, an unnamed ornithopod has been reported from Isisford, representing the first central-eastern Winton Formation dinosaur (*Salisbury et al., 2019*) (Fig. 2A). As yet, no dinosaurian fossils from the south-western or western extremities of the Winton Formation have been found, excepting for a weathered bone from Munga-Thirri (Simpson Desert) that may be dinosaurian (S.A. Hocknull, 2002; 2011, personal observation & A. Yates, 2019, personal communication). A newly dated, now considered semi-contemporaneous dinosaurian fauna, from the Surat Basin Griman Creek Formation, occurs approximately 600 km southeast of Eromanga (*Bell et al., 2019b*) (Fig. 2A).

The new southern-central Winton Formation dinosaur sites are structurally dominated by the Mt. Howitt Anticline, a large anticline with associated Cooper Syncline that produces variable surface exposures of Winton Formation sediments, with a relatively thin cover of Cenozoic alluvium. Each fossil site is located on an alluvial plain with gullies and creeks that drain westward to form part of the greater Cooper Creek channel system. The floodplain forms part of the western portion of the Mount Howitt Anticline (Fig. 2B) and is surrounded by erosion-resistant flat-top hills comprised of Cenozoic silcretes and Glendower Formation that overlie extensively chemically-weathered Winton Formation sediments (*Ingram, 1971*; *Senior, 1970*; *Senior, 1968*) (Fig. 2A).

Outcrop of Winton Formation is sparse and confined to resistant sandstones and calcite cemented siltstone-claystone concretions that form part of the resultant deeply weathered regolith (Fig. 3A). A relatively thin, 1 metre (m) to 2 m thick, soil profile containing a deflation lag of the Cenozoic-aged silcretes and Glendower Formation pebbles, covers most of the available Winton Formation (*Draper, 2002*) (Fig. 3B). Faunal remains and silicified wood are initially found at the surface of this soil profile and are usually associated with broken up cemented concretions or rarely within sandstones.

**Table 1 Cenomanian–?Turonian fauna from the Winton Formation.**

| | Northern Winton Formation (Winton, QLD) | Eastern Winton Formation (Isisford, QLD) | Southern Winton Formation (Eromanga-Quilpie, QLD) | Western Winton Formation (Northern Territory) | South-western Winton Formation (South Australia) |
|---|---|---|---|---|---|
| **Freshwater Gastropods** | *Melanoides* sp. indet.[1] | | | | |
| **Freshwater Bivalves** | *Hyridella (Protohyridella) goodiwindiensis* [2,3] *Hyridella (Hyridella) macmichaeli* [2,3] *Megalovirgus wintonensis* [2,3] new genus et sp.[4] | | *Hyridella (Hyridella) macmichaeli*[4,21] | | *Pledgia eyrensis*[5] |
| **Insects** | ?orbatid mite[6] Odonata[7,8] Mecoptera[7,8] Coleoptera[9] | | | | |
| **Fish** | Teleostii[4] *Metaceratodus wollastoni*[10,11] *Metaceratodus ellioti*[10,11] shark[4] | *Cladocyclus geddesi*[12] ?haleocomorph[13] | | *Metaceratodus wollastoni*[4,10,11] | *Metaceratodus wollastoni*[10,11] *Metaceratodus ellioti*[10,11] shark[14] |
| **Plesiosaur** | Plesiosaur[15] | | | | |
| **Squamates** | cf. *Coniasaurus*[16] | | | | |
| **Turtles** | Chelidae[15,19] | | Chelidae[4,21] | | |
| **Crocodiles** | Crocodilia indet.[15,19] | *Isisfordia duncani*[17] | | | |
| **Pterosaurs** | *Ferodraco lentoni*[18] | | | | |
| **Sauropods** | *Diamantinasaurus matildae*[19,20] *Savannasaurus elliottorum*[20] *Wintonotitan wattsi*[19] sauropod tracks[21,22] | | *Australotitan cooperensis* (here) sauropod trample[21] | | |
| **Theropods** | *Australovenator wintonensis*[19] Theropodan indet.[23] Megaraptoran[24] Theropod tracks[25] | | Theropod tracks[21] | | |
| **Ornithopods** | Ornithopod indet.[26] Ornithopod tracks[25] | new[27] | Ornithopod tracks[21] | | |
| **Ankylosaurs** | Thyreophora indet.[28] | | | | |
| **Cynodont** | ?cynodont[29] | | | | |
| **Dinosauria** | | | | Indeterminate bone[4] | |

**Note:**
Superscript numbers refer to citations: [1](*Cook, 2005*), [2](*Hocknull, 1997*), [3](*Hocknull, 2000*), [4]Hocknull personal observation (2002, 2009, 2019), [5](*Ludbrook, 1985*), [6](*Fletcher & Salisbury, 2014*), [7](*Jell, 2004*), [8](*Elliott & Cook, 2004*), [9](*Salisbury, 2003*), [10](*Kemp, 1991*), [11](*Kemp, 1997*), [12](*Berrell et al., 2014*), [13](*Faggotter, Salisbury & Yabumoto, 2007*), [14](*Mond, 1974*), [15](*Salisbury, 2005*), [16](*Scanlon & Hocknull, 2008*), [17](*Salisbury et al., 2006*), [18](*Pentland et al., 2019*), [19](*Hocknull et al., 2009*), [20](*Poropat et al., 2016*), [21](*Hocknull et al., 2019*), [22](*Poropat et al., 2019*), [23](*Elliott, 2004*), [24](*White et al., 2020*), [25](*Thulborn & Wade, 1984*), [26](*Hocknull & Cook, 2008*), [27](*Salisbury et al., 2019*), [28](*Leahey & Salisbury, 2013*), [29](*Musser et al., 2009*).

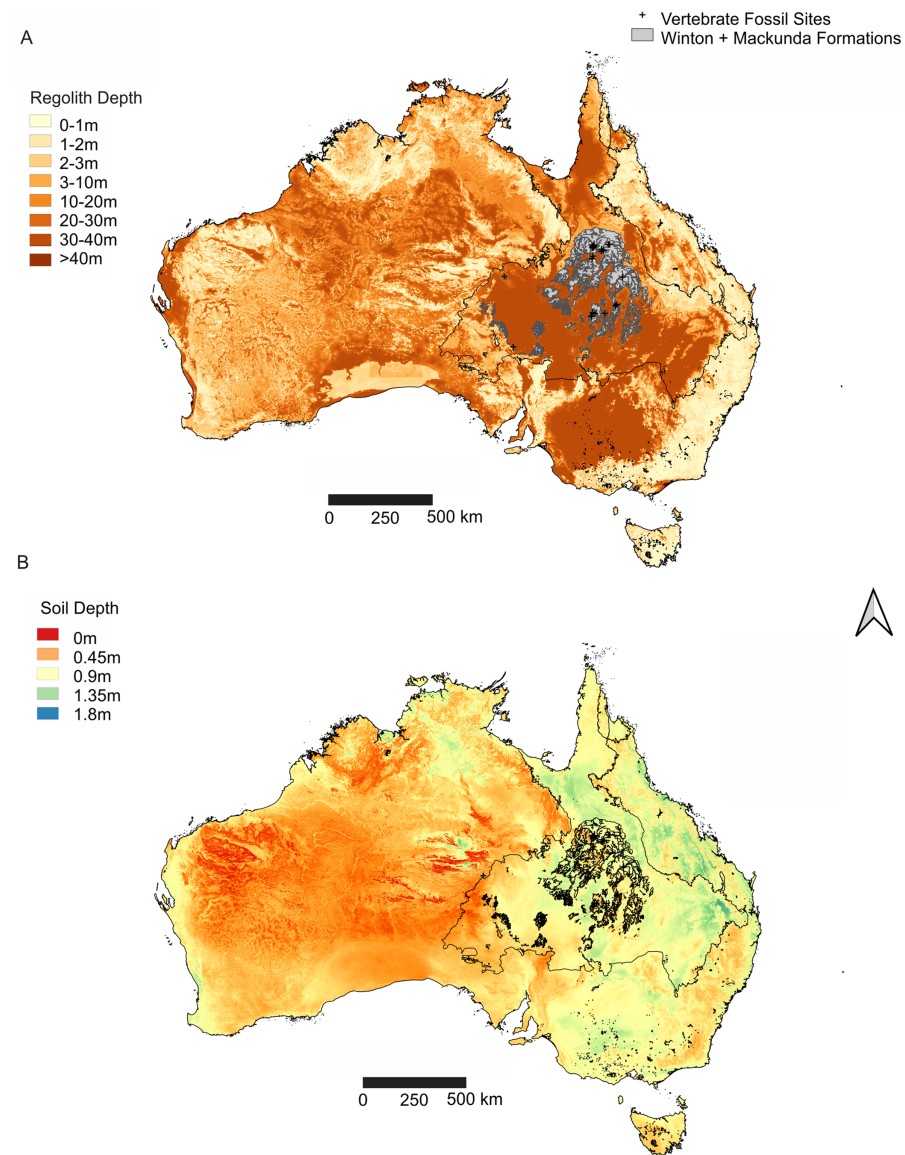

**Figure 3 Distribution of weathering depths of regolith and soil depth, relative to the Winton Formation.** (A) Regolith depth illustrates the significantly deep weathering throughout central and southern Eromanga Basin, which has significantly influenced the Winton Formation in terms of geochemical alteration and post-diagenetic alterations at vertebrate fossil localities. (B) Soil depth illustrates relatively deep soil profiles associated with vertebrate fossils sites from the Winton Formation, reflecting the impact of soil forming processes on available outcrop and vertebrate fossil preservation and exposure. Geographical map data from http://pid.geoscience.gov.au/dataset/ga/61754 used under CC-BY 4.0 AU. Soil Depth dataset retrieved from CSIRO Soil and Landscape Grid National Soil Attribute Maps (https://data.csiro.au/dap/) under CC-BY 4.0. Regolith Depth dataset (*Wilford et al., 2016*) retrieved from CSIRO Soil and Landscape Grid National Soil Attribute Maps (https://data.csiro.au/dap/) under CC-BY 4.0. Outline of Winton and Mackunda formations retrieved for; Northern Territory from STRIKE (http://strike.nt.gov.au/wss.html) under CC-BY 4.0; South Australia from SARIG (https://map.sarig.sa.gov.au) under CC-BY 3.0 AU; Queensland from QGlobe (http://qldglobe.information.qld.gov.au) under CC-BY 4.0; New South Wales and overall Eromanga Basin structure retrieved from *Raymond et al. (2012)* (http://ga.gov.au) used under CC-BY 3.0 AU. Great Artesian (Australian) Basin (*Ransley & Smerdon, 2012*).

The 'self-mulching' actions of the vertosol soils through the expansion and contraction of the smectite-rich clays (*Grant & Blackmore, 1991*) offers a likely mechanism that evidently brings hard material from within the underlying Winton Formation up to the soil surface (e.g., fossilized bones, petrified wood and cemented rock). The vertosol profile itself is derived from the weathering of the underlying Winton Formation, as part of a wider process of cracking clays weathering the Rolling Downs Group surface expression (*Vanderstaay, 2000*). Therefore, over time, as the Winton Formation weathers into a soil profile, the fossil remains rise and concentrate at the surface, breaking into pieces. This same mechanism was originally observed around the township of Winton and led to the discoveries of vertebrate remains at depth and the subsequent new dinosaur discoveries (*Hocknull et al., 2009*). This same process was observed at the Eromanga sites and subsequent excavations proved an essentially identical process yielding similar levels of success for recovering vertebrate fossils and discovering intact bonebeds subsurface.

Inclusions within the soil profile include alluvial sands, clays and gravels derived from major flooding of the Cooper Creek channel system that incorporates the material from the surrounding topographically higher Cenozoic cap rock. Therefore, the soil profile at most sites derives material from two separate sources.

Unlike the northern Winton Formation sites, buried Neogene-Holocene palaeochannels have been observed to cut and erode some of the southern-central Winton Formation dinosaur fossil sites. Therefore, at some time in the past, possibly during wetter periods of the Pliocene or Pleistocene, active channel down cutting likely exposed significant areas of Winton Formation at the surface. Subsequent to this, possibly during the intensifying aridity of the Late Pleistocene, burial of these palaeochannels occurred and vertosols dominated the landscape.

## WINTON FORMATION

The Winton Formation consists of interbedded volcanolithic sandstones, siltstones, mudstones, minor coals and intraformational conglomerates (*Gray, McKillop & McKellar, 2002*). Calcite cemented concretions are common and in places the top approximate 90 m of preserved Winton Formation is highly chemically altered (kaolonitised and ferruginised). The present-day thickness of the Winton Formation ranges from surface exposure on the basin margins that is associated with uplifted structures, to at least 1,100 m of thickness toward the west-southwestern parts of the basin (*Cook, McKellar & Draper, 2013*; *Hall et al., 2015*).

The present-day surface expression, distribution and thickness of the Winton Formation is residual, reflecting modifications of its original distribution and thickness through multiple post-depositional structural and erosional events (*Gray, McKillop & McKellar, 2002*). It represents one of the largest formations (both in terms of thickness and areal extent) from the Cretaceous part of the Rolling Downs Group within the Eromanga Basin and occurs across three States (QLD, NSW, SA) and one Territory (NT) (Figs. 1 & 2).

The Winton Formation forms the uppermost unit of the Rolling Downs Group and the Late Triassic to Cretaceous-aged Eromanga Basin (*Exon & Senior, 1976*). It conformably and transitionally overlies the Mackunda Formation, however, due to the

transitional nature of the Mackunda to Winton Formation it is difficult to establish the base of the Winton Formation, both in outcrop and in the subsurface (*Cook, McKellar & Draper, 2013*; *Draper, 2002*). In some successions in SA where these two formations are more difficult to differentiate, the superseded name Blanchewater Formation (*Forbes, 1966*) was used in the past for the combined undifferentiated interval (*Moore & Pitt, 1985*).

An informal convention has previously been used to define the base of the Winton Formation, using the first appearance of coals or rhizomiferous sediments to define the base (*Draper, 2002*; *Gray, McKillop & McKellar, 2002*). However, coals are not always present and the majority of these transitions are only observable in cores and do not manifest in surface outcrop. This means there is uncertainty when determining the vertical and spatial distribution of the first appearance of coals or palaeosols and thus the base of the Winton Formation. Likewise, the last occurrences of marine shells, such as *Inoceramus*, are considered in numerous stratigraphic and petroleum well logs to be good indicators of the transition from the marine and tidally influenced Mackunda Formation to the freshwater fluvial and lacustrine deposits of the Winton Formation. However, in core samples, it is very difficult to confidently discern the difference between *Inoceramus*, or other marine invertebrate shells, in comparison to the freshwater-restricted invertebrate taxa, such as unionoid bivalves. Therefore, whether using the last presence of marine-tidal invertebrate taxa and/or the first indications of palaeosols, freshwater taxa or coals, the clear distinction of the Winton Formation base remains equivocal.

## Stratigraphic position of dinosaur sites

Due to the lack of contiguous Winton Formation outcrop it is practically impossible to directly trace and define the relative local stratigraphic position between any one of the many dinosaurian body-fossil sites found throughout the Winton Formation. Even at sites in relative close proximity to one another where the surface expression of fossilized bones is spaced 10 s to 100 s of meters apart it is impractical to define a local stratigraphic succession. Heavy earth-moving machinery must be used to create long and deep (4m+) stratigraphic trenches that remove the 1m+ soil and weathered vertosol-Winton Formation covering to expose enough primary sedimentological structure to enable bonebed layers to be traced laterally. This is both impractical and unrealistic in terms of developing a good understanding of local stratigraphic control between dinosaur bonebeds and site clusters.

Ground penetrating radar has been tried in places but with limited results. The clay-rich vertosol soil is variably moist at depth and possesses large voids and cracks, all of which impact the resistivity profiles and thus potential for accurate subsurface interpretations. The uniform sedimentological signature of the Winton Formation itself, being mostly siltstones to fine-grained sandstones, with small to large cemented concretionary zones also obscures lateral continuity.

Within the local context, the overall dip of strata is generally low; however, sites occur 100 s of meters to several kilometers apart and are mostly associated close to poorly defined structural features such as concealed faults or the crests of anticlines (Figs. 2, 4 and 5). Therefore, these local and poorly mapped structural features potentially create differences

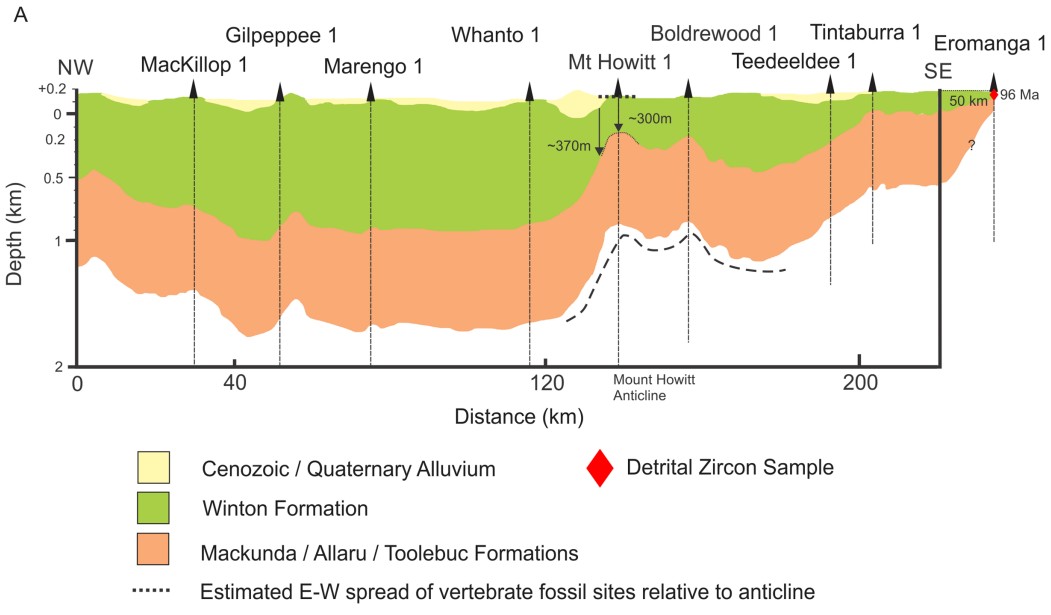

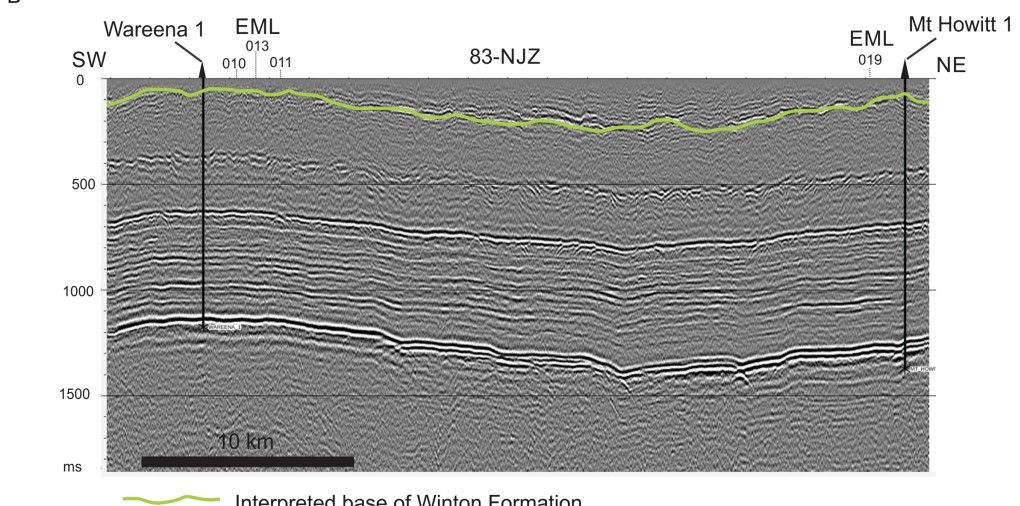

Interpreted base of Winton Formation

**Figure 4 Interpretations of Winton Formation thickness associated with the vertebrate fossil sites described here, including the type locality for *Australotitan cooperensis* gen. et sp. nov.** (A) Cross-sectional thickness of the Cenozoic/Quaternary deposits overlying the Winton Formation. Cross-section adapted from Figure 11d of (*Hall et al., 2015*) under CC-BY 4.0. Mt. Howitt 1 well, which occurs close to the northern-most Plevna Downs vertebrate fossil sites (e.g., EML019), provides an approximate estimation of 300 m of Winton Formation thickness. However, the thickness of the preserved Winton Formation rapidly increases away from the crest of the anticline on the eastern and western flanks of the Mt. Howitt Anticline. (B) Seismic Line 83-NJZ data has been reinterpreted by Santos Pty Ltd for this research project and includes the interpreted base of the Winton Formation by M.W. The base of the Winton Formation interpreted in Wareena 1 from petro-physical data is 270–300 m. Interpretation of seismic line 83-NJZ indicates the dinosaur sites EML 010-013 are at a similar structural level to Wareena 1, near the crest of the anticline. Therefore, the type locality for *A. cooperensis* gen. et sp. nov. is interpreted to be 270-m from the base of the Winton Formation (see text for additional justification). Seismic Line data available CC-BY 4.0 from Qglobe and GSQ Open Data Portal (http://qldglobe.information.qld.gov.au and http://geoscience.data.qld.gov.au/seismic/ss095410).

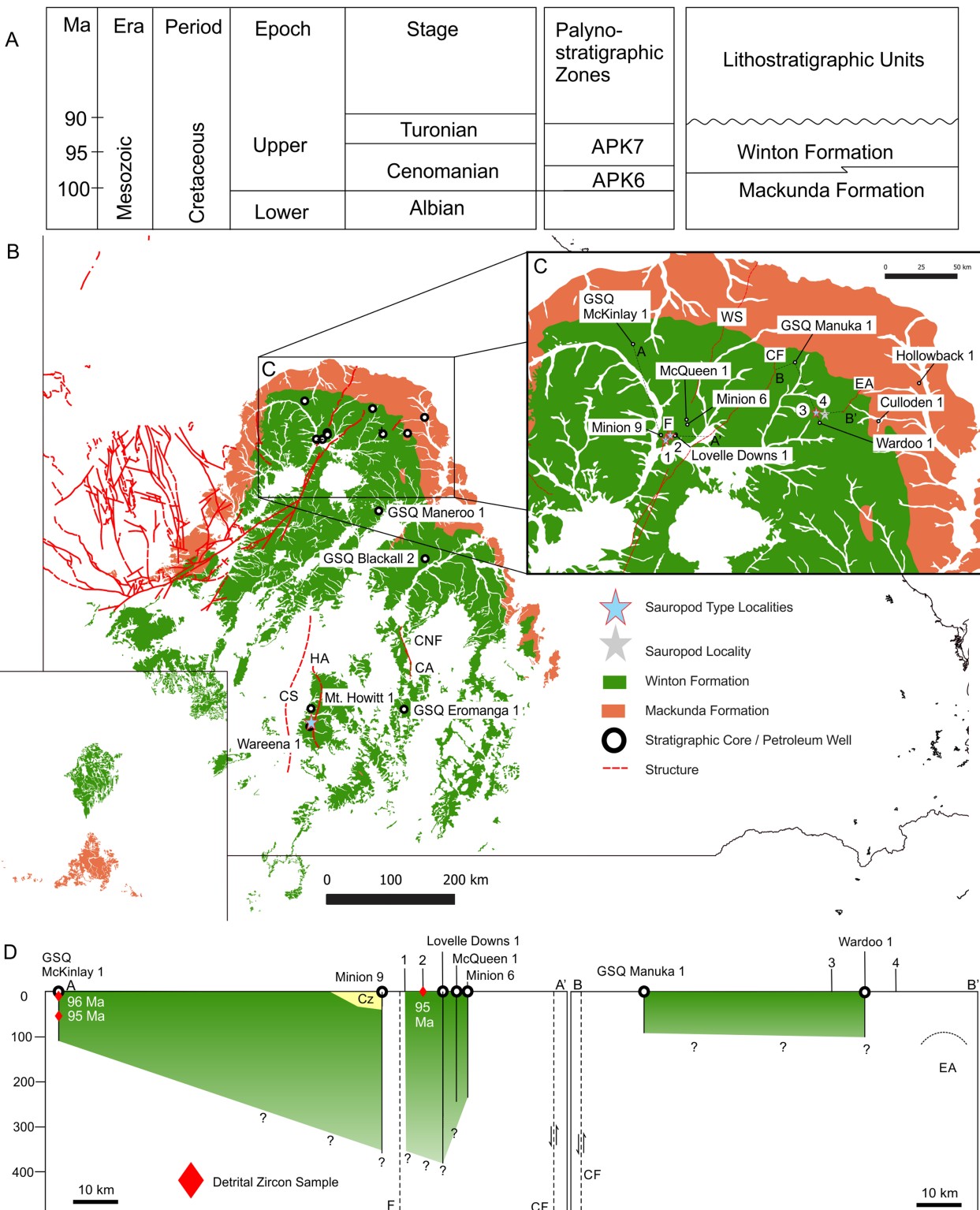

**Figure 5 Winton Formation Thickness and Age.** (A) Chronostratigraphic scheme showing the palynostratigraphic zones and lithostratigraphic units discussed in the text. (B) Mackunda and Winton Formation outcrop distribution map showing dominant structural elements associated with sauropod type localities, position of stratigraphic cores and petroleum wells used to estimate the thickness of Winton Formation at the four sauropod type localities, 1: *Wintonotitan wattsi* type locality QML313, 2: *Diamantinasaurus matildae/Australovenator wintonensis* type locality AODL085, 3:

**Figure 5** (continued)
*Savannasaurus elliottorum* type locality AODL082, 4: *Diamantinasaurus matildae* (referred) and QMF43302 discussed here from QML1333. (C) Close up of the northern Winton Formation sauropod type localities associated with stratigraphic cores, petroleum wells, geological structures (faults and anticlines). Dashed lines A-A′ and B-B′ indicate cross-sections provided in D. (D) Two generalised cross-sections of the Winton Formation, west (A-A′) and east (B-B′) of the Cork Fault, showing the relative position of the sauropod type localities in relation to the estimated base of the Winton Formation. Red diamonds indicate the core depth of zircon samples with the age in millions of years (Ma) provided for the youngest graphical detrital zircon age peak (YPP) (*Bryan et al., 2012*; *Tucker et al., 2016*). Abbreviations: CA, Canaway Anticline; CF, Cork Fault; CNF, Canaway Fault; CS, Cooper Syncline; EA, Eyriewald Anticline; F, unnamed Fault; HA, Mt. Howitt Anticline; WS, Wetherby Structure. Geographical map data from http://pid.geoscience.gov.au/dataset/ga/61754 used under CC-BY 4.0 AU. Winton and Mackunda formations retrieved for South Australia from SARIG (https://map.sarig.sa.gov.au) under CC-BY 3.0 AU; Queensland from QGlobe (http://qldglobe.information.qld.gov.au) under CC-BY 4.0; New South Wales from *Raymond et al. (2012)* (http://ga.gov.au) used under CC-BY 3.0 AU. Stratigraphic and petroleum wells, water bores and geological structures retrieved from Qglobe (http://qldglobe.information.qld.gov.au) under CC-BY 4.0.

in vertical profile position of 10 s to 100 s of meters between individual fossil sites. Although the sites may be regarded as topographically similar and assumed to be contemporaneous, this is unverified, and concealed stratigraphic differences could be greater than expected. Such unverified stratigraphic position makes determining whether the taxa recovered from one or more sites are sympatric near impossible. This is especially relevant for the Winton Formation where there is no control on relative positions of bonebeds or the sedimentation rate of these deposits and the Winton Formation unit as a whole.

Regionally, defining the relative stratigraphic position of dinosaur fossil sites is equally difficult with the added complexity of; (1) regional subsurface structuring (*Exon & Senior, 1976*; *Hoffmann, 1989*); (2) rapid exhumation and pre-Cenozoic erosion of the Winton Formation (*Keany, Holford & Bunch, 2016*; *Rodgers, Wehr & Hunt, 1991*); (3) Cenozoic basin filling (*Cook & Jell, 2013*; *Day et al., 1983*; *Krieg et al., 1990*); (4) deep Winton Formation chemical weathering (*Idnurm & Senoir, 1978*; *Senior & Mabbutt, 1979*); (5) broadly defined palynomorph zones with no refinement within the Winton Formation (*Monteil, 2006*); and (6) considerable geographical distance between localities ranging from ~105 km to over 500 km apart.

The multiple levels of uncertainty at both local and regional scales, over such an extensive and thick geological formation, renders the level of stratigraphic accuracy needed for meaningful chronological comparisons between faunas difficult, and even more so when comparing fauna from semi-contemporaneous formations from separate basins. Such uncertainty requires a greater future effort to place each fauna within a local and regional context, currently leaving only broad-sweeping generalisations possible (*Wilkinson, Hocknull & Mackenzie, 2019*).

We have attempted here to place the type localities of all four sauropod taxa into a regional stratigraphic context, but local stratigraphic context for each site is near impossible to ascertain. For the southern-central Winton Formation sauropod sites we begin by using a published interpretation of seismic and well data that produced an approximation of Winton Formation thickness (*Hall et al., 2015*) (Fig. 4A). Importantly, it provides a NW-SE cross-sectional interpretation across the crest of the Mt. Howitt anticline, the key geological structure associated with all new dinosaur sites described here.

All of the new dinosaur sites occur within 5 km of the western flank of the Mt. Howitt anticline with one locality (EML019) located close to the Mt. Howitt 1 well (*Delhi Petroleum, 1966*). The thickness of the Winton Formation at Mt. Howitt 1 approximates 300 m, with thicker sections preserved on the flanks of the Mt. Howitt anticline (Fig. 4A).

Next, we used well and seismic data proximal to the sites to estimate the thickness of the Winton Formation closest to the dinosaur sites. The stratigraphic position of the type locality for *Australotitan cooperensis* (EML011(a)) relative to the base of the Winton Formation was estimated by examining data from nearby petroleum well bores, Wareena 1-5 (*Gauld, 1981*; *Lawrence, 1998*; *Lowman, 2010*; *Robinson, 1988*; *Turner, 1997*) and Navalla 1 (*Boothby, 1989*). Wareena 4 is located approximately 1.33 km to the east of EML011. In addition to this, seismic data was investigated to determine the influence of faulting and structural features within the vicinity of the dinosaur localities (*Delhi Petroleum, 1991*; *Finlayson, 1984*; *Flynn, 1985*; *Garrad & Russel, 2014*; *Seedsman, 1998*).

Data from the petroleum well bores is limited, as no cores were taken, and the lithological descriptions do not indicate the clear presence of coal or palaeosols, thus determining the base of the Winton Formation or top of the Mackunda Formation was not possible. The closest stratigraphic core, GSQ Eromanga 1, occurs 130 km to the east, where the base of the Winton Formation is interpreted to be 164 m below ground surface (*Almond, 1983*).

Without a good lithological control, we considered wireline petrophysical logs to interpret the base of the Winton Formation. Changes in petrophysical character of the gamma-ray, sonic, resistivity and self-potential wireline logs have previously been used to define the Mackunda and Winton Formations in the subsurface (*Gray, McKillop & McKellar, 2002*; *Moore, Pitt & Dettmann, 1986*). We used these same features to pick the base of the Winton Formation with a thickness of 270–300 m for the Wareena and Mt. Howitt wells.

We correlated the petrophysically interpreted base of the Winton Formation at Wareena 1 and Mt. Howitt 1 wells with the uppermost prominent seismic reflection event for seismic line 83-NJZ (Fig. 4B). This seismic line includes the Mt. Howitt 1 and Wareena 1 wells and runs in a NNE-SSW direction close to the axis of the Mt. Howitt anticline (Fig. 2B). This seismic reflection event is not continuous which is likely due to small scale faulting. This again reflects the uncertainty likely to pervade local stratigraphic differences mentioned above. Interpretation of the seismic line indicates that the Wareena 1 and Mt. Howitt 1 wells are located near to the crest of the Mt. Howitt anticline and are therefore likely to contain the thinnest section of preserved Winton Formation. Therefore, on the basis of the four dinosaur localities (EML010-013) being located in close proximity to the Wareena 1 well on the crest of the Mt. Howitt anticline, the sites are likely to be 270–300 m from the base of the Winton Formation (*Wilkinson, Hocknull & Mackenzie, 2019*). This is supported by previous interpretations (*Hall et al., 2015*) (Fig. 4A).

Applying similar methods to the northern Winton Formation sauropod type localities, we focused our assessment of the Winton Formation base and thickness by assessing stratigraphic and petroleum wells found closest to the type localities of *Diamantinasaurus*

*matildae* and *Australovenator wintonensis* at AODL85 (*Hocknull et al., 2009*); *Wintonotitan wattsi* at QML313 (*Hocknull et al., 2009*); *Savannasaurus elliottorum* at AODL82 (*Poropat et al., 2016*); and the referred specimen of *Diamantinasaurus matildae* at QML1333/AODL127 (*Poropat et al., 2016*) (Fig. 5A).

The type localities of *D. matildae* and *W. wattsi* are close to one another (~3.5 km apart) and occur 2.6 km and 1.1 km east of a concealed (unnamed) fault respectively. The closest petroleum wells are Minion 9 (*Pangaea Resources, 2013*) to the west of the concealed fault and fossil sites, and Lovelle Downs 1 (*Watson, 1973*) that occurs east of the concealed fault and east of the type localities. Lovelle Downs 1 is 4 km due east of the type locality for *D. matildae*.

At Lovelle Downs 1, the base of the Winton Formation was assessed to be 880 feet (268 m) (*Watson, 1973*); however, lithological descriptions indicate first coal at 1,210 feet (368 m); therefore, we agree that the base of the Winton Formation is at least 268 m from surface but it is more likely to be 368 m or more from the surface. At Minion 9, west of the type localities and the unnamed fault, the base of the Winton Formation was assessed on first coals to be 352 m from the surface but with 31.6 m of overlying Cenozoic sediments; thus a thickness of 316 m (*Pangaea Resources, 2013*). We agree with this assessment (Fig. 5).

Both type localities are situated over a structural low termed the Lovelle Syncline/ Depression, and occur about 18–20 km west and downthrown of a major fault, termed the Cork Fault, which would provide the structural means for a relatively thick Winton Formation across this area. Therefore, we propose a Winton Formation base from surface for the type localities of *D. matildae* and *W. wattsi* of at least 350 m (Fig. 2A).

The closest stratigraphic core to the type localities of *D. matildae* and *W. wattsi* comes from GSQ McKinlay 1 (*Hoffman & Brain, 1991*), 70 km to the northwest and very close to the Winton Formation outcrop edge (Fig. 5). The Winton Formation base at GSQ McKinlay 1 is interpreted to be approximately 112 m from the surface although no coals are present. *Inoceramus* shell is identified at ~125 m, therefore, we agree that the base of the Winton Formation is at around 112 m, but it could be higher in the core. Therefore, there is a difference of over 200–250 m of Winton Formation thickness between the Minion 9 and Lovelle Downs 1 wells (and type localities), relative to the closest stratigraphic core (GSQ McKinlay 1).

In contrast, the type locality of *S. elliottorum* and another sauropod locality preserving a specimen referred to *D. matildae* (QML1333) occur approximately 70 km to the east of the Cork Fault on the upthrown section, and approximately 18 km west of the Eyriewald Anticline. These sites are located closer to the Winton Formation outcrop edge than the type locations for *D. matildae* and *W. wattsi* and therefore we would expect them to be closer to the base of the Winton Formation.

The closest petroleum well is Wardoo 1 (*Exoma Energy, 2013*), positioned 6–7 km south and southwest of the *S. elliottorum* type locality and QML1333 respectively. The base of the Winton Formation at Wardoo 1 is reported as 311 m, however, the first coals are indicated at 90 m (*Exoma Energy, 2013*). Therefore, we treat the reported depth and thickness of the Winton Formation at Wardoo 1 with some caution and propose that it is

more likely closer to 100 m (Fig. 5). Wardoo 1 and the dinosaur localities are close to the Winton Formation outcrop edge, which is similar to that seen in the stratigraphic cores of GSQ McKinlay 1 (Winton Formation base at 112 m) (*Hoffman & Brain, 1991*) and GSQ Manuka 1 (Winton Formation base at ~92 m) (*Balfe, 1978*); therefore, we propose a 90 m depth based on the first appearances of coals as a more realistic estimate for the base of the Winton Formation at Wardoo 1. Therefore, we propose a depth to base of Winton Formation for the *S. elliottorum* type locality and QML1333 to be less than 100 m (Fig. 5).

## Summary of the stratigraphy of the Winton Formation sauropods

Taken together, our assessment of the depth to base of Winton Formation in relation to the four sauropod type localities illustrates the uncertainty discussed above in relation to a lack of clear delineation for the base of the Winton Formation, and the relative stratigraphic positions of the sites both locally and regionally. On the available published data from stratigraphic cores, wells and seismic lines located closest to the type localities, we propose that; (1) the *S. elliottorum* type locality and QML1333 site with a referred specimen to *D. matildae* are positioned less than 100 m above the base of the Winton Formation; (2) the new type locality for *A. cooperensis* is positioned somewhere between 270 and 300 m above the base of the Winton Formation; and (3) the type localities of *D. matildae* and *W. wattsi* are positioned approximately 350 m (or somewhere between 316 and 368 m) above the base of the Winton Formation (Fig. 5).

Although this proposed series of positions above the base of the Winton Formation likely constitute real stratigraphic, and thus chronological differences between the sauropod type localities, we urge caution in using this proposed stratigraphic sequence for palaeontological interpretations due to the diachronous uncertainty of it and the unknown spatiotemporal sedimentation rates across the entire Winton Formation.

## WINTON FORMATION AGE

The Winton Formation was assigned a Late Albian to Cenomanian chronostratigraphic age on the basis of spore-pollen zonation (*Monteil, 2006*). The presence of Late Albian index species *Phimopollenites pannosus* to Cenomanian index species *Hoegisporis uniforma* (=*Appendicisporites distocarinatus*) within the Winton Formation reflects this assessed chronostratigraphic age range (*Helby, Morgan & Partridge, 1987*). On the basis of well-preserved palynomorphs indicating the *Coptospora paradoxa* and *Phimopollenites pannosus* zones, a latest Albian age was interpreted for a surface locality located close to the type localities of *Diamantinasaurus matildae*, *Wintonotitan wattsi* and *Australovenator wintonensis* (*Dettmann, Clifford & Peters, 2009*). The palynomorphs from this site indicated an age of no older than Late Albian. With the absence of Cenomanian indicator species such as *Hoegisporis uniforma* and *Appendicisporites distocarinatus* a Cenomanian age could not be given. The type localities for three dinosaurian taxa (*D. matildae*, *W. wattsi* and *A. wintonensis*) from nearby sites were thus considered to be latest Albian in age (*Hocknull et al., 2009*).

Subsequent to this, two independent age assessments of the Winton Formation were conducted using modelled U-Pb radiometric assessments of detrital zircons, and calculated age probability distributions, to determine the maximum depositional age of dinosaurian fossil sites (*Bryan et al., 2012*; *Tucker et al., 2013*). Modelled interpretations from these probability distributions were used to propose true depositional ages for the layers from where the zircons were sampled and to construct an age profile for the Winton Formation, defined into lower, middle and upper Winton Formation (*Tucker et al., 2017*; *Tucker et al., 2016*). See Tucker (*Tucker et al., 2016*; *Tucker et al., 2013*) for explanations of each age model type and methodology used.

The reliability of the detrital zircon dating technique for sedimentary sequences will not be reviewed here, having been discussed and assessed by many others who have identified biases, methodological issues, and interpretative problems with detrital zircons (*Allen & Campbell, 2012*; *Andersen, Elburg & Magwaza, 2019*; *Coutts, Matthews & Hubbard, 2019*; *Horstwood et al., 2016*; *Johnstone, Schwartz & Holm-Denoma, 2019*; *Klötzli et al., 2009*; *Košler et al., 2013*; *Sharman & Malkowski, 2020*).

Considering this uncertainty, the results so far produced for the Winton Formation need to be treated cautiously. Nevertheless, they all indicate a probable temporal age range of between 103 to 92 million years ago (Late Albian to earliest Turonian) for the maximal depositional ages of portions of the Winton Formation.

Key to determining the depositional age and age range for the Winton Formation is the source of the youngest zircon grains that likely came from eastern Australian volcanicity that continued throughout the Early to mid-Cretaceous (*Bryan et al., 2012*; *Tucker et al., 2017*). Substantial volumes of mostly silicic pyroclastic material and coeval first cycle volcanogenic sediment accumulated in the Eromanga Basin during deposition of the Winton Formation (*Bryan et al., 2012*). This material was transported over very large distances along with the semi-contemporaneous development of a southwest draining river system dubbed the 'Ceduna River'. The 'Ceduna River' depocentre was the Ceduna delta, a very large deltaic lobe that filled the tectonically subsiding southern Australian Bight Basin, which formed the contemporaneous paralic White Pointer supersequence (*Espurt et al., 2009*; *King & Mee, 2004*; *Lloyd et al., 2016*; *Sauermilch et al., 2019*; *Totterdell & Krassay, 2003*).

However, it is unclear, not only of the magnitude and continuity of explosive events, but also the ultimate cessation of volcanicity. If volcanicity ceased before the end of Winton Formation deposition, this raises the possibility of erosion and reworking of older zircons within the Winton Formation without the arrival of new zircons entering the system, which could obscure a more refined true depositional age, and this may impact the ages of the four type locality deposits.

### Age of the dinosaur sites

A single population of detrital zircons has been published for the *D. matildae* type locality (*Bryan et al., 2012*), but no detrital zircon populations have been published for the other three type localities. The closest stratigraphically controlled detrital zircon populations for all three northern sauropod taxa, *D. matildae*, *W. wattsi* and *S. elliottorum*, comes from

GSQ McKinlay 1 (2 samples) (*Tucker et al., 2016*). Whilst for the southern-central Winton Formation sites, the closest stratigraphically controlled detrital zircon population comes from GSQ Eromanga 1 (1 sample) (*Tucker et al., 2016*).

Of these four zircon populations recovered closest to our type localities, the two GSQ McKinlay 1 samples were taken closest to the Winton Formation base, at 102.7 m and 58 m from the Winton Formation base respectively. The lowest sample was defined to represent the 'middle' Winton Formation and the higher sample the 'uppermost' Winton Formation (*Tucker et al., 2017*; *Tucker et al., 2016*). The stratigraphically lower sample returned modelled zircon ages of between $92.1 \pm 1.8$ Ma (YC1$\sigma$ (+3)) to 95 Ma (YPP), whilst the stratigraphically higher sample returned discordant older ages of between $93.5 \pm 4.4$ Ma (Weighted average (+3)) and 98 Ma +0.9/-4.1 Ma (TuffZirc (+6)) (see *Tucker et al., 2016* for model descriptions).

The next highest zircon population was taken from GSQ Eromanga 1 within the core, at approximately 146 m above the Winton Formation base and defined as the 'lower' Winton Formation (*Tucker et al., 2017*; *Tucker et al., 2016*), 44 m higher than the 'uppermost' Winton Formation of GSQ McKinlay 1. This sample returned modelled maximum depositional ages ranging between $93.1 \pm 1.1$ Ma (YSG) and 101.1 +1.3/−1.4 Ma (TuffZirc (+6)), representing a similar modelled age range compared to the 'uppermost' Winton Formation of GSQ McKinlay 1. Of note, a similar age range was also given for a sample taken between 20.8–35.8 m below surface at GSQ Blackall 2 stratigraphic core, to the north east of GSQ Eromanga 1 (*Tucker et al., 2016*). This sample comes from the 'lower' Winton Formation, taken between 113–128 m from the Winton Formation base (~149 m below surface) (*Coote, 1987*). This zircon population returned modelled ages ranging between $93.4 \pm 1.8$ Ma (YPP) and 98.7 +2.2/−5.3 Ma (TuffZirc (+6)).

Finally, the highest zircon population was sampled at the *D. matildae* type locality, which sits at least 350 m from the Winton Formation base. This sample sits twice to three times higher in the Winton Formation when compared to the 'lower' Winton Formation GSQ Eromanga 1 and GSQ Blackall 2 and 'middle' to 'uppermost' Winton Formation of GSQ McKinlay 1 (*Tucker et al., 2017*; *Tucker et al., 2016*). The ages for the type locality include a single youngest grain age of $94.29 \pm 2.8$ Ma and two youngest age peaks at ~95 Ma and ~102 Ma (*Bryan et al., 2012*; *Greentree, 2011*).

Considering each zircon sample's stratigraphic position above the base of the Winton Formation with each sample's youngest single grain age, it would be expected that the sample taken closest to the base of the Winton Formation would return the oldest youngest single grain age, and that the sample taken furthest from the Winton Formation base would have the youngest single grain age. This is not the case, the lowest sample, taken 58 m from the Winton Formation base has a single grain age of $93.4 \pm 1.5$ Ma, which is within the error of the highest sample (350 m+) single grain age of $94.29 \pm 2.8$ Ma. The youngest single grain ages for the intermediate samples are also within error of the lowest and highest zircon populations; therefore, the maximal depositional age based on youngest single grain detrital zircons is similar throughout the 350 m+ sampled Winton Formation and does not indicate a change in age with stratigraphic position.

Taking the youngest age peak for the zircon populations, a similar situation exists, with the sample taken closest to the base of the Winton Formation returning an age of 95 Ma and the sample taken furthest from the base of the Winton Formation also returning an age of 95 Ma.

Such similarities in ages across 350 m+ of Winton Formation can potentially be reconciled in several ways. The similarities in ages could represent the loss of new zircons entering the system after the cessation of volcanicity, resulting in reworking of the youngest available grains up the profile. Or, the sedimentation rate across the Winton Formation was exceptionally variable across the basin producing considerable differences in depositional thicknesses across relatively small geographical areas. Alternatively, the base of the Winton Formation may be diachronous across the basin, resulting in areas with similar positions relative to the base of the Winton Formation being of dissimilar ages. It is conceivable that one or more, or even all, of these processes were operating during deposition of the Winton Formation. We note that all samples within the Winton Formation contain recycled detrital zircons and as yet no in situ pyroclastic beds have been recorded.

The detrital zircon samples taken closest to our new dinosaur sites is GSQ Eromanga 1 (*Almond, 1983*) and as discussed above the sample comes from close to the base of the Winton Formation (~146 m). The type locality for *Australotitan cooperensis* is estimated to occur 270–300 m above the base of the Winton Formation, therefore, twice as high within the sequence relative to GSQ Eromanga 1, located 130 km east of it. The age range for this detrital zircon population is also within the error of the samples from the northern Winton Formation, with a youngest single grain of 93 ± 1.1 Ma, and ranging up to 101.1 +1.3/−1.4 Ma (*Tucker et al., 2016*). The youngest population peak sits at 96 Ma, slightly older than the lowest samples from the northern Winton Formation stratigraphic cores. We therefore consider that the age of the type locality EML011(a) and other associated localities have a maximum depositional age of between 93–96 Ma.

## Summary of the age of the Winton Formation sauropods

The combined uncertainties expressed above in regards to the stratigraphic positions of all of the type localities, uncertainties with detrital zircon dating, and the lack of other techniques to better refine the absolute ages of the deposits, the actual age of all four taxa remains equivocal. A maximum depositional age of mid-Cenomanian (~95–96 Ma) for the four type localities discussed here is favoured but with the caveat that all four type localities could be considerably different in relative and absolute age. Any further refinement will require much greater control of both stratigraphy and chronometric age. We note that the uncertainty of the maximum depositional age has been suggested to range for the 'lower', 'middle' and 'upper' Winton Formation of between 92–94 Ma (*Tucker et al., 2016*). We generally agree with this level of uncertainty but propose a slightly greater range (92–96 Ma).

The uncertainty surrounding the chronometric dates for the maximum depositional age of either portions of, or the whole, Winton Formation presents significant difficulties when proposing testable hypotheses focused on local or regional sauropod biogeography,

palaeoecology and evolution. Additionally, these stratigraphic and age uncertainties further render chronological comparisons of the Winton Formation dinosaurian fauna with the semi-contemporaneous Griman Creek Formation at Lightning Ridge (*Bell et al., 2019b*) of limited value.

## DEPOSITIONAL & TAPHONOMIC SETTINGS

The dinosaurian skeletal remains from these southern-central Winton Formation sites are exclusively represented by sauropods. In spite of a large number of sites having been excavated over the last decade, only the remains of a freshwater turtle (?chelid) and an isolated poorly preserved hyriid bivalve represent fauna not attributable to sauropods (*Hocknull et al., 2019*). There is a distinct lack of higher taxonomic representation relative to the fauna from the northern Winton Formation sites. Currently missing fauna from the southern-central Winton Formation include gastropods, insects, teleost fish, lungfish, crocodilians, pterosaurs, theropods, ornithopods, and ankylosaurs (Table 1).

Preservation of sauropod remains range from isolated, fragmentary remains that have undergone considerable pre- and post-depositional modifications through to articulated partial skeletons preserved within thick cemented siltstone concretions (Figs. 6I & 6K). Preserved alongside these sauropod remains are macrofloral remains ranging from isolated leaves to thick layers of woody debris (Figs. 6A–6I). In addition, ichnological evidence points to considerable bioturbation (dinoturbation) at EML011, which includes the type locality of *Australotitan cooperensis*. (Fig. 6J; Figs. 7C & 7D and Figs. 8A–8N). One such feature is a near 100 m long trampled silt and bonebed unit, also preserving a partial associated skeleton.

## SITE DESCRIPTIONS

At least fourteen dinosaur bone-bearing fossil sites have so far been discovered in the southern-central Winton Formation. These sites are divided into two areas of northern and southern Plevna Downs Station, located 85 km west of the Eromanga Township (Fig. 2B). The type locality for type specimen of *Australotitan cooperensis* comes from the southern Plevna Downs Station, EML011(a), with referred remains from EML010 and EML013.

**EML 010.** Material; EMF106 & EMF164. EML010 surface scatter was discovered in 2005 within the present-day anastomosing channeled creek system. The bones occur between two weathered units of resistant siltstone-mudstone cemented rock both running in a general East-West direction. The bone scatter occurs between these two units with no surface bone found to the north or south of them. It represents a discrete site with the entire deposit being confined to a single area of surface scatter approximately 1,500 m$^2$. The majority of the surface scatter was made up of fragmented, rounded and winnowed cortical and cancellous bone fragments indicating a long period of surface exposure, but relatively little distal transport from its subsurface source matrix.

Bone preserved with adhering cemented siltstone-mudstone indicates that the bones originated from one of the cemented units and subsequent surface exposure and weathering has broken up the remains into small pieces. Collections of surface specimens

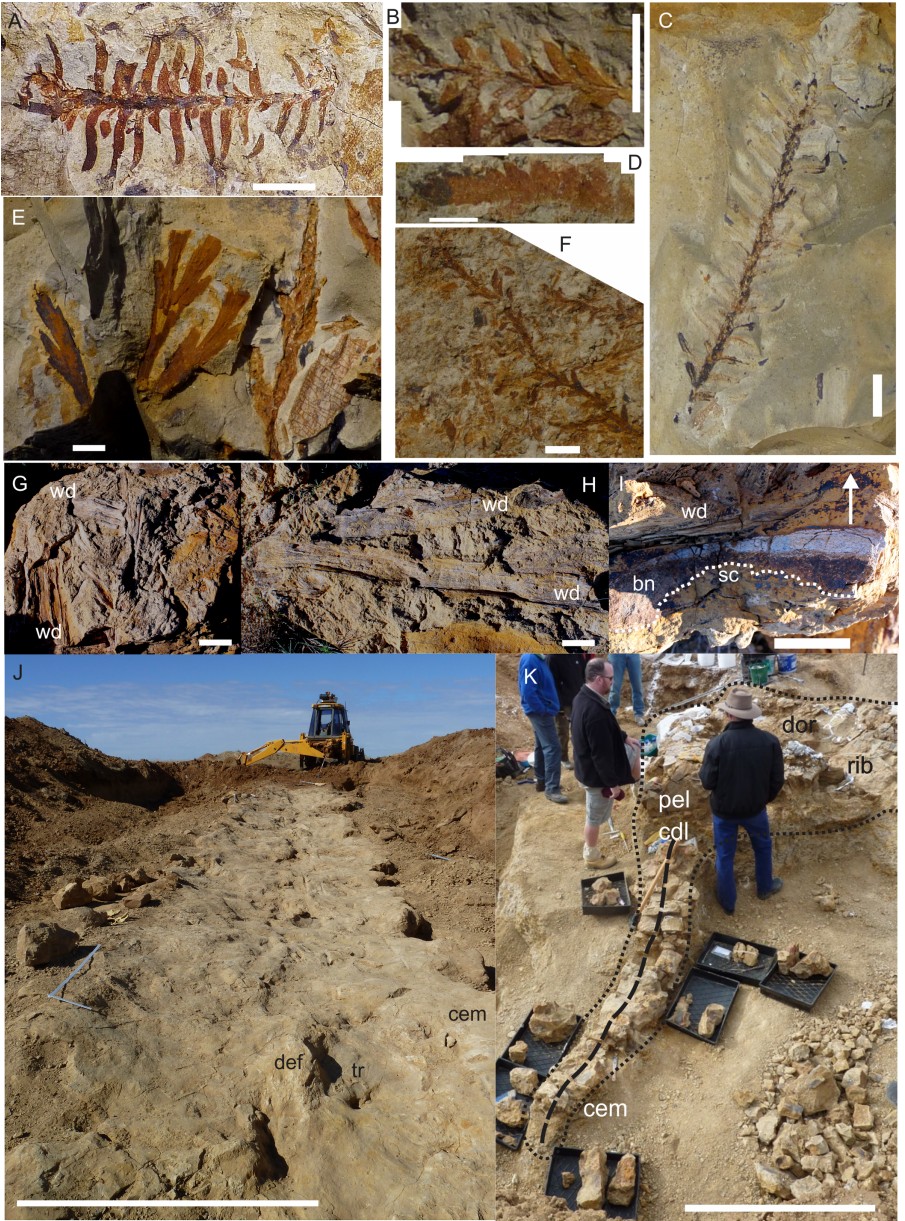

**Figure 6 Preservational examples of leaves, wood debris, bone debris, trampled sediments and articulated remains from southern Plevna Downs sites (EML011, 012 and 013).** (A–F) Leaves preserved indicate a dominance of conifers (Pinophyta) and ferns (Pterophyta). (A) EMF177, conifer twig with leaves. (B) EMF175, ?Bennettitalean leaf. (C) EMF176, conifer twig with poorly preserved leaves. (D) EMF174, Pterophyte leaf (?*Cladophlebis* sp.). (E) EMF172, Pterophyte leaf (?*Sphenopteris* sp.). (F) EMF173, conifer leaf 'mat'. (G & H) Woody (wd) debris impressions in layers showing preferred orientation within thick sections of cemented siltstone. (I) Bone (bn) and woody debris in cross-section with bone occurring at the base of the woody debris beds (arrow indicating upward direction). Underside of bone either corroded or eroded off creating a scoured (sc) underside (EML013). (J) Massive ichnological features showing trampled and cemented (cem) siltstone horizon, sediment deformation buldges (def) and partial sauropod foot imprints (tr) (EML011) (Scale Bar = 1 m). (K) Articulated sauropod skeleton from EML012 preserved within a siltstone concretion, including the torso and tail. Identifiable elements include ribs (rib), dorsal vertebrate (dor), pelvic elements (pel) and caudal vertebrae (cdl). Scale Bars = 10 mm (A–F), 50 mm (G–I), 100 cm (J & K).     

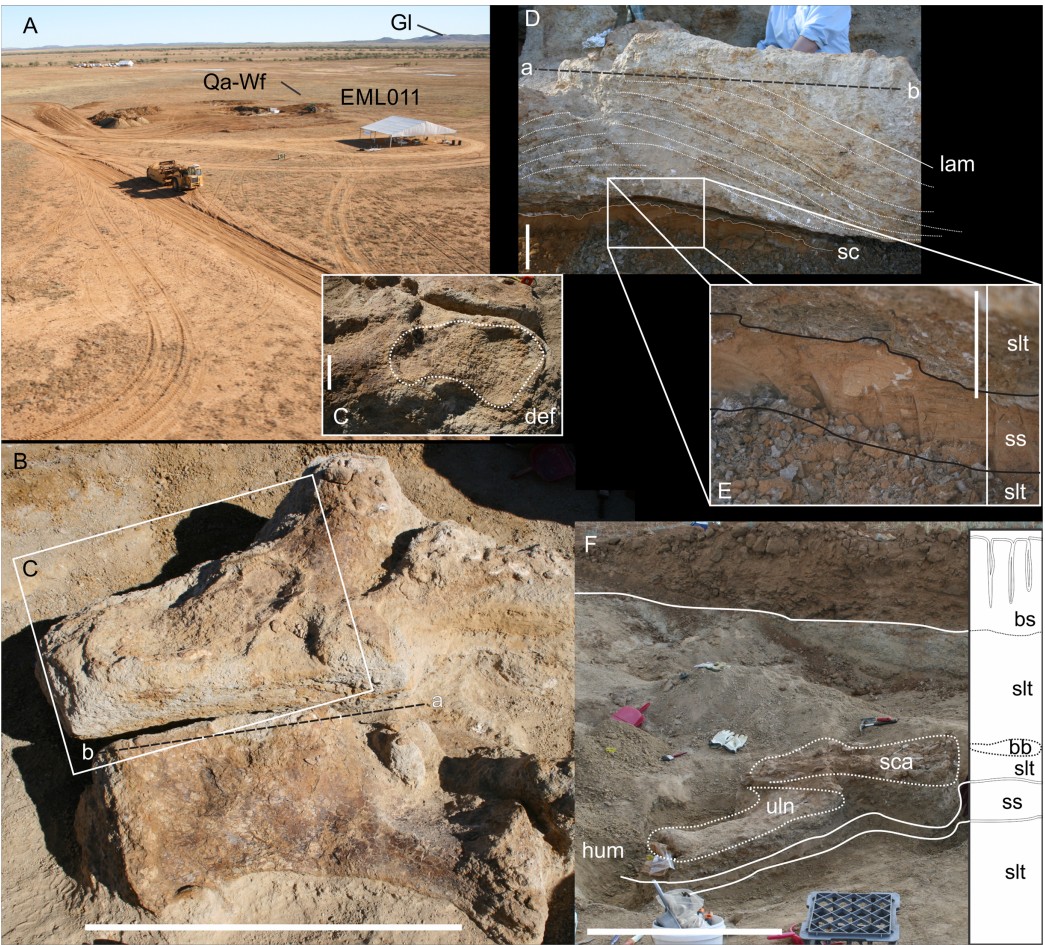

**Figure 7 EML011, type locality of *Australotitan cooperensis* gen. et sp. nov. site sedimentology and taphonomy.** (A) Site overview showing excavation pit, distant weathered geochemically weathered Glendower (Gl) and more proximal weathered Winton (Wf) and Quaternary alluvial (Qa) deposits. (B) Semi-articulated pubes and ischia from *A. cooperensis* gen. et sp. nov. with mediodorsal surfaces of each pubis facing upwards with the dislocated ischia in close articular approximation (arrows indicate d, distal; p, proximal; rd, right dorsal and ld, left dorsal). (C) In situ ovo-lobate deformation (def) of pubis. (D) Cross-section (a-b) of sediment beneath pelvis showing downwardly deformed laminations (lam) of the siltstone (slt) above E. (E) a lower surface-scoured sandstone (ss) layer. (F) Associated humerus (hum), ulna (uln) and scapula (sca) of *A. cooperensis* gen. et sp. nov. within the shallow stratigraphy of the site, including the surface vertosol (blacksoil, bs) that transitions into underlying Winton Formation siltstone (slt) with the bonebed (bb). A thin sandstone (ss) layer occurs below the siltstone and bonebed. Scale bars = 10 cm (C–E) and 100 cm (B & F).

in 2005, 2006, 2010 and 2014 along with excavated subsurface collections in 2006 and 2014 revealed a large number of bone fragments representing pieces from sauropod axial and appendicular elements.

There is no obvious element duplication; however, some remains indicate the presence of two different-sized sauropod individuals within the deposit. At this point, we have separated the identifiable elements of the large individual from those that are from a smaller individual, or those pieces that are unidentifiable. The identifiable remains from the large individual include pieces of a massive femur, pieces of at least one very large

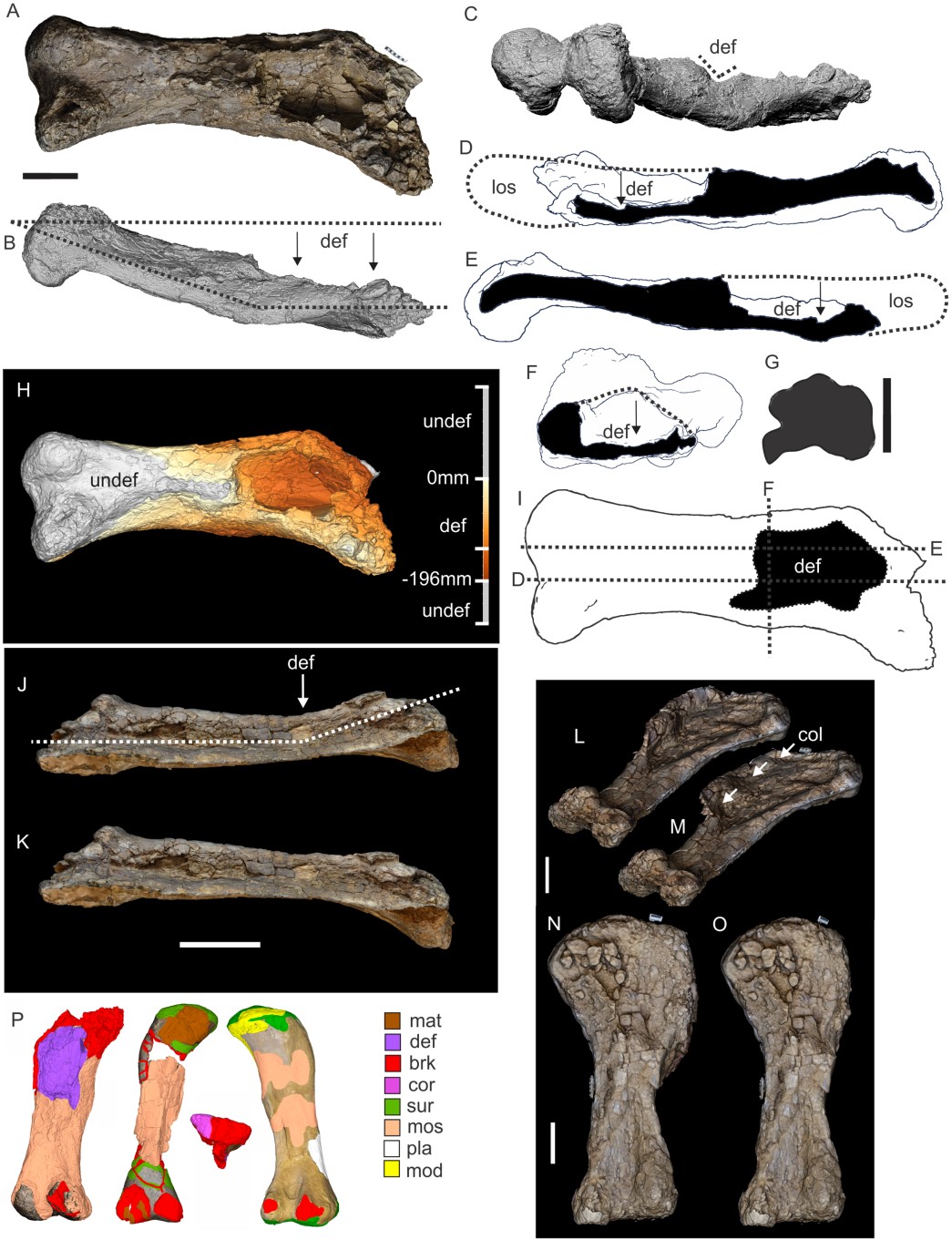

**Figure 8 Examples of sauropod bone preservation and taphonomic alteration, including coloured reference scheme for 3-D models.** (A–F) & (H & I). *A. cooperensis* gen. et sp. nov. (EMF102), right femur showing vertical displacement via a localized downward force acting upon the bone to deform the shaft. (A) 3-D model showing the upward-facing in situ surface. (B) 3-D model in medial view showing the relative downward deformation that has occurred to the bone from horizontal orientation. (C) 3-D model in distomedial view showing a triangular-shaped depressed deformation of the femoral shaft, likely from a manual I claw. (D, E, & F) The large ovo-lobate deformation structure impacting the proximal shaft of the femur and connected to C. (G) Sauropod manus footprint adapted from Fig. 6, p 11 (*Santos, Moratalla & Royo-Torres, 2009*) (CC-BY-4.0) for comparison with crush outline provided in I. (H) Depth of deformation of the depressed (surface) cortical bone. (I) Edge-detected 3-D model outline

**Figure 8** (continued)
with interpreted outline of depression and indicating sauropod manus-like shape. (J & K). EMF102, right ulna showing deformation of the distal shaft in J and the digitally retrodeformed shaft in K. (L–O) The right humerus illustrating the outward collapse of the deltopectoral crest (that occurred during excavation) (M & N) and the digitally retrodeformed deltopectoral crest (L & N). (P) Coloured reference scheme for 3-D models illustrating preservational, taphonomic and 3-D model observations. Abbreviations: brk, broken or missing connecting surfaces; col, collapsed deltopectoral crest; cor, corroded surface; def, deformation; los, bone loss; mat, obscuring matrix; mod, poor model alignment/surface; mos, mosaic-fractured cortical bone surface; pla, plaster/infill; sur, surface/cortical bone missing; undef, undeformed. Scale bars = 20 cm.     

somphospondylous presacral vertebra, fragments of appendicular limb (ulna) and rib shaft pieces. The putative smaller individual is represented by a partial caudal vertebra and fragments of podial elements.

Few fragments could be pieced together with most suspected joins having long weathered away due to long-term exposure. Most are of limited morphological use due to their poor preservation; however, on comparison with other better-preserved specimens from other sites, the large individual represents the largest sauropod specimen so far recorded.

Winnowing and rounding through sand-blasting of the internal cancellous bone is present in most surface collected elements. At depth, the bone fragments are found within a lag of Paleogene-aged silcrete gibber stones close to the transition between the vertosol and underlying Winton Formation siltstone. These gibber stones most likely became incorporated within the vertosol during soil formation processes as lag and channel fill. Therefore, the bone deposit can be considered to be a lag and redeposit derived from the breaking down of the cemented Winton Formation siltstone unit containing the vertebrate fossil remains. Subsequent mixing within the channel has concentrated bone fragments within the vertosol profile, and recycling of these fragments within the soil profile makes it impossible to determine the original relationship of the bones to one another within the siltstone unit itself. However, the total confined spread of the fragments and uniform preservation indicates no secondary bone mixing from other localities. We conclude from this that an in situ siltstone shelf preserving the dinosaur skeletal remains was broken apart through the combined weathering and development of the vertosol with the recycling actions of a small palaeochannel sometime during the Quaternary.

One additional possible taphonomic agent at this particular site is bioturbation of the deposit by wombats. A tooth of a wombat, probably a species of *Lasiorhinus* (Hairy-nosed Wombat), was recovered within the vertosol during initial excavations in 2005. Although there are no preserved indications of burrows, the presence of wombats in the area in the past does offer an alternative mechanism for dislocation of fossil remains at depth and transport of these remains to the surface. The burrowing behaviour of wombats may have also contributed to the surface expression and bone fragments in Winton, at QML1333 (*Hocknull, 2005*).

Once exposed at the surface, lateral movement of the bone fragments has been limited due to the very low topographic relief and channel velocity during flooding events. It was observed in 2011 that exposed bone fragments can withstand high volume flow during

large-scale flood events, whereby the specimens move very little during the event and remain exposed at the surface on pedestals of sediment. So although flooding occurs within the channel system, the impact of this on the surface expression of dinosaur bones seems minimal. Together, these observations suggest that EML010 represents the longest-term surface expression of dinosaur fossils so far found in the region.

EML010 is unique within the sites so far recovered from Eromanga having experienced the greatest amount of surface weathering of any of the sites and the only site demonstrating the impact of winnowing by windblown abrasion. This form of bone weathering is unique in all of the sites so far observed in the Queensland section of the Winton Formation. Thus, EML010 probably represents one of the most weathered dinosaur localities from the Winton Formation that still preserves bone at the surface.

Fossil bone observed by SAH in 2002 and 2011 at the Museum of Central Australia, Alice Springs, Northern Territory, and via (A. Yates, 2019, personal communication), represent vertebrate fossil remains from the Winton Formation located in the Munga-Thirri (Simpson) Desert. These bone fragments show similar levels of surface weathering and wind-blown sand abrasion. The proximity of the Eromanga and Northern Territory sites to the sand dunes of the Munga-Thirri Desert provides adequate mechanisms for sand abrasive conditions to be present especially throughout the intensified aridity of the late Quaternary (*Hocknull et al., 2007*; *Hollands et al., 2006*; *Maroulis et al., 2007*). In comparison, the dinosaur localities of Winton and Isisford to the north and east are distal to these dunes and probably did not experience this kind of abrasive surface weathering.

**EML011(a–c).** Material; EMF102, EMF103 & EMF111. EML011 was first thought to be a single large surface scatter over an area of 5,000 m$^2$. It was treated as a singular entity whilst excavations proceeded from 2007–2010. However, during this period, three discrete subsurface fossil beds were recognised representing semi-contemporaneous deposits, but containing different associated skeletons representing three individual sauropod specimens and including unusual ichnological features that indicate a trampled surface (Figs. 6–8).

The trampling is localized to EML011 and is not observed in other northern or southern Plevna Downs sites. EMF102 from EML011(a) and EMF103 from EML011(b) are two associated skeletons recovered 72 m apart, and are divided by an approximately 100 m linear ichnological feature interpreted to be a sauropod 'trample zone'. Silty sediments have been turbated and compressed by the footsteps of numerous heavy tetrapods, likely sauropods walking single file, creating a trodden 'pathway' or 'pad' (*Hocknull et al., 2019*). Partial tracks are discernable, and resemble sauropod footprints, along with clear deformation structures and subsurface sediment deformation. However, complete tracks or trackways are difficult to decipher due to the similarity of the siltstone matrix infilling the depressions made within the trampled sediment. The siltstone has preferentially cemented along the compressed 'pathway' as seen in Fig. 6J. This feature, along with other ichnological features, will be fully described elsewhere.

EMF103 was located within the middle of this linear trampled features and is represented by a series of associated dorsal vertebrae and isolated teeth. The vertebrae are

heavily compressed from trampling, making referral of it to known sauropod taxa difficult, and erection of a new taxon is premature at this stage. It will be described fully in a future study.

**EML011(a) (Fig. 7).** Material; EMF102, Holotype of *Australotitan cooperensis* EML011 (a) was located in 2005 as a small surface scatter of bone fragments that were able to be joined with unweathered fits indicating that this locality was likely to preserve in situ fossil remains that were better preserved in comparison to the heavily weathered remains 1 km to the south at EML010. The total area of EML011(a) is approximately 480 m$^2$.

Excavations produced several massive sauropod appendicular elements including a partial left scapula, partial left and complete right humeri, a complete right ulna, partial left and near complete right femora, both pubes and ischia and indeterminate corticocancellous bone that was originally suspected to be of osteoderm origin. In total, ten elements were recovered in association with the pelvic elements in semi-articulation. No duplicate bones were found and each element corresponds to a sauropod individual of comparable size. Therefore, these elements are treated as the same individual and thus can represent a describable holotype specimen (EMF102) and new taxon, *Australotitan cooperensis*.

The upward-facing surface of each bone has experienced a greater degree of cortical bone weathering than the downward-facing bone surfaces due to the actions of the vertosol soil-forming processes active at the site. The bone surfaces are split into a mosaic of pieces, superficially resembling the mosaic weathering stages of exposed bone (*Behrensmeyer, 1978*; *Lyman, 1994*).

Instead of cracking occurring prior to fossilisation, the surface splitting of the cortical bone observed on these specimens occurred after fossilisation and during the period of weathering at the vertosol-Winton Formation transitional zone. The cracking vertosol penetrated the cemented mudstone matrix encasing the surface bone. Expansion and contraction of theses clays split the cemented matrix into quadrangular sections. The surface cortical bone is indurated with the matrix above it which indicates that when these cracks penetrated the cemented matrix, they also cracked the surface bone, lifting these sections off of the main body of the specimen. The weaker corticocancellous bone layer is a region of weakness and splits before the matrix-cortical bone interface does.

Subsequent infilling of these cracks with vertosol sediment widens the cracks and eventually lifts the cemented matrix with surface bone off of the main body, exposing cancellous bone from inside. As the matrix lifts, sediment penetrates below the surface bone and forms a soft clay infill. Subsequent gypsum precipitation within this clay infill creates a crystalline surface between the lifted matrix-surface bone and the underlying corticocancellous bone. Preparation of the matrix removes the cemented matrix from the thin adhering surface bone, and removal of the gypsiferous layer allows the original cortical bone surface to be repositioned back onto a cleaned surface. These quadrangular pieces present themselves as a mosaic-like pattern across the surface of the bone in a similar way to sauropod remains reported from Argentina (*González Riga & Astini, 2007*).

Most of the bones show post-burial to pre-induration distortion created by localised directional compression forces exerted from above the bone and specifically focused above

the area of distortion. These distortions do not occur uniformly across all of the bones or across the entire surface of a single bone. Therefore, the distortion is not a result of diagenetic and lithostatic compression. Instead, the bones are crushed in localised areas and this direction of crushing is from above and locally generated by forces orthogonal to the in situ horizontal orientation of the bones (Figs. 7 & 8). The best interpretation of these distortions is as a result of crushing through dinoturbation, which involves the actions of trampling by dinosaurs, likely sauropods (Britt et al., 2009). Clear evidence of this crushing has been observed in the right femur, which preserves a well-delineated sauropod manus-shaped crush mark within the proximal diaphyseal shaft (Figs. 8A–8H).

The forelimb elements (scapular blade, humeri and ulna) were all found together with each element touching one of the other elements. Their long axes were oriented in a NW-SE direction for the humeri and ulna and in a N-S direction for the scapular blade. The hind limb elements (puboischial complex and right femur) were found close to one another, whilst the left proximal femoral head was found disassociated from this group, at the surface and downslope from the right femur's position. Between the two appendicular bone groups, a small patch of indeterminate corticocancellous bone was recovered, likely the internal corticocancellous remains derived from within the femur nearby.

The orientation of the in situ bones shows a degree of skeletal sorting by water flow with the long axis of the bones oriented horizontally in either a NW-SE, or a near-normal to this (N/NE-S/SW), direction. The right femur was oriented with a NE-SW long axis direction whilst the pelvis was oriented in a NW-SE long axis direction.

Due to the flat aspect of these broad bone elements, they are oriented either with their long axis in the direction of flow or perpendicular to it, indicating the direction of water flow was the key driver of their final orientations (Kreutzer, 1988; Lyman, 1994; Voorhies, 1969). Based on the dominant direction of orientation, the palaeocurrent was in a NW-SE direction.

Much of the fine primary sedimentary structure has been destroyed by the cementation and concretion formed around the bones, along with significant post diagenetic growth of gypsum throughout the sediment. The bones are preserved in a fine siltstone-mudstone matrix which is cemented, predominantly on the undersides of the bones. There is very little structure to the sediment surrounding the bones other than gross horizontal laminations. These laminations have been compressed in parts, likely through dinoturbation (Fig. 7D).

Below the bonebed, a very thin lens (<10 cm) of cross-laminated yellow-orange coloured sandstone occurs with a scoured top surface that is filled with the overlying siltstone that preserves the bones. This layer was most evident underneath the preserved pelvic elements but was also observed below the ulna and scapula (Figs. 7D–7F).

The cross-laminations indicate a palaeocurrent parallel to the long axis of the pelvic elements (NW-SE) suggesting higher energy flow which was followed by a scouring event with the subsequent deposition of silts along with sparse plant remains and bones. Settling of finer muds produced the gross horizontally laminated siltstone-mudstone matrix which entrained the bones. Following deposition of this thick silty-mud unit with
the entrained bones, the water-saturated soft bones were deformed via trampling (dinoturbation) of the sediment. This, along with post-depositional processes, destroyed much of the primary sedimentary structures available.

Small-sized pieces of woody plant debris covered the top surface of the bones, having settled out with and onto the exposed bone surfaces prior to burial. The largest pieces of wood debris have a preferred long axis orientation of a NW-SE direction, therefore, supporting the dominant NW-SE palaeocurrent direction.

The woody debris is found in close proximity to the surface bone and was most evident during preparation of the femur and scapula, suggesting that these elements formed an obstacle for water flow allowing woody debris to settle. Both these limb elements are oriented normal to the main axis of flow providing a leading edge that would have slowed flow and provided an opportunity for the woody plant remains to settle out.

**EML013.** Material; EMF105 (femur), EMF165 (humerus), EMF166 (metacarpal).

EML013 was discovered in 2007 and is located 860 m northwest of EML011. A small patch of bones within cemented mudstone was found at the surface including a fragmented anterior caudal vertebra and partial ribs. There was no immediate subsurface connection of this scatter to a bonebed; however, after extensive excavation, a line of bones was discovered at depth and within the Winton Formation. This bonebed lay just below a thick rock unit preserving densely packed woody debris, that was well-sorted with a dominant long-axis orientation, NW-SE.

The rock unit shows sorting of the plant debris from large log-jams with directional orientation, with isolated and broken bones, at the base, overlain by smaller suspended plant pieces in matrix, and densely packed woody fragments in the upper-most section (Figs. 6G & 6H). The entire unit has been cemented within a siltstone-mudstone that sits above the underlying bonebed. Isolated and broken bones were found at the base of this cemented woody debris unit (Fig. 6I). Transitioning below this level into the un-cemented Winton Formation a series of well-preserved sauropod bones was found. Four limb elements were found lying side-by-side, offset to one another in an east-west direction by approximately 20–40 cm. Each bone was similarly oriented in a NW-SE direction, parallel with the observed orientations of the overlying woody debris.

The bones include a partial humerus, femur, metacarpal and yet-to-be prepared large limb element. Each of these elements was differentially cemented but clearly isolated within the uncemented Winton Formation siltstone layer below the main debris level. Stratigraphically below and south of this bonebed a thin fine mudstone lens ranging from 5–15 cm in thickness preserved leaf and cone scale impressions. The floral remains exclusively preserve leaves and cone scales from gymnosperms, and pinnae and pinnules of pteridophytes and a possible bennetitalean (Figs. 6A–6F).

Macrofloral fossils occur at all of the southern-central Winton Formation sites associated with the sauropod bonebeds, and are predominantly represented by thick plant debris strands of well-sorted woody remains. Occasional clay lenses exclusively preserve pteridophytes and gymnosperm leafy remains with no indication of equisetaleans, ginkophytes, angiosperms or cycadales macroflora typical of northern Winton Formation sites.

The combination of predominantly thick sections of well-sorted woody remains with rare near-monospecific leaf deposits has not been observed by us from any of the faunal or floral sites in the northern Winton Formation, or the Surat Basin Griman Creek Formation.

The combined depositional, taphonomic and ichnological observations here represent a distinct departure from what would be expected based on observations from the northern Winton Formation sites. The combined bias to sauropod skeletal remains, disturbance by trampling over large areas, and the low diversity of flora, indicates either a unique taphonomic bias that has removed those remains from preservation potential, or it establishes the base for palaeoenvironmental differences observed between northern and southern Winton Formation sites. Palaeoenvironmental differences between the two regions are likely the reasons for these differences and will be discussed later.

## MATERIALS & METHODS

### Fossil preparation

The sauropod remains described herein were prepared using pneumatic air-scribes and pneumatic chisels. All remains were preserved within varying thicknesses of siltstone-cemented matrix that also included layers of gypsum-rich mineral precipitation. Mechanical preparation was used to prepare the holotype using a variety of pneumatic air scribes and an electric high-speed diamond wheel cutter. A combination of air scribes were used, including, a WEN pen, HW50, HW10, No 6 & 4 microjacks and Aro. The preserved elements were partially encased in the concretionary mudstone and buried in the surrounding clays. Gypsum crystals had fractured the surface of some of the preserved elements, and in some areas, a thin iron-oxide crust covered the bone surface.

### Specimen 3-D surface geometry creation

Undertaking comparative assessments of morphology for the key taxa during this work came with specific difficulties because of the specimen's geographical location, physical attributes and conservation considerations. In this particular work, three museum collections house the four holotypes referring to the taxa of specific interest here. *Wintonotitan wattsi* QMF7287 is reposited in the collection of the Queensland Museum, Brisbane, southeast Queensland; *Diamantinasaurus matildae* AODF603 and *Savannasaurus elliottorum* AODF660 are reposited in the collection of the Australian Age of Dinosaurs Museum of Natural History, Winton, central Queensland, and the proposed holotype of the new taxon described here, EMF102, is reposited in the collection of the Eromanga Natural History Museum, Eromanga, southwest Queensland. From Brisbane, each location is around 1,000 km apart, representing a next to impossible logistical means for direct specimen comparisons. Traditional plaster or polyurethane replicas do not exist.

Each type specimen presents its own specific difficulties when undertaking comparative work because of their physical location, very large size and great mass, fragility, and conservation needs. For such large specimens simply viewing individual elements from multiple sides (e.g., proximal, distal, anterior and posterior) can be a fraught process both

for the specimen, the researcher and the collection staff. These difficulties in comparative analysis have been manifest since the discovery of dinosaurs, and since then, concessions have had to be made based on the primary protection and conservation of the type specimens relative to access for assessment by researchers.

Advances in three-dimensional (3-D) scanning technology, in particular, the relatively easily learned and affordable process of photogrammetry (*Bates et al., 2010*; *Falkingham, 2012*; *Otero et al., 2020*), have allowed many of these limitations of comparative work to be resolved by creating three-dimensional models of specimens. Digital 3-D models allow multiple comparisons with multiple specimens in a virtual sense, helping to augment direct observations, and more frequently superseding them.

Since 2011, we (SAH & RAL) have collected photogrammetric data of the four taxa used in this work which has allowed regions of morphological interest to be directly compared between the taxa. During this process, it has become evident that changes and damage sustained to the specimens during events occurring pre- and post-deposition, during preservation, exposure and weathering, during excavation and throughout preparation and display, have all altered the specimens and have influenced comparative capabilities and interpretations.

In the past, many of these taphonomic and preparatory changes to the specimens have been unintentionally or intentionally 'rectified' and 'restored', resulting in what might be considered to be a more realistic representation of the specimen prior to alteration. Thus, providing the researcher with a different morphological starting point for comparisons versus what was originally preserved. Many intentional restorations occur in response to display or by connecting isolated portions of a specimen together to estimate a whole. Restorations of this manner can preclude morphological features or unintentionally fabricate morphology that did not exist in the original element.

Such restorations occurred to the holotype specimen of *W. wattsi* (QMF7292), prior to its establishment as a holotype, which included plaster-based restoration of bones and bolting of elements for display armature. Such restorative work was removed for the purposes of description of *W. wattsi*, although this process also meant the loss of some surface bone. This type of specimen alteration is not uncommon, but it does serve to alter the specimens, sometimes irreversibly from what it was in situ in the field. 3-D digital reconstruction and restoration allow a reversible and testable way of assessing and restoring alterations evident in the specimens so that more meaningful comparative assessments can be made. Demonstrating that a feature does or does not exist, or potentially could, but has been altered from some taphonomic or preparatory reason, impacts all interpretations and needs to be communicated in some way.

3-D digital reconstruction, retrodeformation and restoration is becoming a more common element in palaeontology, whereby a 3-D digital restoration or reconstruction is used to assist in morphological, ichnological, body-size and biomechanical studies (*Otero et al., 2020*). Whilst this process is becoming more commonplace, new standards of reporting are required when utilizing these datasets, especially considering the initial limitations that come with accessing specimens to undertake scanning in the first place.

In particular, digital capture and restoration requires several tradeoffs including capacity of hardware, software and personnel, along with financial and time constraints.

Tradeoffs also include ease of access to capture the specimens in the first instance, which includes lighting, physical location, speed of capture and ultimately resolution and fidelity of the final digital 3-D geometry. This has led to the development of some standards and procedures of capture that may assist collection managers, curators and researchers when deciding about the relative advantages and disadvantages of different scanning procedures and taking into account these tradeoffs (*Bitelli et al., 2020*; *Brecko & Mathys, 2020*; *Lautenschlager, 2016*; *Le Cabec & Toussaint, 2017*; *Otero et al., 2020*; *Vidal & Díez Díaz, 2017*). However, it is unlikely that all standards can be met at all times, and in our present experience, this was the case.

Here we will take the opportunity to describe the methods and processes used as a way to describe the limitations of resulting 3-D models, but also how they provide clear advantages over traditional methods of morphological comparison.

We generated 3-D surface models of the fossil specimens using digital photogrammetry and surface rendering from Computed Tomography (CT) X-ray scans. The process of 3-D model creation using photogrammetry and CT data is well documented across many disciplines and readily available through software manuals, online tutorials, YouTube demonstrations and simple, but iterative, trial and error.

From 2011–2014 specimens in this study were captured using two Panasonic Lumix DMC-TZ30 cameras. These cameras were chosen due to their portability, affordable price, rapid shooting, tough body, and image LED review screen. This allowed them to serve multiple purposes for capturing specimen and field site photogrammetry. Their small compact size with LED review screen allowed us to position and focus on specimens quickly and evenly, and in very awkward and tight positions, such as on darkly-lit shelves, within fiberglass cradles in preparation laboratories, on display, or in very small spaces within cramped working spaces.

The settings were set to 'Fine JPG' resolution, using f-stop settings between F12-18, ISO Auto or 100, under autofocus. Lighting was balanced as best possible during each shooting session; however, individual bones may have been captured over a period of several months or years depending on the point of preparation of each available side of the specimen. The difference in lighting and colour can be seen on a number of specimens where the shooting occurred at different times with different lighting arrangements, creating dissimilar coloured surfaces. This did not affect the geometric reconstruction.

Rapid and close-range images were taken of each specimen with the user moving around the specimen. Foreground and background elements were initially recorded for alignment control, and then later removed from the dense point cloud. We also opted to 'over-shoot' each specimen, focusing on capturing as much fine surface detail as possible.

Due to the massive size and impossibility of building a large enough turn-table, undertaking standard turn-table techniques were not employed. In addition, due to the location of many of the specimens occurring either in a preparation facility or within close range to very dusty environments, it was impossible to control dust and therefore, creating a uniform coloured background was not possible. Instead, we opted to include the

foreground and background elements within the photograms, so that although the main focus of the reconstruction was on the specimen, the shooting included elements that would assist in alignment and would be removed later. We found the more irregular these features, the better the overall alignment. Therefore, in future, if a uniform clean background and stage with turntable is not possible, we suggest creating a very geometrically complex stage and remove unwanted dense point cloud data after this phase of reconstruction.

Although we understood the tradeoff of the number of images taken relative to additional geometry, digital storage space, and processing time, we opted to 'over-shoot' each specimen. This created close to two or three times as many images as was generally required for a usual turn-table approach where all factors such as light, camera stability, camera resolution and processing time are all controllable. We also focused on capturing as much fine surface detail as possible each session within the timeframes available.

Due to the large number of images captured per specimen and long processing time, subset image batches were processed in Agisoft Photoscan Standard versions 0.8.2 (June 2011) to 1.0.4 (April 2014 and then in Agisoft Metashape 1.6.1 build 10009 (20 January 2020)), retrieved from http://agisoft.com. All images of each specimen were reprocessed in Reality Capture software, retrieved from http://capturingreality.com (beta 2014 onwards) due to its faster processing speed of greater numbers of images whilst using the same processing power. This new process returned a greater detail of surface geometry, especially in areas with detailed image clusters.

Each specimen needed to be captured from at least two sides due to their large size, fragility and housing cradle. If possible, a significant overlap of an area was captured from each side so that both could be neatly aligned later. Images were aligned and positions reconstructed in the software, with a dense point cloud generated from these positions. Surface geometry was reconstructed in Reality Capture using Normal Settings with vertex and polygon colouration. All outputs were exported as Stanford Triangle Format (i.e. .ply files).

Removal of unwanted geometry, such as background structures and specimen housing was undertaken in Reality Capture and Agisoft Photoscan at the dense point cloud stage, leaving only the geometry representing the specimen and the included scale bar. If poorly reconstructed geometry was observed, usually below the edges of specimens where there was overhang or shadowing, this geometry was also removed to reduce the production of inaccurate additional geometry when the surface models were aligned to one another.

The scanned components of the specimen were scaled to real-world dimension in Meshlab (Callieri et al., 2012; Cignoni et al., 2008), by measuring the included scale bar or a known distance on the specimen using the measuring tool. The real-world measurement was then divided by the measurement given in Meshlab, thereby providing a scaling factor. This scaling factor was then used to scale the object in Meshlab using the Scaling option, whereby the scaling factor occurred in all directions (x, y and z). The scale of the specimen was then re-checked by measuring within Meshlab the included

scale bar or known length. We then 'Freeze the Current Matrix' so that the new scaling factor is coordinated to the vertex positions. Finally, the model is exported as a .ply file.

Each component of the specimen model is then aligned together in Meshlab (*Cignoni, Rocchini & Scopigno, 1998*; *Pietroni, Tarini & Cignoni, 2009*) using the alignment tool by point picking multiple corresponding positions of overlap on each component and adjusting this alignment for maximum best fit. Ideally, specifically corresponding geometries or specimen numbering written on the specimen are chosen to allow for quick and accurate point picking to occur. The two aligned meshes are more precisely aligned using the default alignment parameters within Meshlab. If alignment is not clear, we cross-check this in Cloud Compare software (*Girardeau-Montaut, 2016*), using the alignment tools of this software. Once aligned, the two separate components (layers), are merged using the 'Flatten Visible Layers' tool and exported, creating a single model.

This combined, merged model is re-meshed using the Poisson Surface Mesh reconstruction tool with the Reconstruction Depth set to 12, and the Adaptive Octree Depth set to 8 (*Cignoni et al., 2008*; *Kazhdan & Hoppe, 2013*). We have found that these meshing parameters produce the most accurate resulting full surface geometry. However, some components may create additional geometry along the seams between two parts that had limited overlap. For example, the large limb elements that are fixed within firm housing fibreglass cradles are missing approximately 5–10 mm of overlap due to the obscuring nature of the cradle. Therefore, alignment needed to take this into account, and the Reconstruction Depth using the Poisson reconstruction method may need to be reduced to 10 or 8. Although this reduces the overall detail in the surface geometry, it also removes the false geometry. A tradeoff is required to attain the best re-meshed model.

Finally, the fully aligned and re-meshed model is colourised by transferring the vertex colour attributes from the original components onto the new uncoloured mesh geometry. We do this using the Vertex Attribute Transfer tool in Meshlab (*Cignoni et al., 1999*). The finalized, coloured model is then exported as a .ply model. We once again take measurements from the included scale bar or known distances to verify correct scaling. We then remove the scale bar from the model and undertake a final model clean using the 'Remove Isolated Pieces' tool in Meshlab (*Cignoni et al., 2008*). We then re-align the model to the correct bounding box position and use the manipulator tool to reorient the model so that the dorsal anatomical direction is aligned to the z-axis within the 3-D model space, and the anteroposterior anatomical direction is in the x-axis plane. The final model is exported again as a .ply file.

In addition to photogrammetry data, where possible we collected CT scan data for the holotype of *W. wattsi* and particular remains associated with EMF102. The ischium of *W. wattsi* was digitized using CT scan data that was aligned and processed in Dragonfly 3.6 (Computer software), from Object Research Systems (ORS) Inc., Montreal, Canada, 2018 retrieved from http://www.theobjects.com/dragonfly.

The ischium was too large to be scanned as one piece, so we scanned the specimen twice, moving it across the gantry to allow all of it to be captured. These two scan datasets were then aligned in Dragonfly 3.6 using the image stack alignment tool. A surface model was then generated from these aligned CT scan datasets.

## Specimen 3-D digital restoration, retrodeformation, reconstruction and annotation

A benefit of 3-D digital geometry of specimens in palaeontology is the capacity to manipulate these specimens in a way not possible with the original specimen. In addition, digital techniques can help restore bones to reflect the known and predicted original shape (*Lautenschlager, 2016*; *Vidal & Díez Díaz, 2017*). In particular, skeletal remains when components of the right and left elements are preserved, but are not complete, can be used together to restore a whole single bone. Here we undertook similar processes to assist in reconstructing the bones we compared.

Before restoration or reconstruction can be accomplished the specimens need to be assessed for matrix obscuration, bone damage and loss, along with deformation. High fidelity models that possess realistic and detailed colour allow the user to see features and textures with the geometry that colourless surface scans cannot, which is a distinct advantage of photogrammetry. Specimens that are digitized in pieces provide an extra level of data if each individual piece is reconstructed, because they can provide cross-sectional information such as cortical and cancellous bone thickness that a completed bone may not reveal.

Computed Tomographic (CT) scans provide another level of detail that can show difficult to distinguish matrix coverage or bone damage, surface corrosion and loss. Together, using these different lines of evidence, each bone can be restored. However, prior to any restoration, the obscured, altered, missing or damaged areas need to be clearly identified on the 3-D model geometry.

To do this, we colourised a duplicate 3-D model of each specimen and digitally painted onto the surface geometry areas of alteration, damage and deformation using a pre-defined colour scheme (Fig. 8O). Meshlab (*Cignoni et al., 2008*) was used to undertake this surface geometry painting, including singular colour choices without gradation or feathering, with the brush set to 100% opacity and 100% hardness. This provided a clear distinction between a painted surface and the colour data from the original surface scan, thereby indicating clearly what has been intentionally coloured and what has not.

The colour scheme used the following preferences using the Meshlab (*Callieri et al., 2012*; *Cignoni et al., 2008*; *Cignoni et al., 1999*) standard HTML HEX colour coding: Brown (#aa5500) indicating obscuring matrix; Purple (#aa55ff) indicating bone deformation; Red (#ff0000) indicating significantly broken/missing surfaces; Magenta (#ff55ff) indicating corroded surfaces; Dark Green (#55aa00) indicating loss of cortical bone surface; Very light orange (#ffaa7f) indicating mosaic broken surface (cortical bone); White (#ffffff) indicating plaster fill; Yellow (#ffff00) indicating poorly rendered 3-D model geometry (Fig. 8P); Light Blue (#55aaff) indicating pneumatic pores and cavities. All images rendered from these models for the figures used herein were produced in Meshlab using natural vertex colour, ambient occlusion, x-ray or radiance scaling rendering (*Cignoni et al., 2008*; *Vergne et al., 2010*), or by using the edge detect feature in Dragonfly 3.6 with the 3-D model placed in orthogonal projection and 100% transparent.

After completion of the 3-D specimen model, the regions of deformation and alteration were identified and segmented into separate components using the model cutting tool in Agisoft Metashape. The lasso cutting tool was used to trace the line of deformation, which then broke the model into at least two components. If this region was deformed further, additional segments were created. Each segmented piece was saved as a separate model to be re-aligned in Meshlab. After identifying the greatest degree of deformation, usually in the downward direction relative to the field site position, the segmented components were rotated in the x- or y-axis to align to the un-deformed portion of the model. Once the new alignment was determined, all of the components were merged using the 'Flatten Visible Layer' tool in Meshlab. The resulting merged model was then re-meshed using the same process described above and the resulting closed mesh exported as a new model.

Bone retrodeformation was undertaken by SAH where such deformation would clearly influence comparative understanding. The focus of this procedure was to retrodeform the surface scan models of EMF102 elements so that they could be compared to other taxa without the influence of distortions leading to misinterpretation of similarities or differences between taxa (Fig. 8). If the bone was undeformed, or the deformation features did not alter the overall shape of the element substantially, or a better preserved contralateral pair existed, comparative assessments were undertaken directly between these elements as preserved. These regions included the scapula (excluding the acromion plate), humerus, (excluding the deltopectoral crest), ulna (excluding the diaphyseal curvature), pubes and ischia (excluding the right ischium) and femur (excluding the proximal half of the diaphysis).

Retrodeformation was applied to the humerus to restore the deformed deltopectoral crest of the left humerus. The deltopectoral crest was deformed during removal at the point of excavation where the crest relaxed outward from its original position due to the compressive weight of the specimen and lack of reinforcement of the plaster jacket. The preserved extent of the right humerus (digitally mirrored) provided a guide to the direction of the distal end of the deltopectoral crest for the left humerus. Field images prior to removal provided additional guidance as to the shape of the overall element. Finally, each segment could not overlap each other, which provided the key limitation to the overall shape of the crest and the proximal margin.

The right ulna diaphysis was clearly bent downwards in the site, through the processes of trampling. The diaphysis was segmented into components and realigned so that the shaft was straightened. The pubes and ischia were segmented apart due to each element being slightly dislocated from their articular margins. They were then relocated, re-articulated along their articular margins. It was evident that the right pubis and ischium had suffered most deformation and crushing so the left puboischium (and its duplicate mirror) was used as the base model for the reconstruction of the pelvic floor and comparisons of this element.

The right femur was deformed downwards in the site having also been crushed from trampling. The proximal half of the shaft was segmented into components and realigned so that the shaft was straightened. The distal end was not deformed but some areas of the

condyles had been lost post-deposition. To restore the proximal region of the femur, the isolated and associated left femoral head of EMF102 along with a referred proximal femoral head (EMF164) were used to reconstruct an entire femur. We subsequently used the referred complete femur (EMF105) to compare our resulting reconstruction.

With the elements of EMF102 retrodeformed and/or reconstructed using specimens referable to the new taxon VK undertook to digitally sculpt complete bones using these retrodeformed elements as the basis for the models. VK used ZBrush digital sculpting software retrieved from https://pixologic.com/ to generate a new geometry for each element, using the retrodeformed models as a subtool basis for this new geometry. Also at this stage, any additional small deformations, weathering features or cracked surfaces were digitally 'repaired'. The overall geometric shape and size were not altered. Where areas of articulation were missing articular surfaces, these were estimated based on the preserved trajectory of such features in the reconstructed models or by reference to better-preserved titanosaurians from the literature. To be clear, these sculpted ZBrush models were not used in any comparative assessments between taxa, or for the establishment of the diagnostic characteristics of the taxon. They serve only as a guide to the overall shape and size of the reconstructed bones, allowing us to produce 3-D printed 1:1 scale versions of them and to assist in recreating a skeleton for exhibition.

## PHYLOGENETIC ASSESSMENT

A preliminary phylogenetic assessment is undertaken using recently published datasets, including those that included the three previously described Australian taxa (*Poropat et al., 2021*; *Royo-Torres et al., 2020*). The phylogenetic dataset (*Mannion et al., 2019b*) used in both of these recent analyses included the three Australian taxa of interest here, each adding new taxa and characters.

Using these two recent assessments (*Poropat et al., 2021*; *Royo-Torres et al., 2020*), we score the character states for *Australotitan cooperensis*, along with revised character state scores for the Australian taxa, where needed. In particular, we observed a number of characteristics of *Wintonotitan wattsi* that are poorly preserved or not preserved, making some of the previously scored states equivocal, in our opinion. We could, however, make direct comparisons and estimates of these states using 3-D cybertypes created from the holotypes of each Australian taxon. All character state score changes are provided as Supplementary Information and we indicate where scores have changed and whether estimated scores are used. The updated datasets were entered into MESQUITE 3.61 software (*Maddison & Maddison, 2019*) and analysed using TNT 1. 5 software (*Goloboff, Farris & Nixon, 2008*) (Supplementary Information).

In both assessments we ran a series of computations using the same protocols and parameters as previously set out (*Mannion et al., 2019b*; *Poropat et al., 2021*; *Royo-Torres et al., 2020*). These included a priori exclusion of fragmentary and unstable taxa, although we note that at least two, perhaps three, of the Australian taxa would fit within this similar protocol based on the level of preservation and unstable nature of their phylogenetic position. However, for the purposes of this preliminary assessment of Australian taxa, we did not exclude them. The excluded taxa were *Astrophocaudia, Australodocus,*

*Brontomerus, Fukuititan, Fusuisaurus, Liubangosaurus, Malarguesaurus*, the Cloverly titanosauriform, and *Ruyangosaurus*. Multi-state characters that were previously ordered, were retained as ordered. No new characters were added, numbering 542 (*Royo-Torres et al., 2020*) and 552 (*Poropat et al., 2021*) characters. For all assessments the maximum number of trees saved was set to 99,999 (TNT 1.5, Windows no taxon limit).

The first assessment included all of the remaining taxa from each previous study, assigning *Shunosaurus* as the outgroup taxon. First, a New Technology Search was undertaken using sectorial, drift and tree fusing with the stabilize consensus set to 5 times. Weighting for all characters was equal. The resulting most parsimonious trees were then saved and subjected to a strict consensus to produce a single tree that we then used in the discussion. Bootstrapping and Bremer support were trialed on all analyses, however, all results returned very poor results with Bremer support of <1 and bootstrap support of <50 for all clades. However, the resulting strict consensus trees provided some areas for discussion.

We undertook a second assessment using identical parameters to the first, except we changed the character weightings. Following previously developed protocols for weighting characters in sauropod phylogenetic analyses we increased the implied weighting k value to 9.0 (*Tschopp & Upchurch, 2019*). As with the unweighted analysis, we saved all most parsimonious trees and subjected them to a strict consensus. Again, Bremer support and Bootstrapping were unsuccessful in returning useful supporting statistics.

We then undertook a 'Traditional Search' using the Tree Bisection-Reconnection (TBR) algorithm, a method traditionally used in maximum likelihood phylogenetic analyses (*Swofford, 2003*). We set the number of replicates to 1,000 and number of trees saved per replicate to 100, totaling a possible 100,000 maximum trees to be retained. Both analyses used equal weighting for characters.

To assess whether the non-Macronarian taxa were potentially influencing the tree topology, we excluded all taxa, retaining only those considered to be within Macronaria (*Mannion et al., 2019b*), and placed *Camarasaurus* as the outgroup taxon. With such a large reduction in taxa, we opted to use the TBR 'Traditional Search' with 1,000 replicates and 100 trees saved per replicate. We weighted the characters using a k value of 9.0.

Next, to assess the possible influence of a lack of non-appendicular characters in *A. cooperensis* on its phylogenetic position, we excluded all non-appendicular characters from the assessment. Differing from all of the previous assessments, we did not exclude any taxon, including the fragmentary or unstable taxa, because many of these are known from appendicular elements and thus could be useful in comparisons. *Shunosaurus* was selected as the outgroup and we undertook a TBR 'Traditional Search' with 1,000 replicates and 100 trees saved per replicate. We weighted the characters using a k value of 9.0.

Finally, we undertook to exclude unstable taxa and taxa aged younger than the Turonian, whilst retaining all characters. We chose to do this as an exploration of the data by excluding taxa that are temporally unrelated to our target group. By excluding younger taxa, we expected this would reduce potential descendant homoplasy. We ran two analyses for each dataset, using a TBR 'Traditional Search' with 1,000 replicates, saving 100 trees per

replication, undertaking both equal character weighting and implied character weighting with a k value of 9.0.

## Body-size estimation

Body mass estimation is a fraught exercise for fragmentary skeletons (*Bates et al., 2015*; *Bates et al., 2009*; *Bates et al., 2016*; *Campione & Evans, 2012*; *Campione & Evans, 2020*; *Paul, 2019*). Recent body mass estimates of giant sauropods (*Carballido et al., 2017*; *Lacovara et al., 2014*) using humeral and femoral circumferences (*Benson et al., 2014*; *Campione & Evans, 2012*; *Campione & Evans, 2020*) have come under scrutiny and are shown to be implausible or inaccurate (*Bates et al., 2015*; *Otero, Carballido & Moreno, 2020*; *Paul, 2019*). However, a recent review of these inaccuracies has suggested that the estimation methods themselves can be reconciled, albeit with reservations when dealing with particular groups of tetrapods, like giant sauropods (*Campione & Evans, 2020*). Therefore, although it is tempting to produce an estimate of body mass for *A. cooperensis* based on the preserved and reconstructed stylopodial circumferences we consider that this will not add significant interpretative value to our main purpose of describing this taxon, and comparing it to other members of the Titanosauria from the Winton Formation and semi-contemporaneous faunas.

Based on limb-size, a feature that is easily comparable, we can compare *A. cooperensis* to other sauropods of similar size globally. We used the limb element sizes provided in (*Benson et al., 2014*) for our comparisons to *A. cooperensis*.

Humerus and femur lengths, along with humerus and femur circumferences from known taxa were plotted against the type specimen of *A. cooperensis* (EMF102) and a reconstructed femur (EMF164) to see where this new Australian taxon falls in regards to the largest sauropods known from femora and humeri (Supporting Information).

## New taxonomic name

The electronic version of this article in Portable Document Format (PDF) will represent a published work according to the International Commission on Zoological Nomenclature (ICZN), and hence the new names contained in the electronic version are effectively published under that Code from the electronic edition alone. This published work and the nomenclatural acts it contains have been registered in ZooBank, the online registration system for the ICZN. The ZooBank LSIDs (Life Science Identifiers) can be resolved and the associated information viewed through any standard web browser by appending the LSID to the prefix http://zoobank.org/. The LSID for this publication is urn:lsid:zoobank.org:pub:AF1FA65A-5351-45B1-B0CB-EC1225590A0F. The online version of this work is archived and available from the following digital repositories: PeerJ, PubMed Central and CLOCKSS.

## RESULTS

**Systematic Palaeontology**
**Dinosauria** Owen, 1842
**Saurischia** Seeley, 1887

**Sauropodomorpha** von Huene, 1932
**Sauropoda** Marsh, 1878
**Eusauropoda** Upchurch, 1995
**Neosauropoda** Bonaparte, 1986
**Macronaria** Wilson and Sereno, 1998
**Titanosauriformes** Salgado et al., 1997a
**Somphospondyli** Wilson and Sereno, 1998
**Titanosauria** Bonaparte and Coria, 1993

*Australotitan* gen. nov.
Type Species. *Australotitan cooperensis* gen. et sp. nov.

**Diagnosis.** As for species.
*Australotitan cooperensis* gen. et sp. nov.

**Material.** Holotype: EMF102, consists of ten appendicular elements and pieces of corticocancellous internal bone. The appendicular elements include a partial left scapula, partial left and complete right humerus, right ulna, right and left pubes and ischia, and partial right and left femora.
Referred Specimens: EMF164, a fragmented femur, a fragmented ulna, presacral vertebral centrum fragments and rib fragments. EMF105, a complete femur and EMF165, a distal humerus.

**Age & Horizon**. Cenomanian-? Turonian, Winton Formation.

**Type Locality.** EML011(a). Referred Specimen Localities, EML010 & EML013.

**Etymology.** *Australo*–meaning southern in Greek and in reference to the southern continent of Australia; *titan*–from the Greek mythological Titan gods and in reference to its gigantic size; *cooperensis*–being from the Cooper-Eromanga Basin, Cooper Creek system & "Cooper Country".

**Diagnosis**
A large titanosaurian sauropod with the following combination of characters that differentiate this new taxon from all others. Proposed autapomorphies indicated by an asterisk. Scapular blade, narrow and straight with sub-parallel dorsal and ventral margins with lateral ridge situated near the ventral margin. Humerus with a rounded ridge that extends from the distal end of the deltopectoral crest to just proximal of a tri-lobate distal epiphysis. Ulna with heavily reduced anterolateral and olecranon processes relative to much enlarged and elongate anteromedial process. Ulna with a distinct radial interosseous ridge within the distal half of the radial fossa*. Anterolateral process of the ulna with a distal accessory projection* proximal to a proximally beveled distal epiphysis*. Pubes and ischia broad and contact each other medially forming a cohesive pelvic floor. Distal ischial blades curve ventrally to produce a dorsal face that is posteriorly directed. Femur with a medially sloped proximolateral margin, diaphysis narrow anteroposteriorly, and distal condyles directed anterolaterally to posteromedially.

## Description

**Holotype, EMF102.** *Scapula* (**Figs. 9** & **10**; **Table 2**). The scapula will be described with the long axis of the blade held horizontal and the short axis of the blade held vertically (dorsoventrally) with the acromion process vertical (dorsally oriented). A partial left scapula is represented in the holotype preserving from the mid-section of the anterior supracoracoideus fossa, including the acromion ridge and process, to a large proximal portion of the scapular blade. The anterior portion of scapular plate that articulates with the coracoid, including the proximal portion of the supracoracoideus fossa, coracoid suture (articulation), glenoid fossa and proximal portion of the supraglenoid buttress is not preserved having been broken off before fossilisation. It is missing the distal portion of the scapular blade including the distal-most margin. The proximoventral margin of the scapular blade base has been crushed and pushed dorsomedially into the medial side of the scapular blade.

The surface cortical bone of the scapular plate and blade is broken into a mosaic-like fracture pattern with minor distortions due to collapse and some crushing from trampling; however, the overall morphology is intact.

The preserved section of the scapular plate proximal of the acromion ridge is very thin in mediolateral thickness and is deflected medially. This makes what would have been the anterior fossa very shallow and angled medially, thus the coracoid articulation was also most-likely medially positioned and coracoid angled medially. The bone is very thin along the exposed (broken) margins of the proximal and proximoventral regions of the scapular plate, indicating that these missing regions making up the supracoracoideus fossa, coracoid suture (articulation) and glenoid were gracile.

The proximal dorsoventral expansion of the acromion region is hard to estimate; however, the thickness of the bone at the preserved proximal margin suggests that it wasn't expanded to a level seen in similarly large and gracile scapulae like that of *Dreadnoughtus schrani* (see Fig. 2 in (*Ullmann & Lacovara, 2016*)). Instead, it is most similar to the scapula of *Yongjinglong datangi* (see Fig. 11 in (*Li et al., 2014*)).

*Lateral View.* The acromion is not fully preserved, with the ventral margin missing, therefore, the relative acromion dorsoventral height to minimum dorsoventral height of the scapular blade is not precisely known. However, based on the preserved extremities, the proximal region of the acromion at its broadest part was not significantly expanded dorsoventrally. Based on our reconstruction, the ratio of minimum scapular blade dorsoventral height to acromial plate dorsoventral height would be 0.48 for *A. cooperensis* (Table 2). *Y. datangi* (see Fig. 11 in (*Li et al., 2014*)) approaches this with a ratio of 0.5 derived from a minimum scapular blade dorsoventral height of 230 mm and an acromial plate dorsoventral height of 460 mm. Comparing this ratio across other titanosauriform sauropods, there is variation from 0.29 to 0.5 (e.g. *Muyelensaurus pecheni*: 0.29 (*Calvo, González-Riga & Porfiri, 2007a*); *Elaltitan lilloi*: 0.30 (*Mannion & Otero, 2012*); *Dr. schrani*: 0.34 (*Ullmann & Lacovara, 2016*); *Patagotitan mayorum*: 0.38 (*Carballido et al., 2017*); *Saltasaurus loricatus*: 0.4 (*González Riga et al., 2019*); *W. wattsi*: 0.42 (*Hocknull et al., 2009*; *Poropat et al., 2015a*); *Jiangshanosaurus lixianensis*: 0.42 (*Mannion et al., 2019a*);

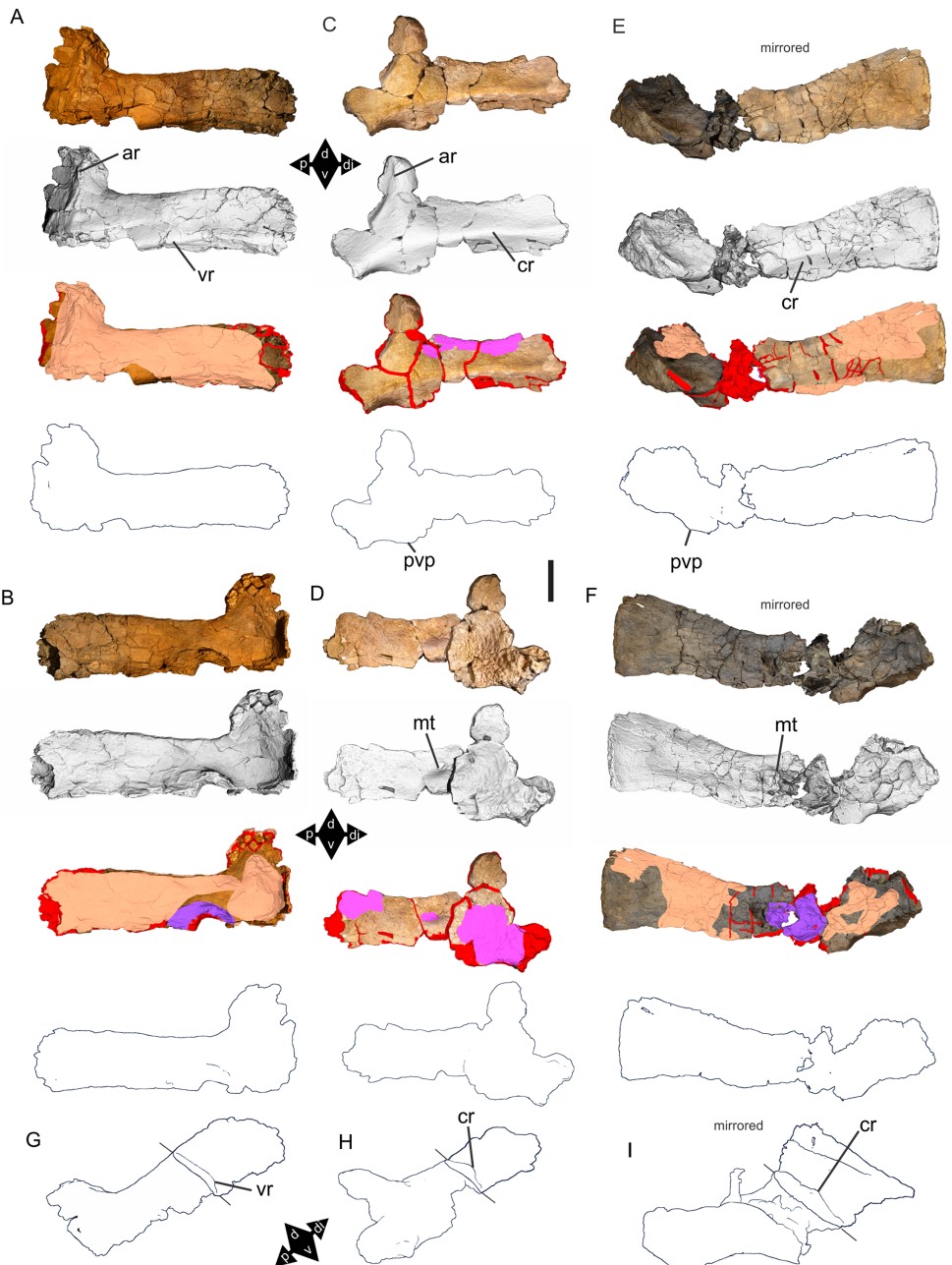

**Figure 9 Scapulae of *Australotitan cooperensis* gen. et sp. nov. (EMF102), *Wintonotitan wattsi* (QMF7292) and *Diamantinasaurus matildae* (AODF603).** Each element is rendered using four methods from top to bottom, natural; ambient occlusion with radiance scaling; coloured schematic (see Fig. 8); and orthogonal outline edge detection. (A & B) 3-D model of *A. cooperensis* gen. et sp. nov. left scapula in lateral (A) medial (B) views. (C & D) 3-D model of *W. wattsi* left scapula in lateral (C) and medial (D) views. (E & F) 3-D model of *D. matildae* right scapula (mirrored) in lateral (E) and medial (F) views. (G–I) Proximoventral views showing mid scapular blade cross-sectional profile in *A. cooperensis* gen. et sp. nov. (G), *W. wattsi* (H) and mirrored in *D. matildae* (I). Arrows indicate direction (d, dorsal; di, distal; p, proximal; v, ventral). Feature abbreviations: cr, central ridge of scapular blade; mt, medial tuberosity; pvp, proximoventral process; vr, ventral ridge of scapular blade. Scale bar = 20 cm.

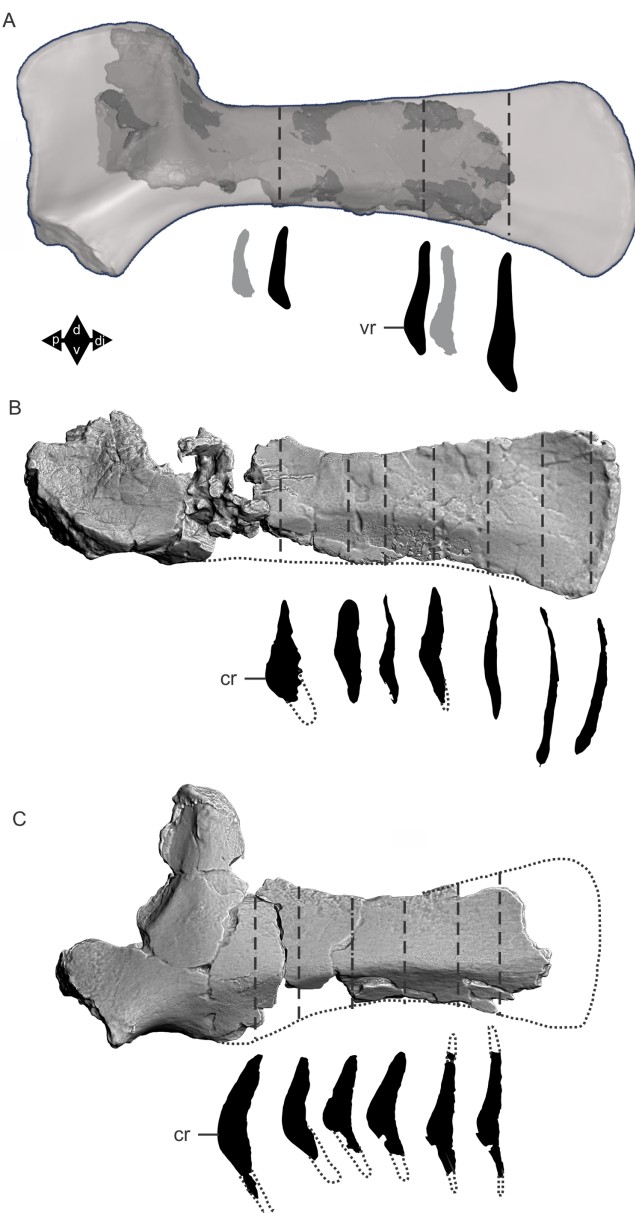

**Figure 10 Scapulae of *Australotitan cooperensis* gen. et sp. nov. (EMF102), *Diamantinasaurus matildae* (AODF603) and *Wintonotitan wattsi* (QMF7292) and showing relative cross-sectional profile across the scapular blade.** (A) *A. cooperensis* gen. et sp. nov. preserved scapula aligned within the reconstructed scapula. Aligned models rendered using transparency tool and orthogonal outline edge detection. (B) *D. matildae* (mirrored right 3-D model). (C) *W. wattsi*. Dashed vertical lines indicate position of cross-section. Dotted lines indicate estimation of missing scapular blade. All three scapulae are isometrically scaled to the minimum scapular blade dorsoventral height. Arrows indicate direction (d, dorsal; di, distal; p, proximal; v, ventral). Feature abbreviations as per Fig. 9.

*Suuwassea emilieae*: 0.43 (*Harris, 2007*); *Vouivria daparisensis*: 0.45 (*Mannion, Allain & Moine, 2017*)), and *Y. datangi*: 0.5.

The dorsal process of the acromion is short, straight and oriented perpendicular to the long axis of the scapular blade. The acromion ridge is nearly straight along its dorsoventral

**Table 2 Scapula measurements of Winton Formation sauropods.**

| | Preserved | Reconstructed |
|---|---|---|
| **EMF102 *Australotitan cooperensis*** | | |
| Maximum proximodistal length | 1220.5+ | 2182.98 |
| Maximum acromial plate dorsoventral height (c) | 498.94+ | |
| Minimum scapular blade dorsoventral height (a) | 264.89 (at base) | 264.89 |
| Maximum scapular blade dorsoventral height | 313.36+ | |
| Maximum proximodistal scapular blade length | 911.12+ | |
| Maximum mediolateral scapular blade thickness (b) | 65.86 | 65.86 |
| (b)/(a)–relative thickness of blade | 0.25 | 0.25 |
| (a)/(c)–relative acromion plate to minimum scapular blade height | | 0.48 |
| **AODF603 *Diamantinasaurus matildae*** | | |
| Maximum proximodistal length (d) | 1485.48 | |
| Maximum acromial plate dorsoventral height | 354.46+ | |
| Minimum scapular blade dorsoventral height (a) | 283.15 (mid-blade) | |
| Maximum scapular blade dorsoventral height | 407.37+ (distal expansion) | |
| Maximum proximodistal scapular blade length (c) | 876.94 | |
| Maximum mediolateral scapular blade thickness (b) | 59.13 | |
| (b)/(a)–relative thickness of blade | 0.21 | |
| **QMF7292 *Wintonotitan wattsi*** | | |
| Maximum proximodistal length | 1088.48+ | |
| Maximum acromial plate dorsoventral height (c) | 563.14 | |
| Minimum scapular blade dorsoventral height (a) | 235.34 (at mid-blade) | |
| Maximum scapular blade dorsoventral height | 287.53+ (distal expansion) | |
| Maximum proximodistal scapular blade length | 652.19+ | |
| Maximum mediolateral scapular blade thickness (b) | 77.42 | |
| (b)/(a)–relative thickness of blade | 0.33 | |
| (a)/(c)–relative acromion plate to minimum scapular blade height | 0.42 | |

**Notes:**
All measurements in mm.
+, full length not preserved.

length expressed as a low and rounded lateral face. The ventral-most portion of the acromion ridge is missing; however, what is preserved is a broad low rise that becomes slightly steeper along its dorsal length where it terminates at the dorsal-most region comprised of roughened surface bone texture. This may be interpreted as a tuberosity; however, we cannot exclude taphonomic alteration of the dorsal margin. The posterior surface of the acromion process is a flat plate running from the acromion ridge to the scapular blade base. There is no posterior acromion fossa or notch present.

The posteroventral corner of the acromion is not preserved in the holotype so it is not possible to determine whether it possessed a subtriangular posteroventral process, similar to that seen in *D. matildae* (Figs. 9E, 9F, 9I and 10B; see also Fig. 4A in (*Hocknull et al., 2009*) and Fig. 8B in (*Poropat et al., 2015b*)), and *W. wattsi* (Figs. 9C, 9D, 9H and 10C; see also Figs. 16G–16H in (*Hocknull et al., 2009*) and Fig. 7B in (*Poropat et al., 2015a*)).

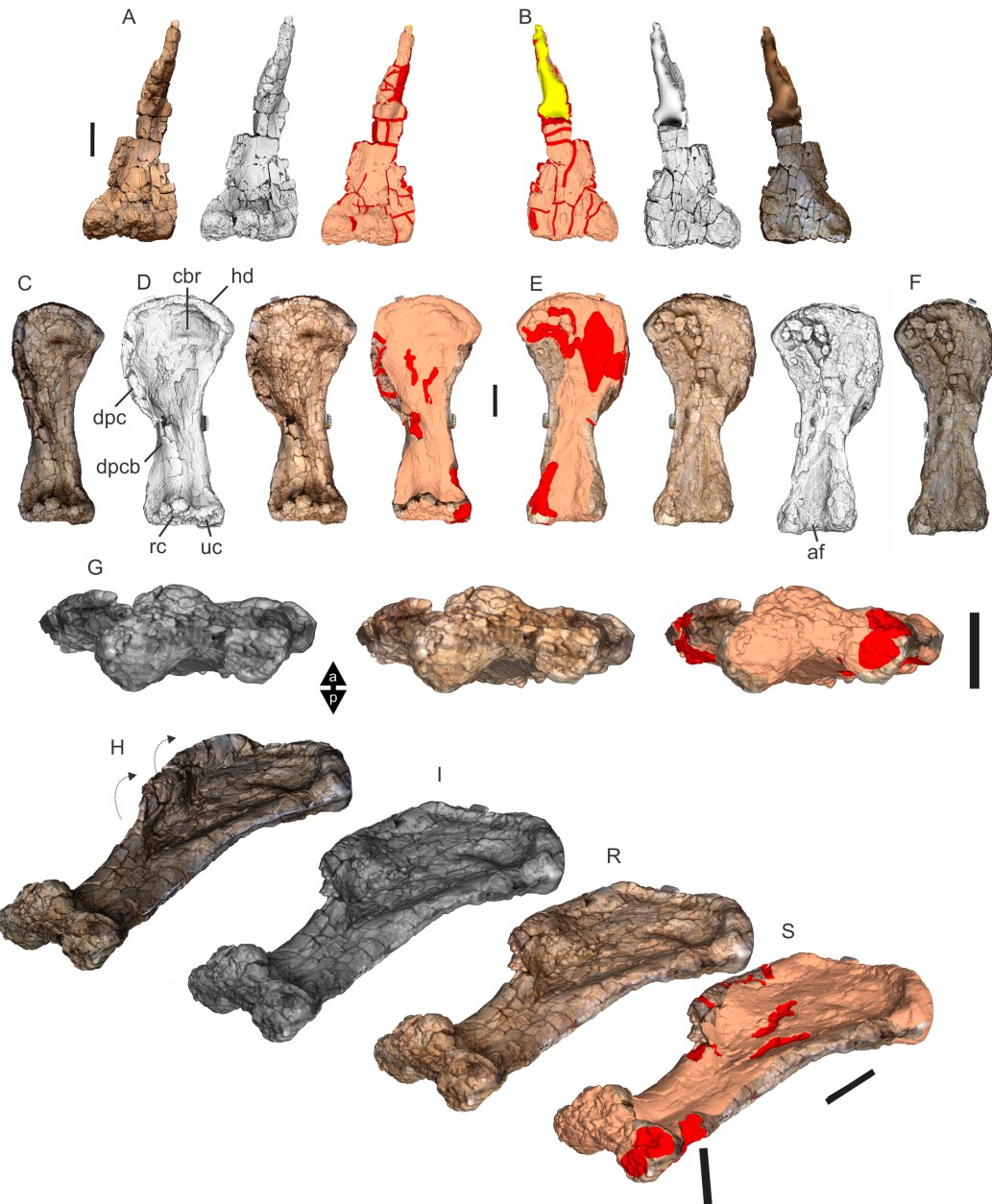

**Figure 11 Humeri of *Australotitan cooperensis* gen. et sp. nov. (EMF102).** (A & B) Left partial humerus in anterior (A) and posterior (B) views. (C–S). Right humerus in anterior (C & D), posterior (E & F), distal (G) and oblique anterodistal (H–S) views. C, F and H are retrodeformed. 3-D image rendering methods used included, natural, A (left), B (right), C, D (middle), E (middle), F, G (middle), H & R; ambient occlusion with radiance scaling, A (middle), B (middle), D (left), E (right), G (left) & I and coloured schematic (see Fig. 8), A (right), B (left), D (right), E (left), G (right) & S. Arrows indicate direction (a, anterior; p, posterior). Feature abbreviations: af, anconeal fossa; cbr, coracobrachialis scar; dpc, deltopectoral crest; dpcb, deltopectoral crest base; hd, humeral head; rc, radial condyle; uc, ulnar condyle. Scale bars = 20 cm.

The scapular blade is dorsoventrally narrowest just distal of the scapular blade base where it meets the acromion plate; in comparison with *W. wattsi* and *D. matildae* where the narrowest point is further distally along the blade. The entire scapular blade is narrow along its entire length with sub-parallel dorsal and ventral blade margins with only a slight expansion of the preserved distal portion of the blade. The distal-most end is not preserved and there is no indication of significant expansion relative to the main blade plate; therefore, it is likely that there is a significant portion of the distal blade missing (Fig. 10A).

On comparison with sauropods possessing mediolaterally thin scapulae with parallel dorsal and ventral margins such as *Y. datangi* (Li et al., 2014) and *Lirainosaurus astibiae* (Díaz, Suberbiola & Sanz, 2013) the scapular blade could conceivably be much longer than is preserved. *D. matildae* (Hocknull et al., 2009; Poropat et al., 2015b) and *W. wattsi* (Hocknull et al., 2009; Poropat et al., 2015a) have shorter, robust, and distally expanded scapular blades by comparison.

A ventral ridge runs along the lateral side of the blade (Figs. 9A & 10A). This feature is most prominent toward the distal half of the blade. A similar ridge is seen in *L. astibiae* (Díaz, Suberbiola & Sanz, 2013) in comparison to the centrally located scapular blade ridge of *D. matildae* (Hocknull et al., 2009; Poropat et al., 2015b) (Figs. 9E & 10B) and *W. wattsi* (Hocknull et al., 2009; Poropat et al., 2015a) (Figs. 9C & 10C), which runs close to the midline of the blade, as observed in many titanosaurians (González Riga et al., 2019).

In *A. cooperensis* the acromion ridge is near straight, curving only slightly at its ventral extent. Both *W. wattsi* and *D. matildae* partially preserve the acromion plate; however, the acromion ridge is only observable in *W. wattsi*. In *W. wattsi* it is curved anteriorly toward its ventral margin and terminates about the midline of the scapular plate and blade. The posterior margin of the acromion process is rounded and narrower in *W. wattsi* compared to the flat and relatively broad region of *A. cooperensis*. In both *W. wattsi* and *D. matildae* the acromion plate is thicker mediolaterally and less medially deflected compared to *A. cooperensis*.

*Medial View*. The scapular plate preserves a deep fossa created by the medial curvature of the scapular plate and an excavated medial side of the acromial ridge and scapular blade base. This large fossa is interpreted to be a proximal location for the M. subscapularis (Fig. 9B). The fossa in *D. matildae* (Hocknull et al., 2009; Poropat et al., 2015b) (Fig. 9F) and *W. wattsi* (Hocknull et al., 2009; Poropat et al., 2015a) (Fig. 9D) is not as deep, and in both of these taxa there exists a small and distinct medial tuberosity muscle scar distal to the fossa near the midline of the scapular blade. This feature has not been observed in other taxa illustrating the medial view of the scapula, so it could be considered a shared characteristic of these two taxa. Such a medial tuberosity is missing from *A. cooperensis* and helps differentiate it from *D. matildae* and *W. wattsi*.

The bone making up the acromion process is thin and excavated from the medial side of the scapular plate to be level with the dorsal margin of the scapular blade. The bone then thickens mediolaterally toward the dorsal margin of the acromion process, forming a rounded buttress for the process. The scapular blade base is straight with sub-parallel

dorsal and ventral margins. The ventral margin has been crushed and the bone making up the proximoventral margin of the scapular blade has been deformed vertically and medially. The ventral margin of the blade is rounded and slightly thicker than the dorsal margin toward the scapular blade base, which on the lateral side, forms a slightly raised ridge running along the ventrolateral margin of the blade. There is no indication of this ridge occurring on the medial side; therefore, the ridge is a lateral expansion of bone only along this lateral margin.

*Distal View.* The scapular blade bends only slightly laterally along its length toward the distal end. Half way along the shaft, the blade is slightly laterally deformed. However, this does not alter the overall form of the blade being very straight and only slightly curved laterally. The distal end of the blade is not preserved, so it is difficult to estimate the distance from the broken margin to the scapular blade's distal extremity. The bone thickness does not alter significantly along its length suggesting the blade could have continued significantly further than what is preserved, especially when comparison is made to the same area of cross-sectional shape in *D. matildae* and *W. wattsi* (Fig. 10), and in comparing the distal cross-sectional shape of *Y. datangi* (see Fig. 1E in (*Li et al., 2014*)). The cross-sectional shape along the length of the scapular blade is shallowly curved and sub-rectangular with no distinct lateral ridge along the midline of the scapular blade or any medial excavation or fossa (Figs. 9G–9I & 10A–10C).

Although not completely preserved, the scapula possesses a combination of features that warrant comparison across titanosauriforms. The taxa that exhibit some of the suite of features seen in the scapula of *A. cooperensis* include *Y. datangi*,(*Li et al., 2014*), *L. astibiae* (*Díaz, Suberbiola & Sanz, 2013*), *Dr. schrani* (*Ullmann & Lacovara, 2016*), *Chubutisaurus insignis* (*Carballido et al., 2011a*) and *V. daparisensis* (*Mannion, Allain & Moine, 2017*). They all possess relatively narrow scapular blades that have close to parallel dorsal and ventral margins with poorly expanded distal margins and lack a central scapular blade ridge.

Considering the diversity of scapulae shapes across Titanosauriformes, taxa tend to possess either; (1) a dorsoventrally broad acromion plate with a dorsoventrally narrow scapular blade that is markedly expanded posteriorly (e.g. *Tehuelchesaurus benitezii*, see Fig. 14 in (*Carballido et al., 2011b*)); (2) a broad acromion plate with a dorsoventrally narrow scapular blade that is not expanded posteriorly with sub-parallel dorsal and ventral margins (e.g. *Dr. schrani*, see Fig. 2 in (*Ullmann & Lacovara, 2016*)); (3) a broad acromion plate with a dorsoventrally deep scapular blade that is expanded posteriorly (e.g. *P. mayorum*, see Fig. 2H in (*González Riga et al., 2019*)); (4) a dorsoventrally narrow acromion plate with a dorsoventrally narrow scapular blade that is not expanded posteriorly with subparallel dorsal and ventral margins (e.g. *Y. datangi*, see Fig. 11E in (*Li et al., 2014*)); and (5) a narrow acromion plate with dorsoventrally broad scapular blade that is expanded posteriorly (e.g. *Mendozasaurus neguyelap*, Fig. 2G in (*González Riga et al., 2019*)). *A. cooperensis* shares features most closely with the titanosaurians similar to *Y. datangi* in scapular morphology, whilst the other Winton Formation taxa that have

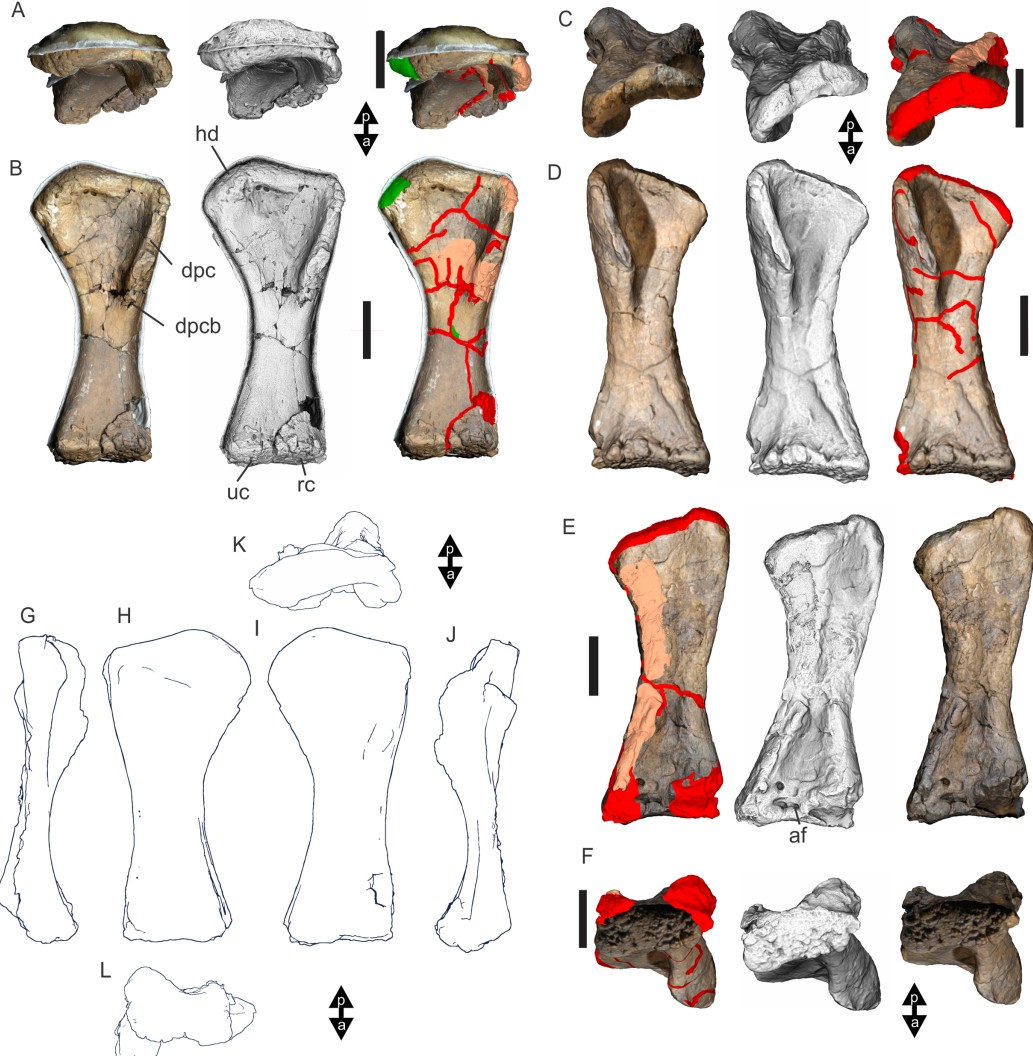

**Figure 12 Humeri of *Diamantinasaurus matildae* (AODF603).** (A & B) Left humerus in proximal (A) and anterior (B) views. (C–F) Right humerus in proximal (C), anterior (D), posterior (E) and distal (F) views. (G–L) Reconstructed left humerus using the left and right (mirrored) humeri in medial (G), posterior (H), anterior (I), lateral (J), proximal (K) and distal (L) views. 3-D image rendering methods used included, natural, (A–D (left), E & F (right)); ambient occlusion with radiance scaling, A–F (middle); coloured schematic (see Fig. 8), A–D (right), E–F (left) and orthogonal outline edge detection (G–L). Arrows indicate direction (a, anterior; p, posterior). Feature abbreviations: af, anconeal fossa; dpc, deltopectoral crest; dpcb, deltopectoral crest base; hd, humeral head; rc, radial condyle; uc, ulnar condyle. Scale bars = 20 cm.

comparative scapulae (*W. wattsi* and *D. matildae*) more closely resemble each other and titanosaurians with scapulae like *M. neguyelap*.

**Humeri (Figs. 11–16) (Table 3).** The humerus will be described with the diaphysis long axis oriented vertically and the distal condyles horizontal and perpendicular to the diaphysis long axis. The holotype preserves both humeri; a partial left and a nearly complete right humerus. The left humerus is missing the proximal epiphysis and much of

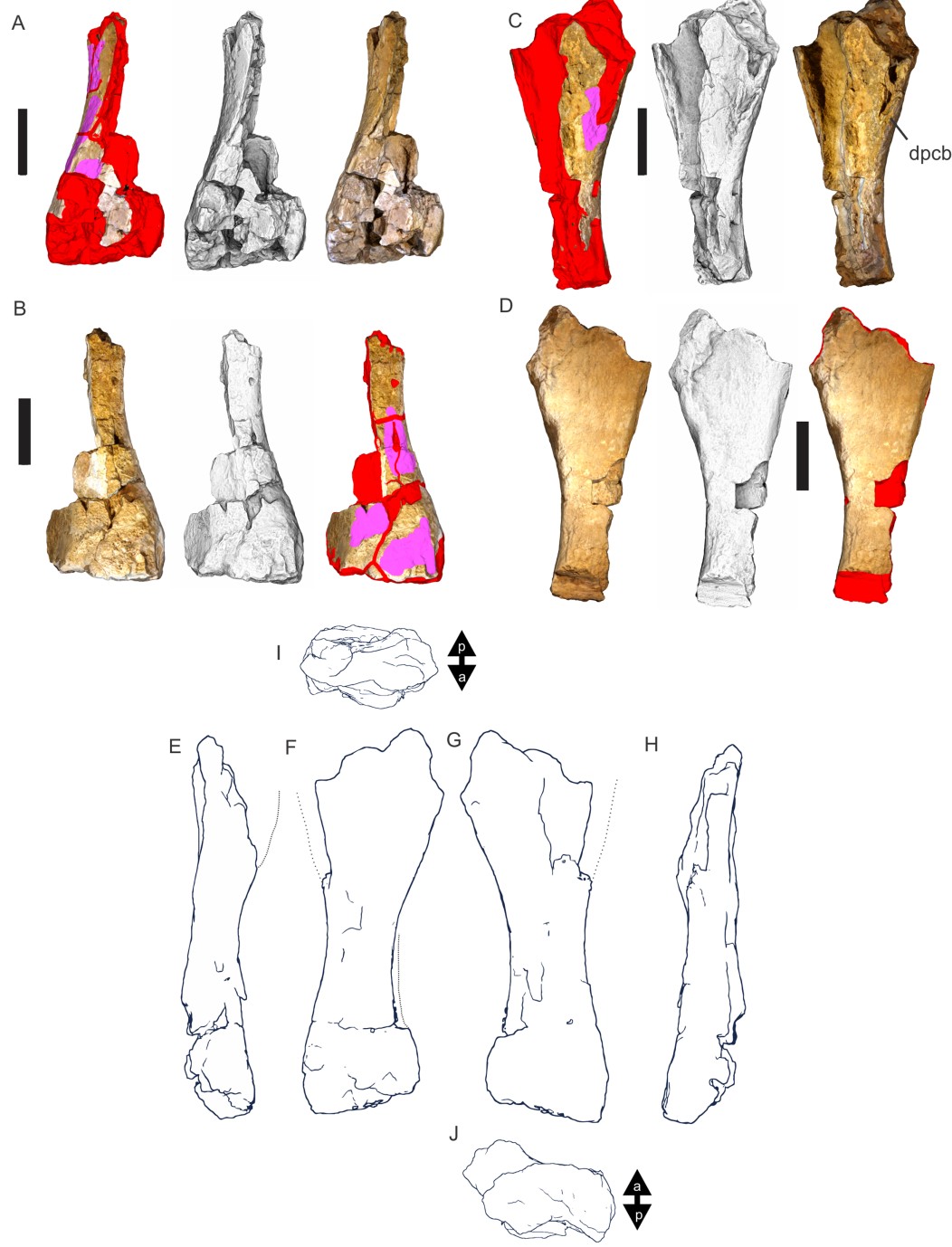

**Figure 13 Humeri of *Wintonotitan wattsi* (QMF7292).** (A & B) Partial right humerus in posterior (A) and anterior (B) views. (C & D). Partial left humerus in anterior (C) and posterior (D) views. (E–J) Reconstructed right humerus using partial left (mirrored) and right humeri in medial (E), posterior (F), anterior (G), lateral (H), proximal (I) and distal (J) views. 3-D image rendering methods used included, natural, A & C (right), B & D (left); ambient occlusion with radiance scaling, A–D (middle); coloured schematic (see Fig. 8), A & C (left), B & D (right); orthogonal outline edge detection (E–J). Arrows indicate direction (a, anterior; p, posterior). Feature abbreviations: dpcb, deltopectoral crest base. Scale bars = 20 cm.      

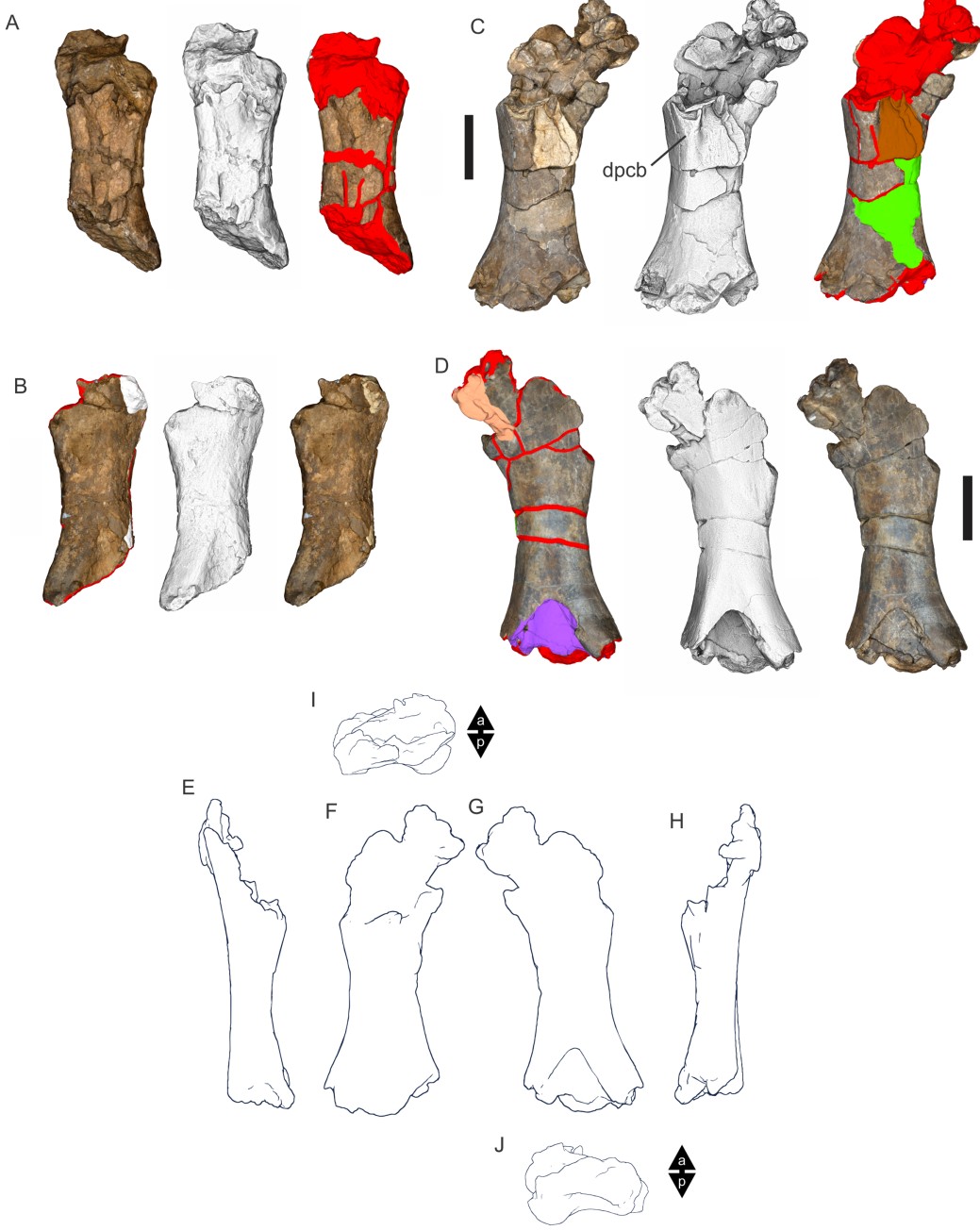

**Figure 14 Humeri of *Savannasaurus elliottorum* (AODF660).** (A & B) Left partial humerus in anterior (A) and posterior (B) views. (C & D) Right partial humerus in anterior (C) and posterior (D) views. (E–J) Reconstructed right humerus using partial left (mirrored) and right humeri in medial (E), posterior (F), anterior (G), lateral (H), proximal (I) and distal (J) views. 3-D image rendering methods used included, natural, A & C (left), B & D (right); ambient occlusion with radiance scaling, A–D (middle); coloured schematic (see Fig. 8), A & C (right), B & D (left); orthogonal outline edge detection (E–J). Arrows indicate direction (a, anterior; p, posterior). Feature abbreviations: dpcb, deltopectoral crest base. Scale bars = 20 cm.

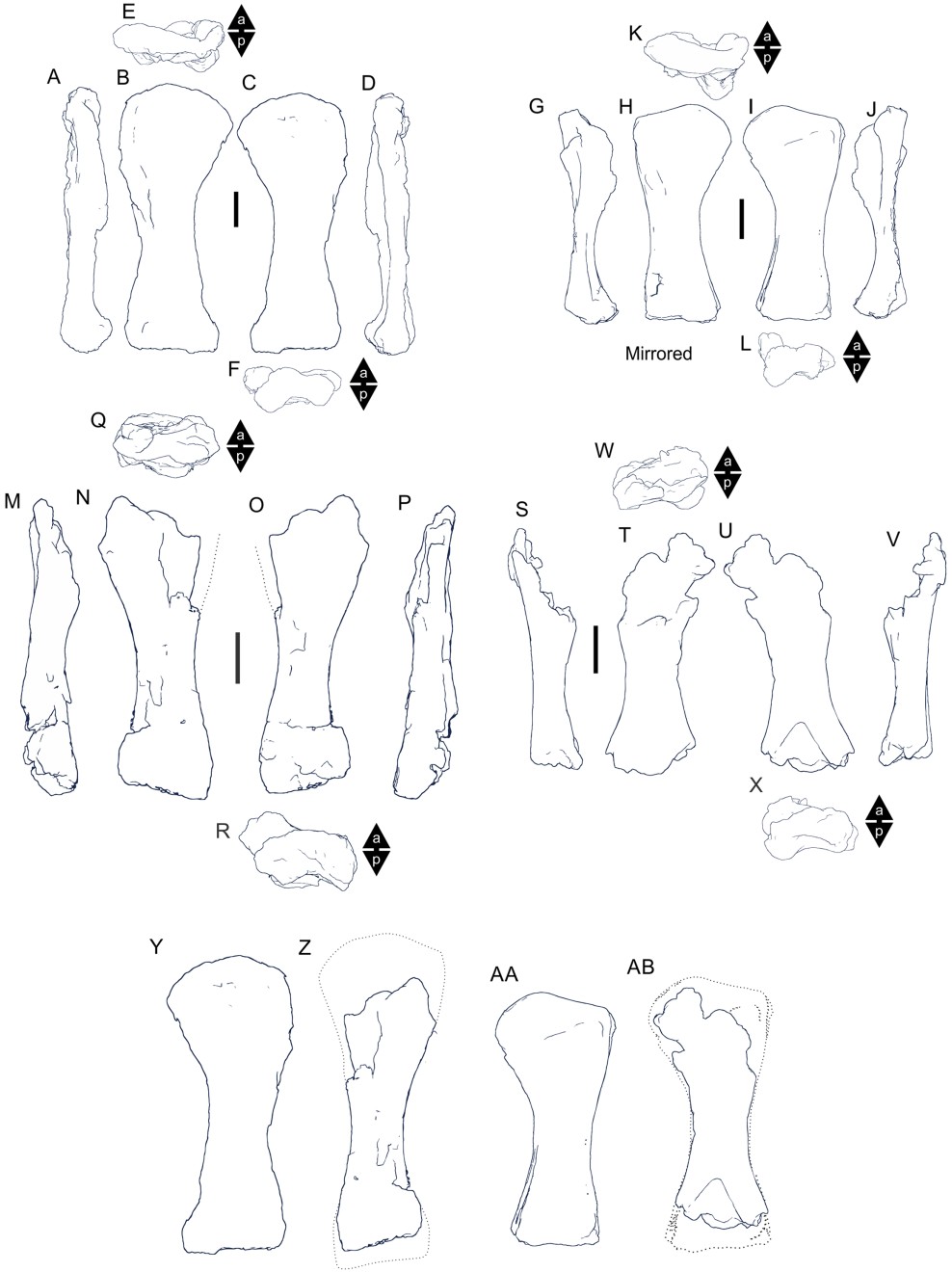

**Figure 15 Comparisons of Winton Formation sauropod humeri.** (A–F) *Australotitan cooperensis* gen. et sp. nov. (EMF102) right humerus in medial (A), anterior (B), posterior (C), lateral (D), proximal (E) and distal (F) views. (G–L) *D. matildae* (AODF602), reconstructed as right humerus, in medial (G), anterior (H), posterior (I), lateral (J), proximal (K) and distal (L) views. (M–R) *W. wattsi* (QMF7292) reconstructed as right humerus, in medial (M), anterior (N), posterior (O), lateral (P), proximal (Q) and distal (R) views. S–X. *S. elliottorum* (AODF660), reconstructed as right humerus, in medial (S), anterior (T), posterior (U), lateral (V), proximal (W) and distal (X) views. Y–AB. Reconstructed right humeri of *A. cooperensis* gen. et sp. nov. (Y), *W. wattsi* (Z), *D. matildae* (AA) and *S. elliottorum* (AB) scaled to minimum mediolateral width of the midshaft. Dotted lines estimating missing portions and shape of humerus. All 3-D models rendered using orthogonal outline edge detection. Arrows indicate direction (a, anterior; p, posterior). Scale bars = 20 cm.

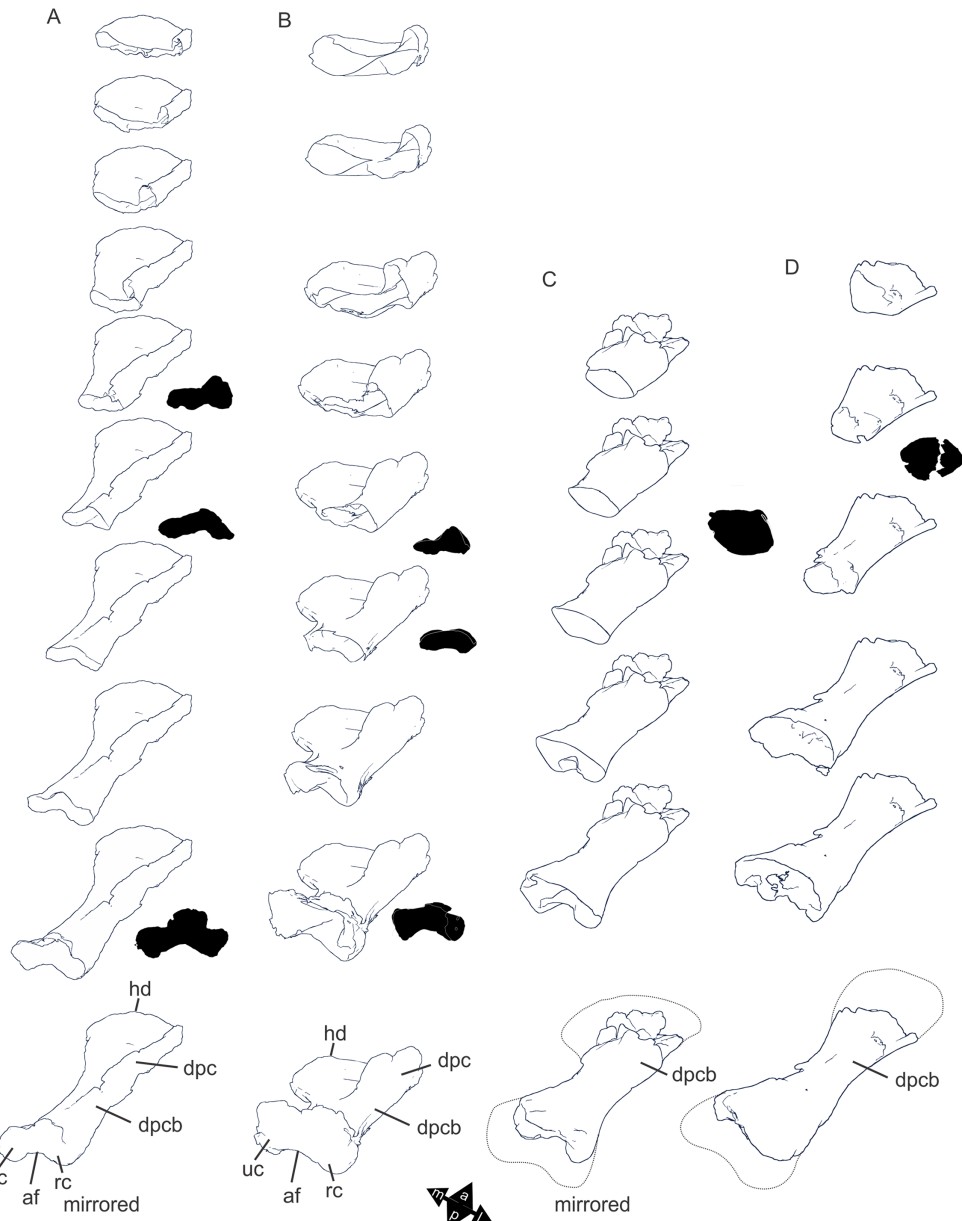

**Figure 16 Comparisons of Winton Formation sauropod humeri in cross-section, scaled to minimum mediolateral midshaft width.** (A) *Australotitan cooperensis* gen. et sp. nov. (B) *Diamantinasaurus matildae*. (C) *Savannasaurus elliottorum*. (D) *Wintonotitan wattsi*. Dotted line represents estimated extent of bone. Arrows indicate direction (a, anterior; l, lateral; m, medial; p, posterior). Feature abbreviations: af, anconeal fossa; cbr, coracobrachialis scar; dpc, deltopectoral crest; dpcb, deltopectoral crest base; hd, humeral head; rc, radial condyle; uc, ulnar condyle.

the medial margin of the diaphysis. Most of the lateral margin of the limb is preserved from just distal of the proximolateral corner along the deltopectoral crest, including the distal portion of the diaphysis and distal epiphysis, from the distolateral flange and ectepicondyle to the distomedial flange and entepicondyle. The cortical bone is heavily split, forming three main sections that join together. Portions of the deltopectoral crest

**Table 3 Humerus measurements of Winton Formation sauropods.**

| | Left (Preserved) | Right (Preserved) | Model |
|---|---|---|---|
| EMF102 *Australotitan cooperensis* | EMF102 *Australotitan cooperensis* | | |
| Maximum proximodistal length (d) | 1,394.87+ | 1,494.73 | 1,500.25 |
| Maximum medial, proximodistal length | | 1,479.75 | 1,448.58 |
| Maximum lateral, proximodistal length | 1,329.61+ | 1,390.54 | 1,433.06 |
| Maximum mediolateral width of proximal epiphysis | | <723.26 | 667.95 |
| Maximum anteroposterior length of proximal epiphysis | | 79.42+ | 114.23 |
| Maximum mediolateral width across distal condyles | <561.87 | 514.83+ | 516.78 |
| Maximum anteroposterior length of distal medial condyle | 118.92+ | 186.25 | 173.11 |
| Maximum anteroposterior length of distal centro-condyle | 126.46+ | 187.27+ | 204.31 |
| Maximum anteroposterior length of distal lateral condyle | 99.44+ | 158.15+ | 172.02 |
| Maximum midshaft mediolateral width (a) | | 333.34 | 335.81 |
| Minimum midshaft anteroposterior length (b) | | 101.80 | 101.67 |
| Minimum mediolateral width (c) | | 292.8 | |
| Proximal epiphysis circumference | | 1564.36+ | 1621.54 |
| Midshaft circumference (incl. base of deltopectoral crest (dpc)) | | 1021.55+ | 1041.69 |
| Minimum diaphyseal circumference | | 759 | 759 |
| Distal condyles circumference | 1,238.31 | 1,351.50+ | 1,368.78 |
| (b)/(a)–midshaft length to width | | 0.30 | 0.30 |
| (c)/(d) | | 0.19 | |
| AODF603 *Diamantinasaurus matildae* | | | |
| Maximum proximodistal length (d) | 1,122.35+ | 1,056.34+ | 1,154.11 |
| Maximum medial, proximodistal length | 1,105.9 | 961.84+ | 1,131.86 |
| Maximum lateral, proximodistal length | 1,049.68 | 1,056.55 | 1,105.53 |
| Maximum mediolateral width of proximal epiphysis | 487.66 | 392.75+ | 510.27 |
| Maximum anteroposterior length of proximal epiphysis | | 119.75 | 119.75 |
| Maximum mediolateral width across distal condyles | 379.56+ | 403.44+ | 448.37 |
| Maximum anteroposterior length of distal medial condyle | | 208.16 | 208.16 |
| Maximum anteroposterior length of distal lateral condyle | | 195.19 | 195.19 |
| Maximum midshaft mediolateral width (a) | 229.38 | 230.38 | 230.38 |
| Minimum midshaft anteroposterior length (b) | | 81.4 | 81.4 |
| Minimum mediolateral width (c) | | | 234.11 |
| Proximal epiphysis circumference | 735.29+ | 1,047.71+ | 1,338.46 |
| Midshaft circumference (incl. base of dpc) | 331.62+ | 580.83 | 580.83 |
| Minimum diaphyseal circumference | | | 559.26 |
| Distal condyles circumference | 627.37+ | 1099.51 | 1,128.89 |
| (b)/(a)–midshaft length to width | | 0.35 | 0.35 |
| (c)/(d) | | | 0.20 |
| QMF7292 *Wintonotitan wattsi* | | | |
| Maximum proximodistal length (d) | 787.55+ | 617.05+ | 924.45+ |
| Maximum medial, proximodistal length | n/a | 575.09+ | 785.63+ |
| Maximum lateral, proximodistal length | 726.01+ | | 878.31+ |

(Continued)

| | Left (Preserved) | Right (Preserved) | Model |
|---|---|---|---|
| Maximum mediolateral width of proximal epiphysis | 333.58+ | | 333.58+ |
| Maximum anteroposterior length of proximal epiphysis | 71.7+ | | 71.7+ |
| Maximum midshaft mediolateral width (a) | 183.49+ | 100.14+ | 241.15 |
| Minimum midshaft anteroposterior length (b) | 115.72 | 94.71+ | 115.72 |
| Minimum mediolateral width (c) | | | 248.81 |
| Midshaft circumference (incl. base of dpc) | 447.16+ | 187.92+ | 674.30 |
| Minimum diaphyseal circumference | | | 583.48 |
| (b)/(a)–midshaft length to width | | | 0.48 |
| **AODF660 *Savannasaurus elliottorum*** | | | |
| Maximum proximodistal length (d) | 577.34+ | 1,020.78+ | 1112 est. |
| Maximum medial, proximodistal length | | 864.07+ | 864.07+ |
| Maximum lateral, proximodistal length | | 878.6+ | 878.6+ |
| Maximum mediolateral width of proximal epiphysis | | 300.73+ | 300.73+ |
| Maximum midshaft mediolateral width (a) | 232.6+ | 243.55 | 243.55 |
| Minimum midshaft anteroposterior length (b) | 136.77 | <171.97 | 136.77 |
| Minimum mediolateral width (c) | | | 223.30 |
| Midshaft circumference (incl. base of dpc) | 666.49+ | <727.66 | 713.04 |
| Minimum diaphyseal circumference | | | 601.54 |
| (b)/(a)–midshaft length to width | | | 0.56 |

**Notes:**
All measurements in mm.
+, full length not preserved.
est., estimated.
<, less than.

were collected as surface scatter, having been dislodged from the main distal epiphysis and weathered and exposed at the ground surface. These elements cleanly fit together and also fit to the main piece recovered within the transitional horizon between the overlying vertosol and the underlying Winton Formation.

The right humerus is relatively well preserved although the cortical surface bone is heavily split into a mosaic-like pattern similar to the left humerus. A thin crust of cemented siltstone with woody debris covered the element prior to preparation. The posterior side of the right humerus was facing up in the deposit as the top surface and has suffered significant weathering of the surface bone through the actions of the vertosol. The anterior face was oriented downwards and had been somewhat protected from this weathering. The right deltopectoral crest is flattened laterally due to collapse that occurred during plaster jacket removal during excavation. However, the relative positions of each distorted region are identifiable and this enables us to reconstruct the pre-collapsed state of the deltopectoral crest and thus understand the shape of the proximolateral corner. By combining the 3-D photogrammetric models created from both humeri, we retrodeformed the deltopectoral crest so that accurate description of the humerus would be possible (see "Methods") (Figs. 8K–8N & 11H).

*Anterior view*. The proximal and distal epiphyses are widely expanded relative to a narrow midshaft, as seen in most sauropod humeri, but further expanded mediolaterally as seen in titanosauriform sauropods. The proximal epiphysis is rounded, with the humeral head proximomedially directed and the proximolateral corner is rounded, similar to *V. daparisensis* (*Mannion, Allain & Moine, 2017*), *Zby atlanticus* (*Mateus, Mannion & Upchurch, 2014*) and *Alamosaurus sanjuanensis* (*Lehman & Coulson, 2002*), in comparison to a distinct right-angled 'corner' that is seen in the outlines of *D. matildae*, *Sa. loricatus*, *Epachthosaurus sciuttoi*, *Neuquensaurus australis* and *M. neguyelap* compared with the same feature in *Panamericansaurus schroederi*, *Tornieria africana* and *Kotasaurus yamanpalliensis* (see Fig. 16 in (*González Riga & David, 2014*)).

The distorted (flattened) proximolateral margin makes the specimen look like it possesses a distinct proximolateral corner; however, this is an artefact of deformation. When reorienting the deltopectoral crest the proximolateral margin exhibits a more rounded appearance in comparison to taxa showing the distinct proximolateral corner. The proximal anterior fossa forms a shallow and broad depression from the proximomedial margin of the deltopectoral crest to the proximolateral margin of the humeral head. A small raised rugosity is just medial to the center of the proximal anterior fossa.

The deltopectoral crest rises anteriorly from the proximolateral corner, thickens toward the midshaft of the diaphysis and is thickest at approximately a third the maximum proximodistal length measured from the proximal margin. This thickening at the apex of the deltopectoral crest is rugose and forms a tuberosity on the crest. The deltopectoral crest forms a shallow curve originating from the proximolateral margin in a distomedial direction onto the anterior face of the diaphysis where it expands into a shallowly rounded ridge that continues distally and expands mediolaterally toward the medial condyle of the radial-ectepicondylar region.

The medial margin distal to the humeral head curves laterally toward the midshaft of the diaphysis, then straightens along the midshaft and curves medially toward a medially expanded entepicondylar margin of the distal epiphysis. At the midshaft of the diaphysis the lateral margin extends distolaterally from underneath the deltopectoral crest into a broad ectepicondylar flange that curves slightly laterally toward the rounded distolateral corner. The distal epiphysis is broad due to both the medial and lateral margins expanding distally to respective epicondylar regions.

The ectepicondylar region comprises two main articular regions, the radial condyle and the flattened ectepicondyle. The radial condyle consists of two small condyles coalesced on the distal articular surface. The medial condyle is rounded and smaller than the sub-triangular lateral condyle, they are split apart by a crack. The ectepicondyle is separated from the radial condyles by a shallow distal anterior fossa; however, it too is connected to the radial condyles through the distal articular surface. The distal articular surface is anteroposteriorly convex curving up onto the distal margin of the distoanterior face. The entepicondylar region comprises a large rounded ulnar condyle that is mediolaterally expanded and rounded medially. The distal articular surface curves anteroposteriorly onto the anterior face, but not to the extent seen in the radial condyle. A

shallow and elongate fossa divides the anterior face of the ulnar condyle from the radial condyle and the low central ridge that extends from the deltopectoral crest.

*Posterior view.* The proximal epiphysis is poorly preserved, missing portions of the humeral head; however, based on the distribution of the surface bone preserved it indicates a relatively thick posterior expansion of the humeral head, thicker than the anterior humeral head bulge. There is a large, broad and rounded posterior ridge that expands from the medial flange laterally to approximately the midline of the shaft. The medial fossa (medial fossa for the M. scapulohumeralis) is significantly reduced to a small flat region along the medial flange. The lateral fossa (lateral fossa for the M. scapulohumeralis) is large, broad and shallow. The lateral margin of the diaphysis, distal to the level of the deltopectoral crest, is curved medially and expanded distolaterally to the ectepicondylar region. This region lacks any representation of a tuberosity or strong bulge as seen in *Opisthocoelicaudia skarzynskii* (see Fig. 7 in (*Borsuk-Bialynicka, 1977*)), but could be preservational loss.

The medial margin of the diaphysis has been distorted by internal collapse to form a narrow fissure along the mid-length of the shaft in a proximodistal orientation. The surface cortical bone is still traceable along the margins of this fissure and shows that the fissure is an artefact of preservation. The olecranon (=anconeal) fossa is elongate and subtriangular in shape with the tallest apex starting at the level of the midshaft of the diaphysis, just distal to the level of the deltopectoral crest termination. The fossa broadens distally and is shallow along its length. The distolateral expansion for the distal condyles creates a steep medial margin for the fossa, whilst the medial side of the fossa remains broadly shallow.

*Proximal view.* Proximal epiphysis cross-section through the mid-level of the anterior fossa is anteroposteriorly narrow, elliptical, and slightly curved posteriorly. Midshaft diaphysis cross-section is bi-lobed subrectangular in shape, taking into account the internal collapse along the medial margin and distal extremity of the deltopectoral crest. The distal epiphysis cross-section through epicondylar region is tri-lobed with shallow fossae dividing each lobe. The anterior portion of the humeral head is anteroposteriorly moderately expanded and rounded anteromedially. The posterior face of the humeral head is poorly preserved with indications of thickening in a posterior direction to form a relatively broad humeral head. The deltopectoral crest is near perpendicular to the proximal anterior fossa and curved medially. The deltopectoral crest remains vertical along its length and its base curves medially toward the center of the anterior face of the diaphysis. The vertical projection and apex of the crest remains vertical and does not curve medially to project across the anterior face of the humerus.

*Distal view.* The distal condylar region is tri-lobed and sub-equal in size. The radial condylar region is made up of a rounded radial condyle, which is divided into two small condyles, and a large ectepicondyle that is similar in size to the radial condyle itself. The ulnar condyle is offset posteromedially from the radial condylar region via a shallow groove. The ulnar condyle is similar in size to the radial condyle. The entepicondylar region is rounded and not as expanded relative to the ectepicondylar corner.

Three of the four currently recognised Australian Cretaceous sauropod taxa possess humeri: *D. matildae* (Fig. 12), *W. wattsi* (Fig. 13), and *S. elliottorum* (Fig. 14) do, whilst *Austrosaurus mckillopi* does not. Only *D. matildae* is complete enough with minimal deformation for good comparisons. Both *W. wattsi* and *S. elliottorum* can only be compared for central diaphysis shape and relative proportions (Figs. 12–16, Table 3). Both are missing the proximal and distal epiphyses due to significant pre-depositional breakage and surface weathering (i.e., *W. wattsi*) or pre-diagenetic loss and crushing (i.e., *S. elliottorum*).

The proximal region of the humerus in *A. cooperensis* differs from *D. matildae* by possessing: a more rounded proximolateral corner; a more rounded proximal articular margin in anterior view; a relatively thinner, more vertically oriented and more distally terminating deltopectoral crest; a relatively narrower humeral head and shallower proximal anterior fossa. Posteriorly, the posterior ridge is broader medially, and the medial fossa is reduced in *A. cooperensis*. *A. cooperensis* has more laterally and medially flared distal condyles (Figs. 15 & 16).

The diaphysis of *A. cooperensis* differs from *W. wattsi* and *S. elliottorum* by being considerably more elliptical in cross-sectional shape where *W. wattsi* and *S. elliottorum* present a much more ovo-rectangular cross-sectional shape relative to *A. cooperensis* and *D. matildae* (Fig. 16).

The humerus is hour-glass shaped, as is typical of most sauropods. The proximal margin compares most favorably with *Al. sanjuanensis* (*Gilmore, 1946*; *Lehman & Coulson, 2002*), *Turiasaurus riodevensis* (*Royo-Torres, Cobos & Alcala, 2006*), *V. daparisensis* (*Mannion, Allain & Moine, 2017*), *Haestasaurus becklesii* (*Upchurch, Mannion & Taylor, 2015*) and *Z. atlanticus* (*Mateus, Mannion & Upchurch, 2014*). These similarities are based on the outline curvature in anterior view of the proximal margin, differing from the 'sigmoidal' or 'sinuous' outline characterising other sauropods with similarly broad proximal epiphyses (e.g., *D. matildae*, *Sa. loricatus*, *N. australis* and *O. skarzynskii*).

The distal epiphysis in distal view forms a tri-lobate articular cross-sectional profile which is not seen in *D. matildae* (Figs. 11, 12 & 16), but is similar to *E. lilloi* (see Fig. 6E in (*Mannion & Otero, 2012*)), *Giraffatitan brancai* and *Ep. sciuttoi* (see Figs. 4F–4G in (*Upchurch, Mannion & Taylor, 2015*)). Contributing to the tri-lobate distal epiphysis is a deep olecranon fossa which is longer and deeper than in *D. matildae* but is similar to that of *E. lilloi*.

Considerable variation exists across titanosauriformes in regards to the overall shape of the humerus as illustrated by the outline drawings in Fig. 7 of (*Lehman & Coulson, 2002*), Fig. 16 of (*González Riga & David, 2014*) & Fig. 4 of (*González Riga et al., 2019*). The humeri of *A. cooperensis* share a combination of characteristics that are missing from more derived titanosaurians. The gently curved proximodorsally convex outline of the epiphyseal head is similar to that seen in *Tehuelchesaurus benitezii* and *V. daparisensis* and differs from the proximodorsally projecting sub-quadrangular outline typical of many titanosaurians like *C. insignis*, *D. matildae*, *N. australis*, *Notocolossus gonzalezparejasi* and *Paralititan stromeri*.

The distal epiphyses of *A. cooperensis* is mediolaterally broad, with clearly defined articular condylar areas that are anteroposteriorly compressed. This overall shape is similar to that seen in *Dr. schrani*, *Pa. stromeri* and *Malawisaurus dixeyi*, but differs from titanosaurians like *D. matildae*, *N. australis*, *E. lilloi*, and *No. gonzalezparejasi*, that possess a more rotund humerus that is not mediolaterally expanded, but anteroposteriorly deep.

**Ulna** (Figs. 17–19) (Table 4). The ulna will be described with the longest proximodistal length, taken from the distal articular surface to the olecranon process, oriented vertically. The main processes of the ulna are oriented anterolaterally and anteromedially with the radial fossa considered anterior. The holotype preserves a single almost complete right ulna. It is one of the best preserved and distinctive bones of the holotype specimen. The proximal region has experienced some weathering; however, much of the articular surfaces remain. The cortical bone of the anteromedial process and anterior and posterior faces of the diaphysis are heavily split into mosaic-like pieces; however, they are tightly arranged and have not moved significantly post-burial and excavation. The diaphysis has been deformed, bent downwards in situ, producing an anterolateral bend. This deformation is unlikely a result of subsurface vertical movement through soil action because the bend was downwards, or post-fossilisation turbation (e.g., wombats) because no evidence of sediment disturbance or infill with soil profile was observed at this site. A more likely conclusion is that this downward bend was a result of pre-fossilisation trampling. Digital retro-deformation of the shaft was possible and allowed a more accurate description of the bone and its dimensions. Referred ulna fragments from EMF164 include parts of the proximal diaphysis and the interosseous ridge of the distoanterior face.

*Anterior view*. Three distinct processes extend from the proximal epiphysis, the anterolateral, anteromedial and olecranon processes, in an arrangement typical of sauropods. The anterolateral and olecranon processes are of similar length with the anteromedial process being much longer than either of these. The anteromedial process is shallowly concave along its length ending at its extremity as a triangular point.
The anterolateral process is short and broad with a rounded extremity whilst the olecranon process is constricted mediolaterally and angled proximally into a tapered articular surface. Between the anterolateral and anteromedial processes, a deep radial fossa extends distally toward a distinct radial interosseous ridge. The lateral side of the fossa is steep, made up by the medial face of the anterolateral process. The medial side of the fossa is shallow and slightly curved, made up by the broad lateral face of the anteromedial process. The radial fossa extends distally to the beginning of the distal epiphysis.

The distal half of the fossa is shallow and a distinct and thick proximodistally oriented interosseous ridge extends along its center, terminating just proximal of the distal articular end. This feature is present in fragments of a large ulna of EMF164; therefore, such a unique feature allows us to confirm referral of EMF164 to this same taxon.

The anterolateral process is broader proximally, but is not a thick process. It extends the length of the diaphysis tapering along its length into a tall thin crest and terminates just proximal of the distal articular end. At the distal end of the anterolateral process a distinct

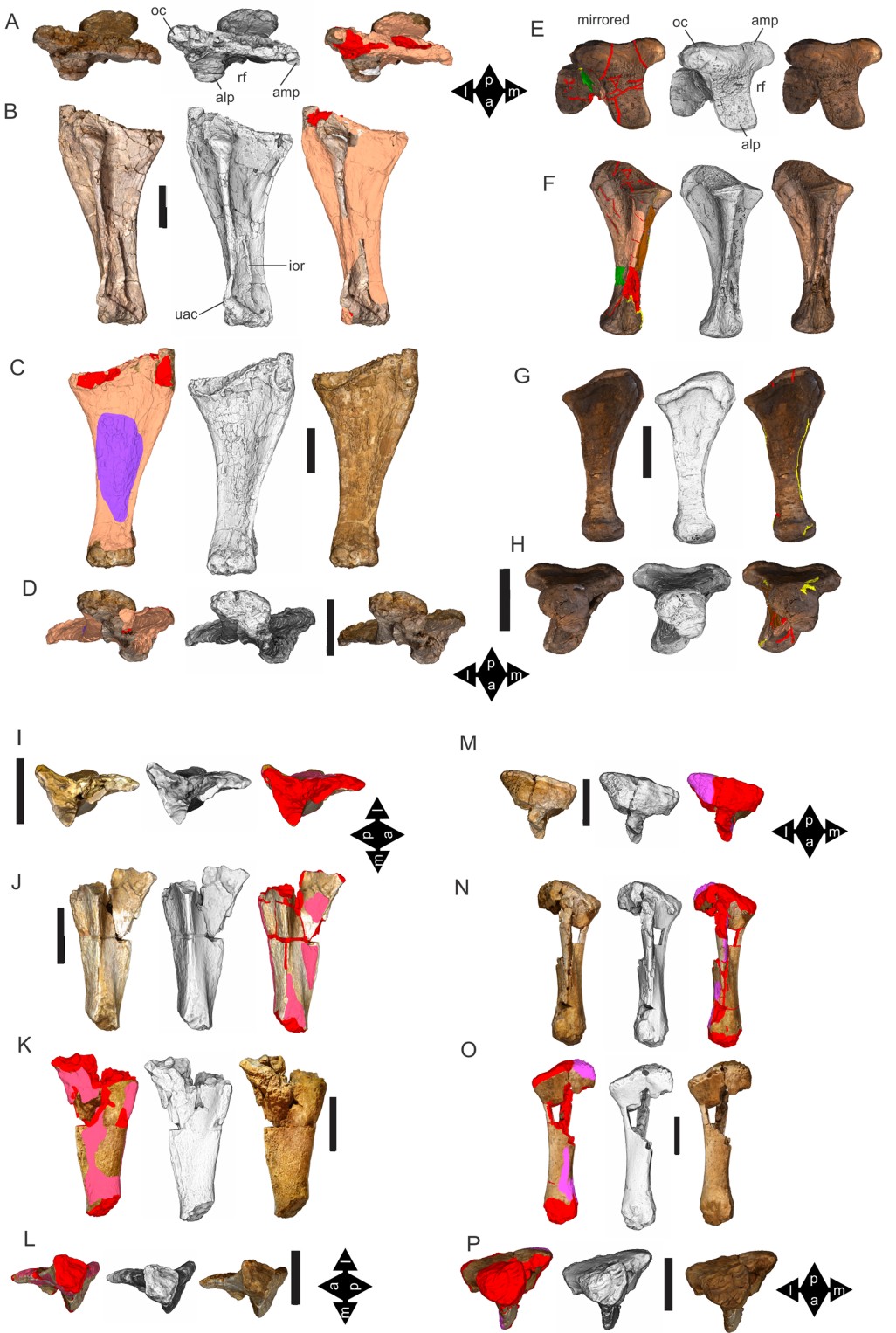

**Figure 17 Ulnae of *Australotitan cooperensis* gen. et sp. nov. (EMF102), *Diamantinasaurus matildae* (AODF603) and *Wintonotitan wattsi* (QMF7292).** (A–D) *A. cooperensis* gen. et sp. nov. ulna in proximal (A), anterolateral (B), medial (C) and distal (D) views. (E–H) *D. matildae* ulna in proximal (E), anteromedial (F), lateral (G) and distal (H) views. (I–P) *W. wattsi* ulnae in proximal (I & M),

**Figure 17** (continued)
anterolateral (J), anteromedial (N), medial (K), lateral (O) and distal (L & P) views. 3-D image rendering methods used included, natural, A, B, E, F, I, J, M & N (left); C, D, G, H, K, L, O, P (right); ambient occlusion with radiance scaling, A–P (middle); coloured schematic (see Fig. 8), A, B, E, F, I, J, M, N (right), C, D, G, H, K, L, O, P (left). Arrows indicate direction (a, anterior; l, lateral; m, medial; p, posterior). Feature abbreviations: alp, anterolateral process; amp, anteromedial process; ior, interosseous ridge of radial fossa; oc, olecranon process; rf, radial fossa; uac, distal ulnar accessory process. Scale bars = 20 cm.

crest of bone, an interosseous crest, smaller than the process itself extends slightly posterolaterally with a small rounded tuberosity at its apex. This tuberosity sits above another ridge of bone that extends anteriorly along the distal edge of the diaphysis and connects anteriorly to the distal articular region. There is no indication on the surface of the bone or surrounding this region to suggest that this unique set of features is distortion through preservation or from pathology.

Lateral to the anterolateral process is a narrow and deep posterolateral fossa bounded by the lateral face of the anterolateral process and the anterolateral face of the olecranon process. The fossa is broadest proximally and extends distally to about the midshaft level where it tapers to a shallow point before meeting the distal epiphysis. The anteromedial process curves steeply from its proximomedial extremity to the distal articular surface. The olecranon process is the highest of the three processes with its articular face oriented anteroproximally.

*Posterior view.* The anteromedial process is broad and flat with a shallow medial fossa extending across the process and distally to approximately two thirds of the proximodistal length. The olecranon process extends distally making a shallow sigmoidal curve, convex proximally and concave distally to the distal articular surface. The anterolateral process is straight in profile and sharply tapers distally to the distal tuberosity and accessory process and ridge.

*Proximal view.* Tri-radiate proximal end made up of an anterolateral, anteromedial and an olecranon process. Olecranon process smallest of the three, anterolateral process second largest whilst the anteromedial process is much longer than both extending approximately two and a half times the length of the anterolateral process. The angle created between the long axes of the anteromedial and anterolateral processes is approximately 50°.

*Distal view.* The distal articular surface is beveled proximally, and made of two clear lobes, a posteriorly placed mediolateral lobe and a small anterolateral lobe. The overall shape in distal view is oblong for the posterior lobe and rounded for the anterior lobe. The whole articular area is compressed anteroposteriorly so that the posterior region is not prominently expanded and more 'comma' shaped.

Overall, the ulna possesses the characteristic shape seen in many sauropod taxa. The stout nature of the ulna is similar to many titanosaurians like *D. matildae, Sa. loricatus, N. australis, Y. datangi* and *O. skarzynskii*. The presence of an accessory interosseous crest on the mediolateral process and an interosseous ridge within the radial

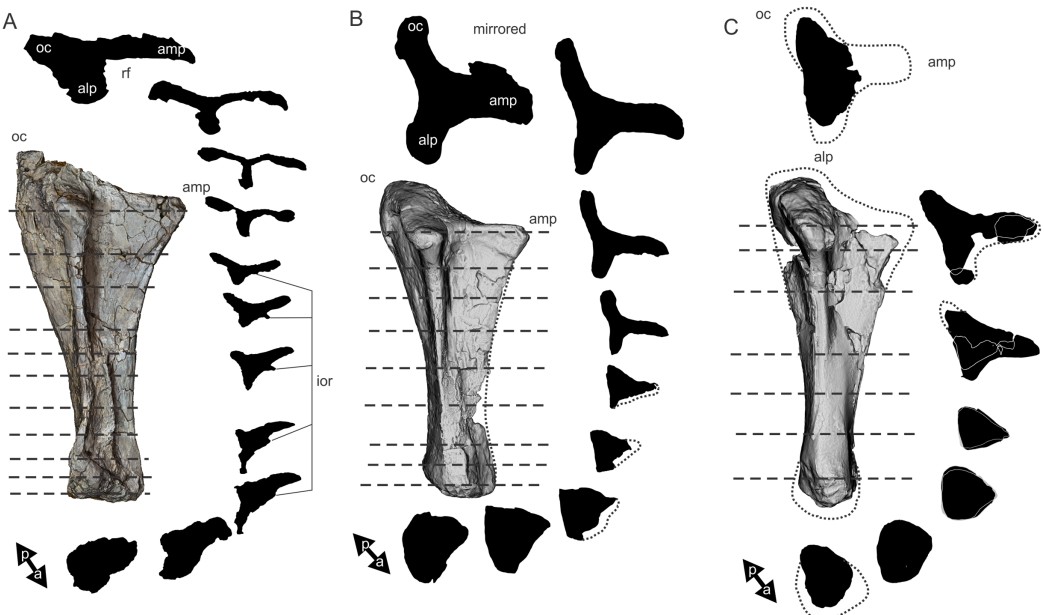

**Figure 18 Comparisons of Winton Formation sauropod ulnae in cross-section, scaled to minimum midshaft width.** (A) *Australotitan cooperensis* gen. et sp. nov. (B) *Diamantinasaurus matildae*. (C) *Savannasaurus elliottorum*. (D) *Wintonotitan wattsi* (reconstructed from both preserved ulnae). Abbreviations as in Fig. 16. Dashed line indicates position of cross-section. Dotted line indicates estimation of missing bone. Arrows indicate direction (a, anterior; p, posterior). Feature abbreviations: alp, anterolateral process; amp, anteromedial process; ior, interosseous ridge of radial fossa; oc, olecranon process; rf, radial fossa.

fossa is unique to this taxon. An accessory interosseous crest has been recently observed in the brachiosaur *V. daparisensis* (see Fig. 20A in (*Mannion, Allain & Moine, 2017*)); however, this feature does not originate from the anterolateral process as it does in *A. cooperensis*. Instead, the crest originates separately from it in a more medial position. Distinct interosseous ridges within the radial fossa of the ulna are observed in *Z. atlanticus* (*Mateus, Mannion & Upchurch, 2014*), *Rapetosaurus krausei* (*Curry Rogers, 2009*), *Bonitasaura salgadoi* (*Gallina & Apesteguía, 2015*) *Narambuenatitan palomoi* (*Filippi, García & Garrido, 2011*); and to a lesser degree of development in *N. robustus* (*Otero, 2018*) and *Dr. schrani* (*Ullmann & Lacovara, 2016*). With the exceptions of *A. cooperensis* and *N. robustus*, the interosseous ridge originates at approximately one third distal of the proximal epiphysis. In *A. cooperensis* and *N. robustus*, the ridge originates in the distal third of the shaft.

The ulna of *D. matildae* differ from *A. cooperensis* by both possessing a similar combination of features not present in *A. cooperensis*; including a relatively shorter anteromedial and relatively longer anterolateral and olecranon processes (in proximal view) (Figs. 17E–17H, 18 & 19); a taller and broader olecranon process; a less sinusoidal posterolateral ridge (in anterior view); the absence of an anterolateral distal interosseous crest or interosseous ridge within the distal radial fossa; a more inflated and rounded anterolateral and anteromedial margins of the distal epiphysis producing an inflated

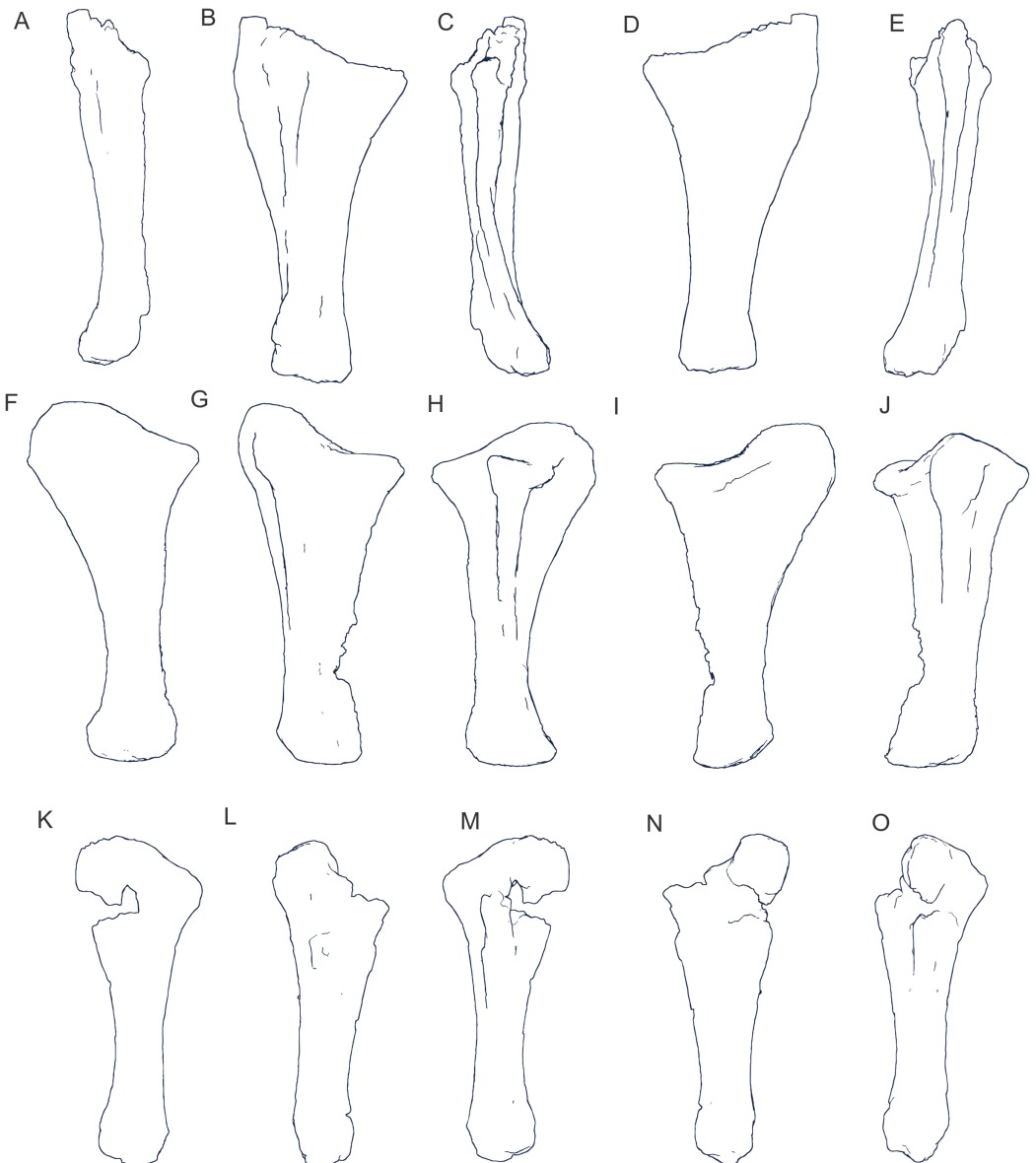

**Figure 19 Comparisons of Winton Formation sauropod ulnae in preserved right ulna outline, scaled to minimum midshaft width.** (A–E) *Australotitan cooperensis* gen. et sp. nov. in lateral (A), anterolateral (B), anteromedial (C), medial (D) and posterior (E). (F–J) *D. matildae* in lateral (F), anterolateral (G), anteromedial (H), medial (I) and posterior (J). (K–O) *W. wattsi* (reconstruction) in lateral (K), anterolateral (L), anteromedial (M), medial (N) and posterior (O). 3-D images rendered using orthogonal outline edge detection.

bean-shaped articular end in distal view; and a deeper fossa between the anteromedial and posterior processes.

The ulnae of *W. wattsi* are both poorly preserved missing the proximal and distal epiphyses and cannot be easily compared with *A. cooperensis* (Fig. 17). The reconstructed ulna (Fig. 18C) shows clear differences between *W. wattsi* and *A. cooperensis* along with *D. matildae* in regards to cross-sectional thickness of the anteromedial and anterolateral processes (Figs. 18 and 19). *W. wattsi* is distinctly more robust in cross-section. Previously

**Table 4 Ulna measurements of Winton Formation sauropods.**

|  | Preserved | Reconstructed |
|---|---|---|
| **EMF102 *Australotitan cooperensis*** | | |
| Maximum proximodistal length | 1,043.90+ | 1,056.35 |
| Olecranon – anteromedial process length | 501.34+ | 518.46 |
| Olecranon – anterolateral process length | 298.85 | 298.85 |
| Maximum distal condylar width | 225.05+ | 241.67 |
| Minimum distal condylar width | 122.65+ | 134.72 |
| Angle formed (amp-oc-alp) | 48° | 48° |
| Angle formed (oc-alp-amp) | 102° | 102° |
| Angle formed (alp-amp-oc) | 29° | 29° |
| **AODF603 *Diamantinasaurus matildae*** | | |
| Maximum proximodistal length | 727.83 | |
| Olecranon – anteromedial process length | 359.09 | |
| Olecranon – anterolateral process length | 321.98 | |
| Maximum distal condylar width | 204.37 | |
| Minimum distal condylar width | 157.53 | |
| Angle formed (amp-oc-alp) | 51° | |
| Angle formed (oc-alp-amp) | 71° | |
| Angle formed (alp-amp-oc) | 57° | |
| **QMF7292 *Wintonotitan wattsi*** | | |
| Maximum proximodistal length | 897.39+ | |
| Olecranon–anterolateral process length | 326.42+ | |

**Notes:**
All measurements in mm.
+, full length not preserved.
°degree of angle.

it has been reported that the left ulna of *W. wattsi* preserves the proximal and distal epiphyses (*Hocknull et al., 2009*; *Poropat et al., 2015a*), however, on inspection, both the left and right ulnae lack preserved proximal or distal articular ends or preserved epiphyses (Figs. 17–19). The proximal end of the left ulna is missing significant portions of the anteromedial and anterolateral processes. The olecranon is also missing the articular end with the surface exhibiting a pitted and corroded surface that can also be seen along the diaphyseal shaft (Figs. 17M & 17O). The distal end is missing and there is some indication of plant-debris adhering to this broken surface. Therefore, observations about the morphology of the ulnar condyles of *W. wattsi* are likely misinterpretations.

**Pelvis.** The right and left pubes and ischia were recovered together in semi-articulation and semi-life position with the dorsal side facing up in the deposit. The ilia were not found. Both pubes are well preserved; however, the cortical bone surface has been split into small mosaic-like pieces across the broad anterodorsal plates of the pubes and posterodorsal plates of the ischia. The pubes and ischia have split along the medial symphysis and reoriented sub-horizontally within the deposit, the cause of which is likely dinoturbation through trampling. The pubic blades are oriented slightly above horizontal. The ischial

blades have been dislocated slightly from their life position relative to the pubes; however, remain in near contact along their articular surfaces between each ischium and pubis.

**Pubes** (Figs. 20–22) (Table 5). *Lateral view*. The lateral (ventrolateral) views of both pubes represent the sides facing downward in the site resulting in this side being better preserved than the medial (dorsomedial) side. The left pubis is best preserved and will be used as the basis for most of the pubic description. The iliac peduncle sits dorsal of a shallow fossa that runs posteroventrally to the obturator foramen. Posterior of the obturator foramen the ischial peduncle is broken with matrix infill obscuring the lateral connection to the ischium. The anterior margin of the proximal blade extends ventrally from the iliac peduncle curving slightly ventrally toward the distal blade expansion. The ischial peduncle is connected and was co-ossified to the ischium along its entire length extending ventromedially to the midline, then joining with its contralateral pair. The ventral margin of the distal blade is divided into two regions of differing bone thickness with a line of collapsed bone forming an irregular groove from the ventral margin of the ischial peduncle across the pubic blade at about a third of the distance from the ventrolateral margin. This line of collapse indicates a distinct change in bone thickness from the main distal and proximal blade to the internal (medially directed) thin bone connection between the two contralateral elements.

*Medial view*. The medial (dorsomedial) view of both pubes represent the face exposed upwards in the site, therefore, the medial surface preserves a number of post-burial alterations to the bone surface. The right pubis has been affected more so than the left, with the surface cortical bone fractured into a mosaic tile of pieces with some collapse of internal bone and compression observed. Both pubes have some distortion to the central portion of the distal blades having been affected by crushing through trampling.

The iliac peduncle is better preserved in the left pubis. In medial view, it is broad and flat, taking up almost the entire proximal portion of the acetabulum. The peduncle is slightly expanded dorsally of the proximal blade plate which extends ventrally and curves medially to the central symphyseal surface. The posterior margin of the proximal blade is made up of the ischial peduncle which was fused to the pubic peduncle of the ischium along its entire length during life. In the left pubis the connection has been split and broken prior to fossilisation with the pubis medially and ischium laterally displaced relative to life position. The medial margin of the ischial peduncle has been split and dislodged vertically above the anterodistal margin of the pubic peduncle of the ischium. The opposite has occurred on the right side element with the pubis displaced laterally and the ischium medially.

Ventral of the posterior margin of the iliac peduncle and anterior of the ischial peduncle is an enclosed ovoid obturator foramen with a long axis oriented posterodorsally to anteroventrally. The symphyseal margin is thickest at both the posterior and anterior ends and has broken away from its contralateral pair exposing broken and open internal bone along its length, indicating that both blades were originally fused together. The bone

Peer J

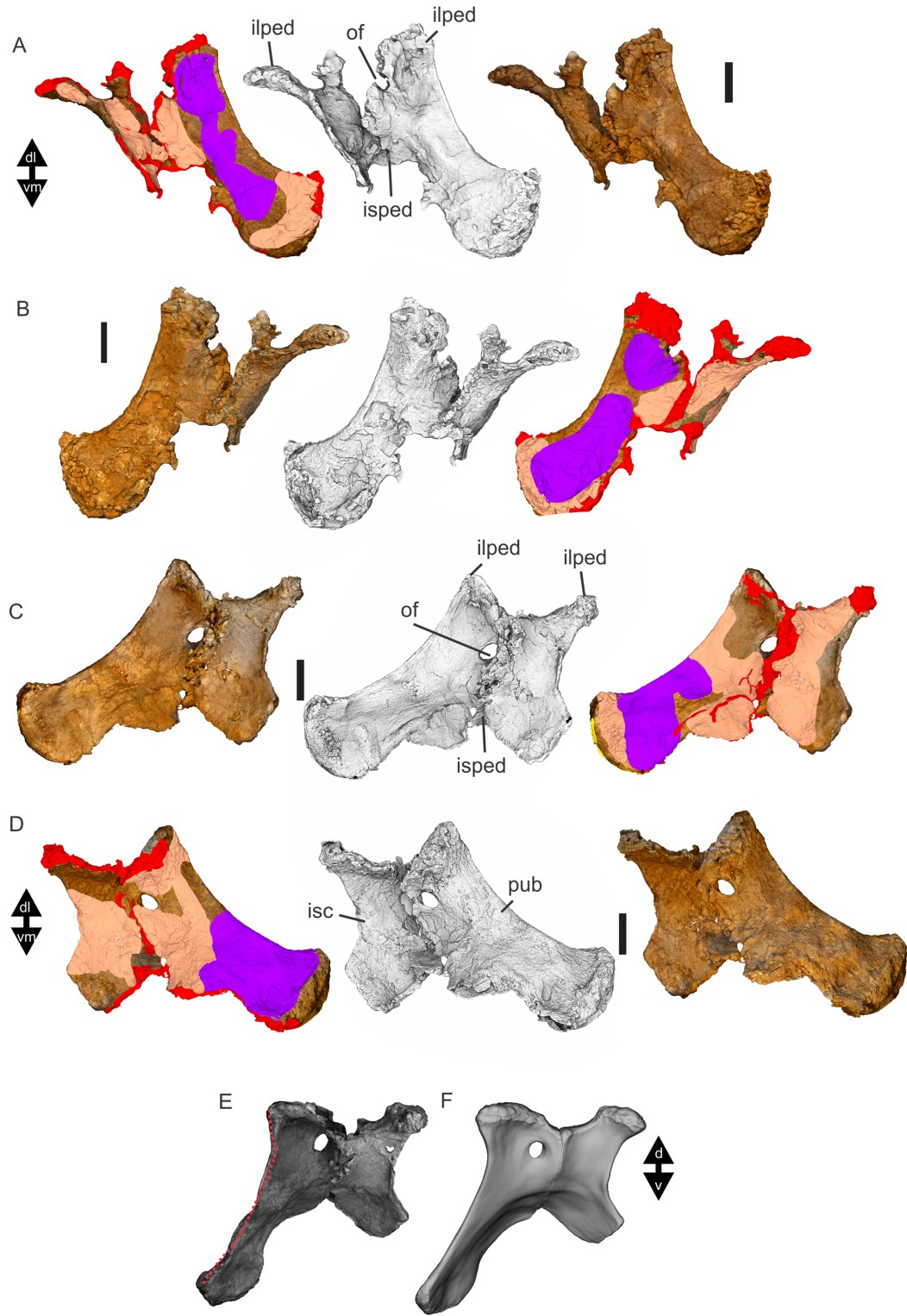

**Figure 20 Pubes and ischia of *Australotitan cooperensis* gen. et sp. nov. (EMF102).** (A & B) Right pubis and ischium in ventrolateral (A) and dorsomedial (B) views. (C & D) Left pubis and ischium in ventrolateral (C) and dorsomedial (D) views. (E) Preserved left pubis and ischium in lateral view, red dotted line indicating region of deformation. (F) Retrodeformed and digitally restored right pubis and ischium. 3-D image rendering methods used included, natural, A & D (right), B & C (left); ambient

Figure 20 (continued)
occlusion with radiance scaling, A–D (middle); coloured schematic (see Fig. 8), A & D (left), B & C
(right); vertex and texture uncoloured (E & F). Arrows indicate direction (d, dorsal; dl, dorsolateral; v,
ventral; vm, ventromedial). Feature abbreviations: ilped; iliac peduncle; isc, ischium; isped, ischial ped-
uncle; of, obturator foramen; pub, pubis. Scale bars = 20 cm.

connecting the contralateral elements is very thin along their length and curves ventrally to the massively expanded distal articular surface.

The distal articular surface is dorsoventrally thickened with a central fossa (preserved best in the left pubis). A shallow fossa runs along the distomedial surface behind the distal expansion. The lateral margin of the proximal blade begins lateral to the anterior margin of the iliac peduncle and curves ventrally at a very low angle toward the distal blade and distal expansion. In the left pubis, two abnormal indentations occur at the junction of the proximal and distal blades and just proximal of the distal expansion. These indentations appear to be the result of bone trampling. The original lateral margin would have been a smooth curved surface along its length as seen in the right pubis.

Based on the better-preserved left pubis, the iliac peduncle is oval in shape with tapered anterior and posterior margins, thickest in an anteromedial to distolateral direction. The region for the ambiens process is indistinct as the pubic blade runs directly ventral of the base of the iliac peduncle. Only a short acetabular surface is present posterior of the iliac peduncle on the pubis.

**Ischia (Figs. 20–22) (Table 6).** The left ischium is the least deformed of the ischia, preserving good and near complete margins and iliac and pubic peduncles. The iliac peduncle of the ischium is teardrop shaped with a rounded posterior and tapered anterior margin that runs into the acetabular surface. The acetabular surface is shallowly concave and approximately the same length as the iliac peduncle.

The anterior corner of the acetabular surface where it meets the pubic peduncle is dislocated posterodorsally from the corresponding puboischial articular surfaces, offsetting this articulation in a dorsoventral and mediolateral direction. The distal ischial symphysis is broken along its anteromedial margin indicating that these two elements were connected in life. However, complete bone is observed close to the central connection of both paired elements, suggesting that the four elements were fused along their respective articular surfaces except for the central point where all four elements meet (Fig. 21).

Instead, we reconstruct this area as having a slight opening that would have resembled a diamond-shaped gap between the four elements or exceptionally thin bone that has not preserved. The posteroventral margin of the ischium is unfused, but when mirrored form a distinct 'v' shaped margin (notch) between the mediodistal ends of each ischium when viewed dorsally. The proximal ischial plate is anteroposteriorly broad along its entire length and continues to retain this breadth distal of the pubic articulation, creating a broad posterodistal, but ventromedially projecting ischial shaft. A lateral tuberosity along the middle of the posterior ischial margin is a long thin buttress of bone.

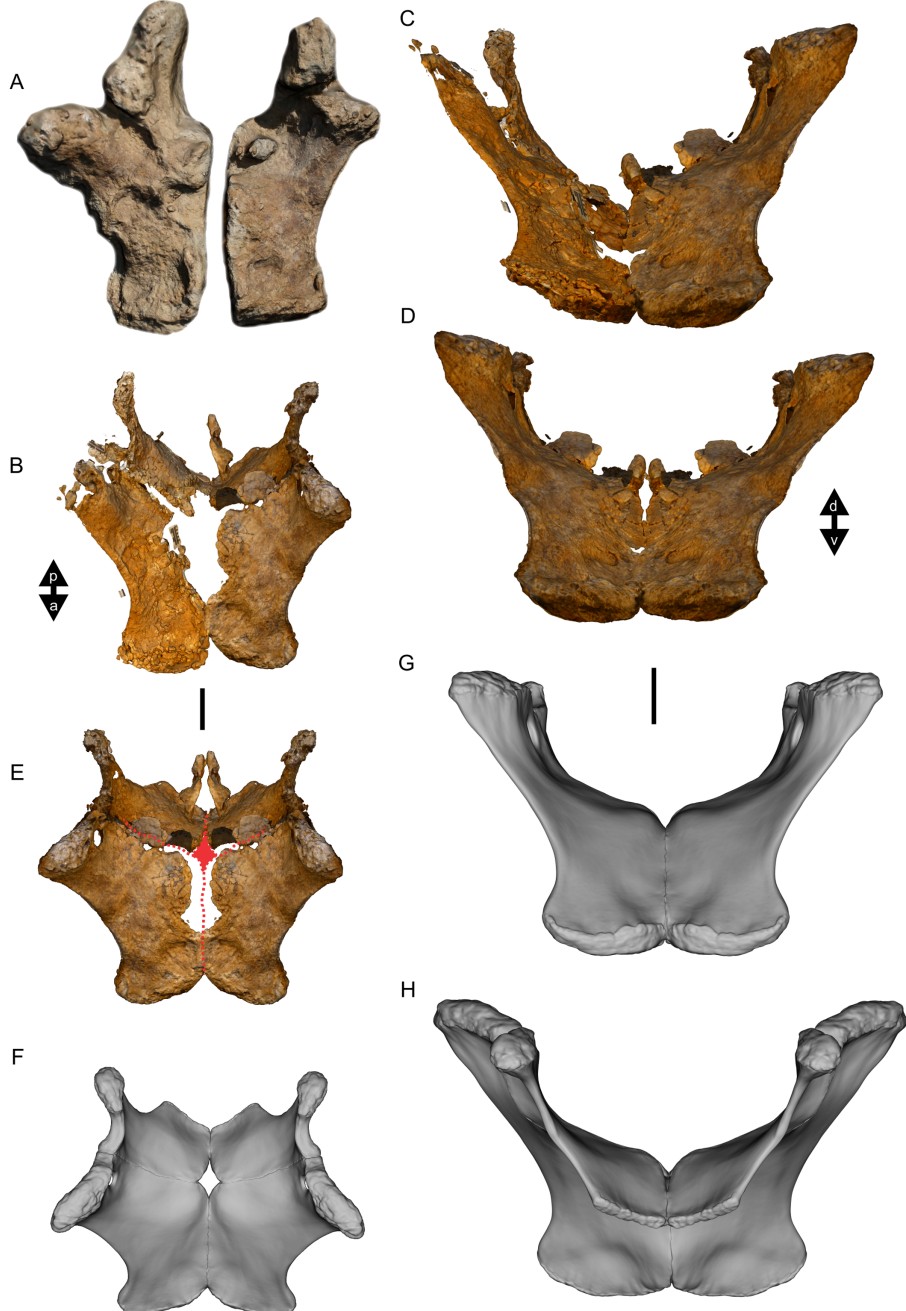

**Figure 21 Pubes and ischia of *Australotitan cooperensis* gen. et sp. nov. (EMF102) continued.** (A) In-field 3-D model of pubes and ischia at EML011. (B & C) After preparation, 3-D model of pubes and ischia reoriented to connect at pubic and ischial symphyses pre-displacement in dorsal (B) and anterior (C) views. (D & E) Mirror of left pubis and ischium (least distorted) to reconstruct overall pelvic floor shape in anterior (D) and dorsal (E) views. Red dotted line indicates estimated extent of pubic and ischial blade contralateral bone with central diamond-shaped gap. (F–H) Digitally restored pubes and ischia in dorsal (F), anterior (G) and posterior (H) views. 3-D image rendering methods used included, natural, A–E and vertex and texture uncoloured in F–H. Arrows indicate direction (a, anterior; p, posterior). Scale bars = 20 cm.

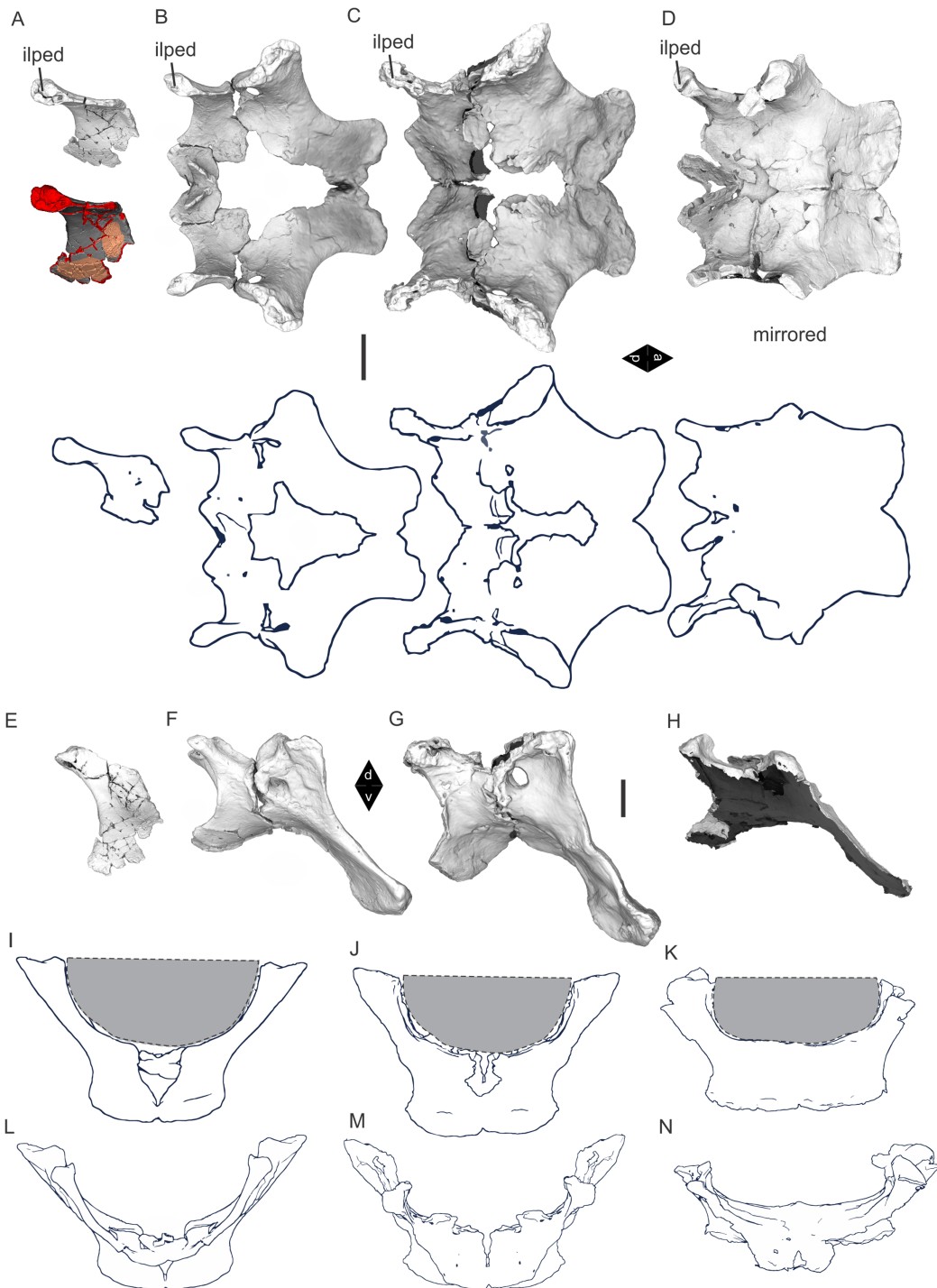

**Figure 22 Comparisons of Winton Formation sauropod pubes and ischia in dorsal, lateral, anterior and posterior views.** (A) & (E) *W. wattsi*, (B, F, I & L) *D. matildae*, (C, G, J & M), *A. cooperensis* gen. et sp. nov. and (D, H, K & N) *S. elliottorum*. 3-D image rendering methods used included, ambient occlusion with radiance scaling, A–H (top); orthogonal outline edge detection, A–D (bottom) and I–N. Dashed line with grey fill indicates estimated ventral pelvic cavity from acetabular opening to pub-o-ischial commissure. Arrows indicate direction (a, anterior; p, posterior). Feature abbreviations: ilped; iliac peduncle. Scale bars = 20 cm.               

**Table 5 Pubes measurements of Winton Formation sauropods.**

| | Preserved | Preserved | Reconstructed |
|---|---|---|---|
| **EMF102 *Australotitan cooperensis*** | Left | Right | |
| Maximum pubis length | 1,262.77 | 1,206.7+ | |
| Maximum proximolateral to distolateral length (b) | 1,118.73 | 1,035.18+ | |
| Maximum length of ischial peduncle | 628.22 | 615.27+ | |
| Maximum anteroposterior acetabular length | 389.58 | n/a | |
| Maximum mediolateral mid-blade distance (a) | 514.98 | 405.26+ | |
| Maximum mediolateral distal-blade length | 513.46 | 492.96+ | |
| Maximum anteroposterior iliac peduncle length (c) | 414.21 | | |
| Maximum mediolateral iliac peduncle width (d) | 158.51 | | |
| Maximum obturator foramen length | 113.87 | 112.41 | |
| Maximum obturator foramen width | 86.43 | 73.28 | |
| Distance between anterior margin of iliac peduncles | | | 1,564.32 |
| (a)/(b) | 0.46 | | |
| (c)/(d) | 2.61 | | |
| **AODF603 *Diamantinasaurus matildae*** | | | |
| Maximum pubis length | 1,056.28 | 1,082.88 | |
| Maximum proximolateral to distolateral length (b) | 942.25 | 957.12 | |
| Maximum length of ischial peduncle | 413.24 | 379.21+ | |
| Maximum anteroposterior acetabular length | 441.29 | 348.15 | |
| Maximum mediolateral mid-blade distance (a) | 386.65 | 370.9 | |
| Maximum mediolateral distal-blade length | 305.24 | 357.74 | |
| Maximum anteroposterior iliac peduncle length (c) | 297.65 | 280.55 | |
| Maximum mediolateral iliac peduncle width (d) | 113.20 | 99.86 | |
| Maximum obturator foramen length | 71.92 | 80.76 | |
| Maximum obturator foramen width | 57.45 | 60.55 | |
| Distance between anterior margin of iliac peduncles | | | 1,219.42 |
| (a)/(b) | 0.41 | 0.39 | |
| (c)/(d) | 2.63 | 2.81 | |
| **AODF660 *Savannasaurus elliottorum*** | | | |
| Maximum pubis length | 894.13+ | 997.18 | |
| Maximum proximolateral to distolateral length (b) | 651.8+ | 802.88 | |
| Maximum length of ischial peduncle | 458.9+ | 366.82+ | |
| Maximum anteroposterior acetabular length | | 209.65 | |
| Maximum mediolateral mid-blade distance (a) | 415.58 | 420.97 | |
| Maximum mediolateral distal-blade length | 409.5 | 407.46 | |
| Maximum obturator foramen length | | 98.02 | |
| Maximum obturator foramen width | | 52.75 | |
| Distance between anterior margin of iliac peduncles | | | 1,083.71+ |
| (a)/(b) | | 0.52 | |

Notes:
All measurements in mm.
+, full length not preserved.

**Table 6  Ischia measurements of Winton Formation sauropods.**

| | Preserved | Preserved | Reconstructed |
|---|---|---|---|
| **EMF102 *Australotitan cooperensis*** | Left | Right | |
| Maximum ischial length | 901.23 | 879.87+ | |
| Maximum proximolateral to distomedial length | 644.46 | 577.35+ | |
| Maximum length of pubic peduncle | 600.37 | 614.43 | |
| Maximum anteroposterior acetabular length (a) | 213.94 | | |
| Maximum anteroposterior mid-blade length (b) | 274.97 | 250.05+ | |
| Maximum dorsoventral (anteroposterior) distal-shaft width (c) | 423.05 | | |
| Minimum dorsoventral (anteroposterior) ischial blade width (d) | 259.15 | | |
| Maximum anteroposterior iliac peduncle length | 227.81 | | |
| Maximum mediolateral iliac peduncle width | 117.49 | | |
| Distance between iliac peduncles (mirrored) | | | 1,171.71 |
| Posterior-most medial projection to posterior-most point on iliac peduncle | 602.5 | | |
| Posterior-most medial projection to anterior-most pubic peduncle | 425.6 | | |
| (a)/(b) | 0.78 | | |
| (c)/(d) | 1.63 | | |
| **AODF603 *Diamantinasaurus matildae*** | | | |
| Maximum ischial length | | 668.7 | |
| Maximum proximolateral to distomedial length | | 558.54 | |
| Maximum length of pubic peduncle | | 366.73+ | |
| Maximum anteroposterior acetabular length (a) | | 182.93 | |
| Maximum anteroposterior mid-blade length (b) | | 207.04 | |
| Maximum dorsoventral (anteroposterior) distal-shaft width (c) | | 381.12 | |
| Minimum dorsoventral (anteroposterior) ischial blade width (d) | | 220.23 | |
| Maximum anteroposterior iliac peduncle length | | 176.69 | |
| Maximum mediolateral iliac peduncle width | | 91.63 | |
| Distance between iliac peduncles | | | 1,002 est. |
| Posterior-most medial projection to posterior-most point on iliac peduncle | | 559.04 | |
| Posterior-most medial projection to anterior-most pubic peduncle | | 372.3 | |
| (a)/(b) | | 0.88 | |
| (c)/(d) | | 1.73 | |
| **QMF7292 *Wintonotitan wattsi*** | | | |
| Maximum ischial length | 776.9+ | | |
| Maximum proximolateral to distomedial length | 643.5+ | | |
| Maximum length of pubic peduncle | 337.6+ | | |
| Maximum anteroposterior acetabular length (a) | 271.3 | | |
| Maximum anteroposterior mid-blade length (b) | 276.6 | | |
| Maximum dorsoventral (anteroposterior) distal-shaft width (c) | 274.1+ (420 est.) | | |
| Minimum dorsoventral (anteroposterior) ischial blade width (d) | 255.23 | | |
| Distance between iliac peduncles | | | 1,065 est. |
| Posterior-most medial projection to posterior-most point on iliac peduncle | 616.92 | | |
| Posterior-most medial projection to anterior-most pubic peduncle | 413.69 | | |

| Table 6 (continued) | | | |
|---|---|---|---|
| | **Preserved** | **Preserved** | **Reconstructed** |
| (a)/(b) | 0.98 | | |
| (c)/(d) | 1.64 est. | | |
| AODF660 *Savannasaurus elliottorum* | | | |
| Maximum ischial length | 578.28+ | 656.08 | |
| Maximum proximolateral to distomedial length | 546.49+ | 601.44 | |
| Maximum length of pubic peduncle | 449.59+ | 375.46+ | |
| Maximum anteroposterior acetabular length (a) | | 198.89 | |
| Maximum anteroposterior mid-blade length (b) | 235.67 | 227.67 | |
| Maximum dorsoventral (anteroposterior) distal-shaft width (c) | 415.22 | 403.22 | |
| Minimum dorsoventral (anteroposterior) ischial blade width (d) | 238.32 | 233.5 | |
| Maximum anteroposterior iliac peduncle length | | 189.59 | |
| Maximum mediolateral iliac peduncle width | | 82.11 | |
| Distance between iliac peduncles | | | 1,045.87+ 1,078 est. |
| Posterior-most medial projection to posterior-most point on iliac peduncle | 611.7 | | |
| Posterior-most medial projection to anterior-most pubic peduncle | 392.47 | | |
| (a)/(b) | | 0.87 | |
| (c)/(d) | 1.74 | 1.73 | |

**Notes:**
All measurements in mm.
+, full length not preserved.
est., estimated.

When compared to other sauropods, the preserved portions of the pelvis are closest in morphology to all three previously described Winton Formation taxa (i.e. *D. matildae, W. wattsi* and *S. elliottorum*). The ischium is preserved for all taxa and warrants specific comparison (Fig. 22). The articular surface of the iliac peduncle is poorly preserved in all taxa; however, the shaft just ventral of this articular surface indicates that all taxa bear a similar tear drop-shaped process that was anteroposteriorly longer than mediolaterally wide. The iliac peduncle is dorsoventrally elongate in *D. matildae, W. wattsi* and *A. cooperensis*, with that of *W. wattsi* being the most elongate. However, this could be a reflection of the significant bone loss around the peduncle in *W. wattsi*, creating an illusion of a more elongate feature (Fig. 22). This feature is fore-shortened in *S. elliottorum* and seems real. However, the iliac peduncles of both the pubis and ischium are somewhat dorsoventrally compressed, suggesting this feature might be due to taphonomic crushing.

The proximal ischial plate is broad anteroposteriorly with a ventromedially curved posterior margin in all taxa, following the curvature of the pubic articulation and co-ossified fusion. Ventromedially the ischial shaft is indistinct from the proximal plate and is best described as a distal plate because it is broad anteroposteriorly along its entire length, and is not differentiated into a posterior process as seen in *O. skarzynskii* (*Borsuk-Bialynicka, 1977*). The distal ischial plate contacts its co-lateral partner medially and was clearly fused to one another in all of the Australian taxa, although broken apart during fossilization in *W. wattsi, D. matildae* and *A. cooperensis*. This fusion is clearly

preserved in *S. elliottorum*, and partially observable in *D. matildae* and *A. cooperensis*. This feature is likely to have been present in *W. wattsi* as well because the distal ischial plate is similarly broad along its entire length and posteriorly foreshortened, with no sign of a completed medial margin. This indicates that bone co-ossification likely occurred with its contralateral pair. Although the medial margin is missing, the thickness of bone suggests a significant area of missing distal ischial plate in *W. wattsi*.

A broad and foreshortened distal ischial plate without a posteriorly projecting blade-like process is not well defined in titanosauriformes; however, *Ma. dixeyi*, *Al. sanjuanensis* and possibly *Uberabatitan ribeiroi* approach this morphology (*Gomani, 2005*; *Silva et al., 2019*; *Tykoski & Fiorillo, 2016*). However, they still retain a posterior process of the ischium shaft blade. *O. skarzynskii* possesses a similar central fusion and broad distal ischial plate, although the distal plate continues posteriorly to form a distinct straight and posteriorly projecting blade-like process (*Borsuk-Bialynicka, 1977*). We therefore consider this combination of features of the ischium a potential synapomorphy for *D. matildae*, *S. elliottorum* and *A. cooperensis*, with the possibility of this feature also uniting the only other Winton Formation taxon, *W. wattsi*, within this group (see Discussion).

When viewed posteriorly, the distal ischial plate retains a gentle medial curvature to meet and fuse medially with its contra-lateral partner in *D. matildae*. However, in *S. elliottorum*, *A. cooperensis* and possibly in *W. wattsi*, the distal plate curves medially to meet its partner, as in *D. matildae*, but before doing this the distal ischial plate curves steeply ventrally creating a posteriorly facing dorsal surface of the distal ischial plate (Fig. 22).

**Femur** (**Figs. 23–26**) (**Table 7**). The femur will be described with the long axis of the shaft vertical and the distal condyles orientated so that they lie flat along a mediolateral horizontal plane. Portions of both the right and left femur are preserved in the holotype. The right femur preserves the diaphysis and distal epiphysis. It is missing the proximal epiphysis and the proximal section of the diaphysis is crushed and distorted, having been pushed downwards from a horizontal position (Fig. 8). This vertical displacement and crushing has distorted the diaphysis from about the midshaft proximally. The crushing is likely due to trampling as discussed above (*Hocknull et al., 2019*) (Fig. 8) and has distorted the longitudinal axis of the diaphysis. The distal half of the diaphysis and distal epiphysis remain undistorted, although the distal medial condyle is damaged with loss of structure on both the anterior and posterior surfaces.

The left femur was recovered on the surface in a large number of fragments and was pieced back together. Surface exposure has removed much of the surface cortical bone; therefore, the femoral head would have been larger and had more of the bulbous femoral head articular surface than what is preserved. Reconstruction of these fragments recovered the proximal epiphysis and the proximal region of the diaphysis to just above the lateral bulge. Both elements preserve overlapping regions of the proximal diaphysis, which allows reconstruction of the femur (Figs. 23 & 24).

In addition to EMF102 (holotype), two other femora, EMF164 and EMF105, are referred to *A. cooperensis* due to significant shared overlap in morphology. EMF164 is

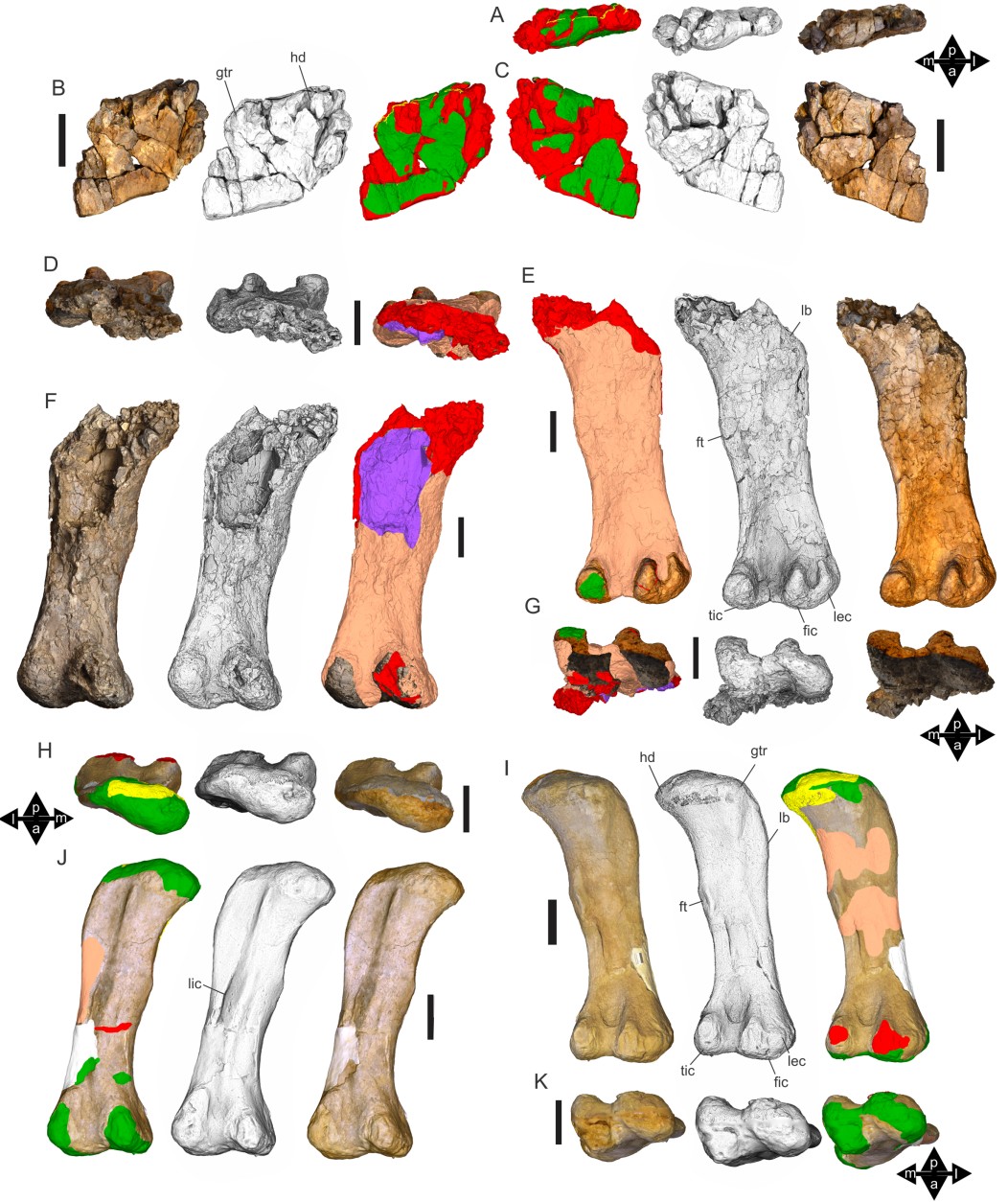

**Figure 23 Femora of *Australotitan cooperensis* gen. et sp. nov. (EMF102) and referred specimen (EMF105).** (A–C) EMF102, left proximal femur head in proximal (A), posterior (B) and anterior (C) views. (D–G) EMF102, right near complete femur in proximal (D), posterior (E), anterior (F) and distal (G) views. (H–K) EMF105, right femur in proximal (H), posterior (I), anterior (J) and distal (K) views. 3-D image rendering methods used included, natural, B, D, F, I, K (left); A, C, E, G, H, J (right); ambient occlusion with radiance scaling, A-K (middle); coloured schematic (see Fig. 8), B, D, F, I, K (right), A, C, E, G, H, J (left). Arrows indicate direction (a, anterior; l, lateral; m, medial; p, posterior). Feature abbreviations: fic, fibular condyle; ft, forth trochanter; gtr, greater trochanter; hd, femoral head; lb, lateral bulge; lec, lateral epicondyle; lic, linea intermuscularis cranialis. Scale bars = 20 cm.

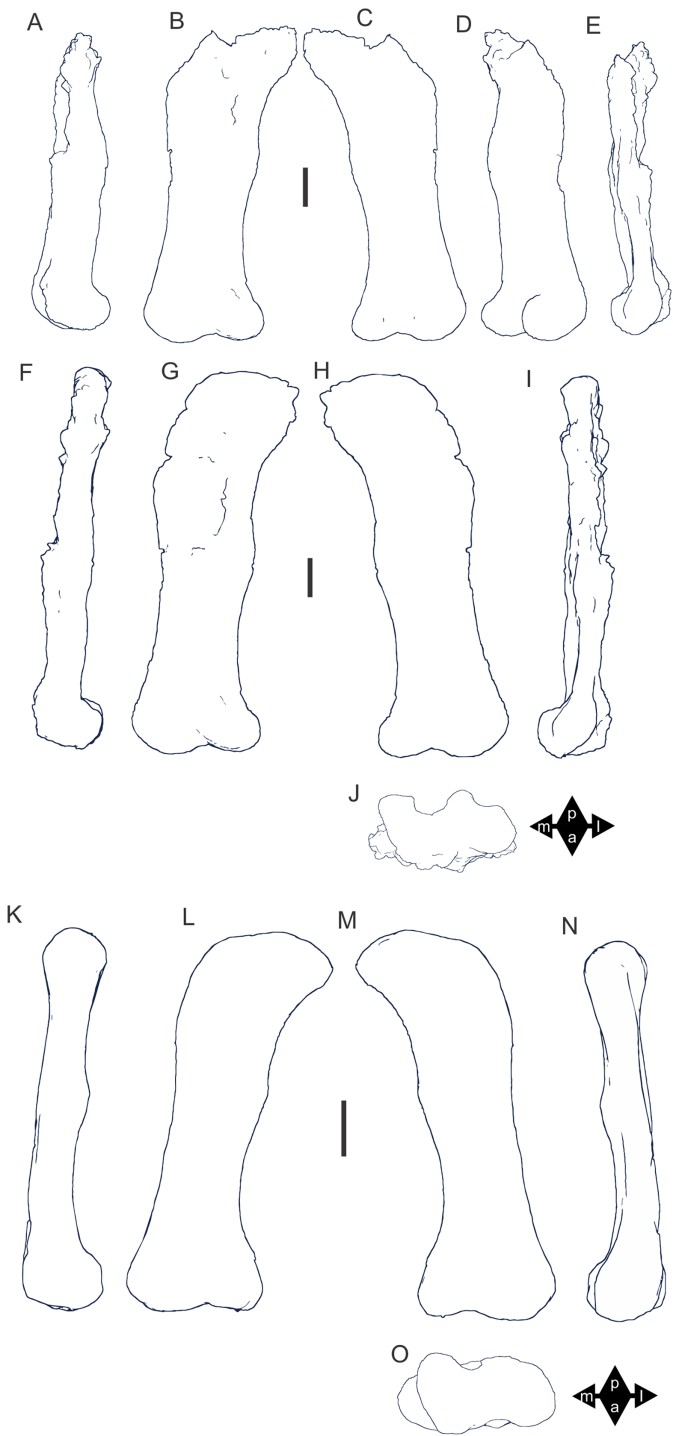

**Figure 24 Femoral orthogonal outlines of *Australotitan cooperensis* gen. et sp. nov. (EMF102) as preserved and reconstructed, and referred femur (EMF105).** (A–E) EMF102 as preserved in medial (A), anterior (B), posterior (C), oblique lateral (D) and lateral (E). (F–J). Reconstructed femur using left and right specimens in medial (F), anterior (G), posterior (H), lateral (I) and distal (J). (K–O) EMF105 as preserved in medial (K), anterior (L), posterior (M), lateral (N) and distal (O). All images scaled to equal minimum mediolateral midshaft width. 3-D image rendering methods used orthogonal outline edge detection. Arrows indicate direction (a, anterior; l, lateral; m, medial; p, posterior). Scale bars = 20 cm.

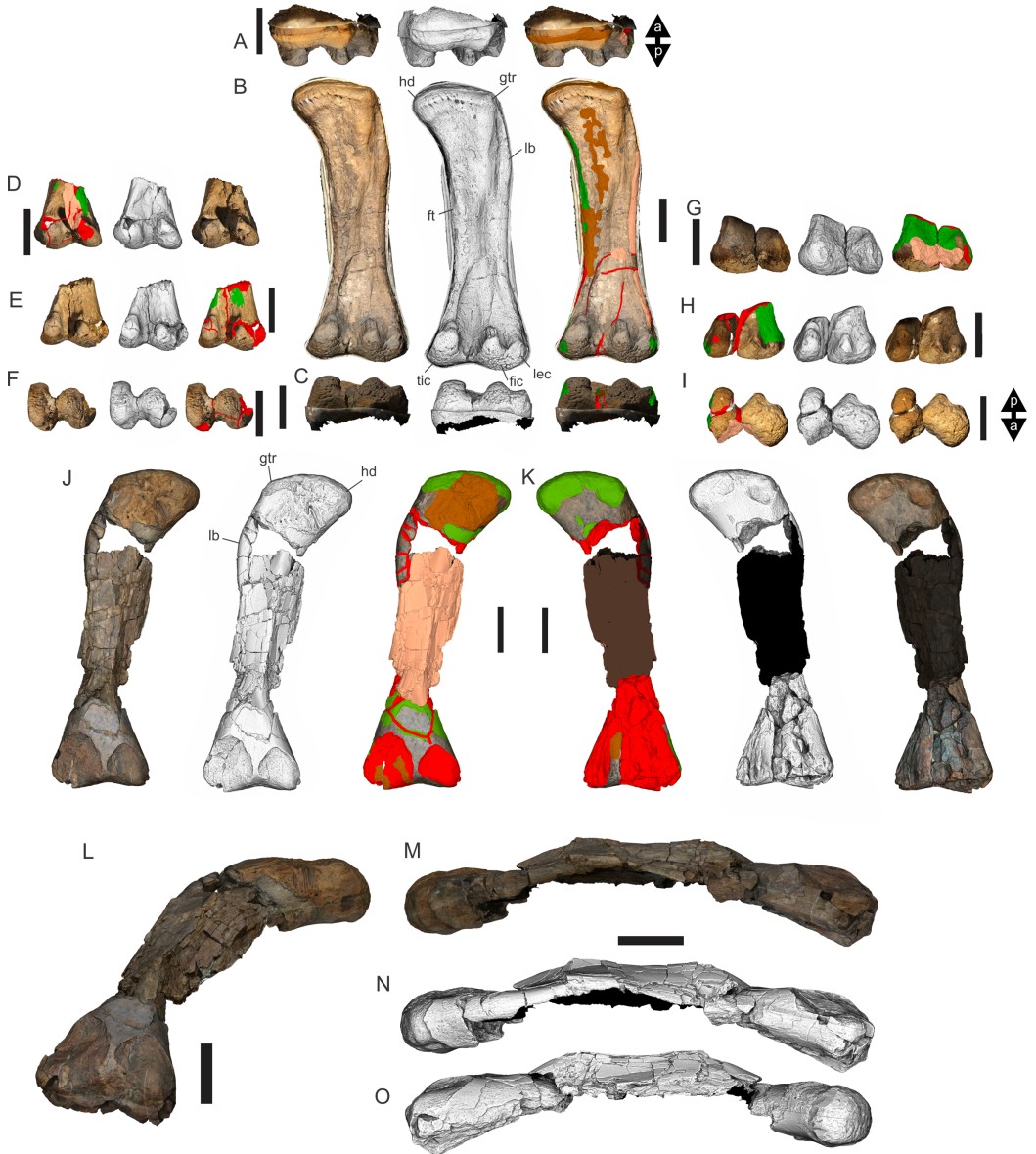

**Figure 25 Northern Winton Formation femora, including the femur of *Diamantinasaurus matildae* holotype (AODF603).** (A–C) AODF603 right femur in proximal (A), posterior (B) and distal (C) views. Anterior face of femur within fiberglass cradle and not available to this study. (D–F) QMF3390 distal right femur in anterior (D), posterior (E) and distal (F) views. (G–I) QMF7291 distal right femur in anterior (G), posterior (H) and distal (I) views. (J–O) QMF43302 partial right femur in anterior (J), posterior (K), oblique medial (L), lateral (M & N) and medial (O) views. Posterior face of femur within fiberglass cradle and not available to this study. 3-D image rendering methods used included, natural, A, B, C, E, F, G, J, (left), D, H, I, K (right), L & M; ambient occlusion with radiance scaling A–K (middle), N & O; coloured schematic (see Fig. 8), A, B, C, E, F, G, J, (right), D, H, I, K (left). Arrows indicate direction (a, anterior; p, posterior). Feature abbreviations: fic, fibular condyle; ft, forth trochanter; gtr, greater trochanter; hd, femoral head; lb, lateral bulge; lec, lateral epicondyle. Scale bars = 20 cm.

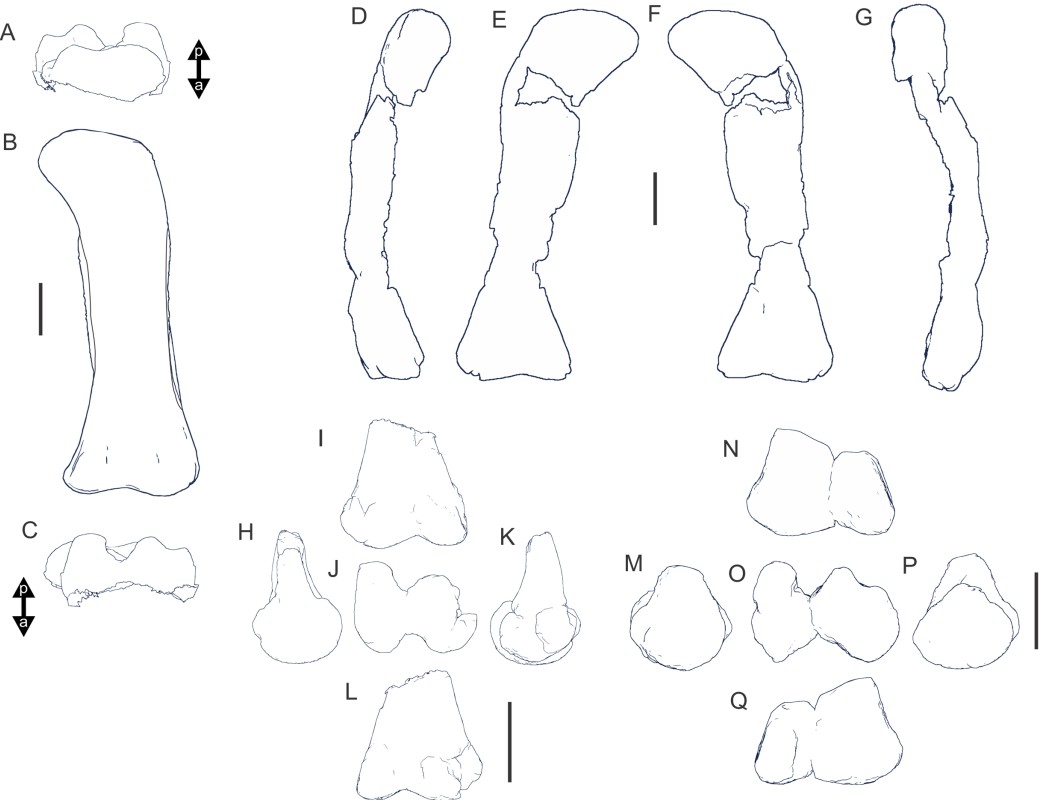

**Figure 26 Northern Winton Formation femora in orthogonal outlines, including the femur of *Diamantinasaurus matildae* holotype (AODF603).** (A–C) AODF603 right femur in proximal (A), posterior (B) and distal (C) views. (D–G) QMF43302 partial right femur in medial (D), anterior (E), posterior (F) and lateral (G) views. (H–L) QMF3390 distal right femur in medial (H), anterior (I), distal (J), lateral (K) and posterior (L). (M–Q) QMF7291 distal right femur in medial (M), anterior (N), distal (O), lateral (P) and posterior (Q). Arrows indicate direction (a, anterior; p, posterior). Scale bars = 20 cm.

highly fragmented but represents a larger femur preserving the proximomedial margin of the proximal epiphysis, along with portions of the lateral bulge, diaphysis, fourth trochanter and medial and lateral condyles. The proximal epiphyseal portion (greater trochanter) has been useful when reconstructing the femur. EMF105 is a complete femur, with some loss of cortical bone around the proximal epiphysis and medial distal condyle. This femur, although smaller than the holotype, provides an accurate independent guide for overall femoral shape when scaled isometrically to the size of EMF102 (*Bonnan, 2004*; *Bonnan, 2007*; *Kilbourne & Makovicky, 2010*). It also provides the best guide to the shape of the dorsomedial portion of the femoral head. The following descriptions of the femur will be based on the holotype but will reference the referred femora where appropriate.

*Anterior view*. The proximal epiphyseal head is rounded and projects proximomedially with preserved articular surface extending across the proximal-most margin from just above the greater trochanter and is assumed to include the missing femoral head.

Table 7 Femur measurements of Winton Formation sauropods.

| | Preserved | Estimate 1 reconstruction | Estimate 2 EMF105 |
|---|---|---|---|
| **EMF102** *Australotitan cooperensis* (holotype) | | | |
| Maximum proximodistal length (b) | | 1,886.02 | 1,888.32 |
| Maximum medial, proximodistal length | 1,587.76+ (right) | 1,854.44 | 1,791.32 |
| Maximum lateral, proximodistal length | 1,582.46+ (right) | 1,833.52 | 1,795.69 |
| Maximum mediolateral width of proximal epiphysis | 525.53+ (left) | 626.93 | 611.85 |
| Maximum anteroposterior length of proximal epiphysis | 161.9+ (left) | 213.16 | 276.88 |
| Maximum mediolateral width across distal condyles | 584.79 | 588.76 | 611.23 |
| Maximum anteroposterior length of distal medial condyle | 357.72+ | 363.64 | 375.12 |
| Maximum anteroposterior length of distal lateral condyle | 316.29+ | 324.63 | 332.69+ |
| Maximum midshaft mediolateral width (a) | <466.03 | 460.09 | 409.63 |
| Minimum midshaft anteroposterior width | 166.21+ | 189.56 | 167.69 |
| Proximal epiphysis circumference | 1,148.61+ | 1,389.47 | 1,427.78 |
| Midshaft circumference | 992.70+ | 1,095.46 | 1,018.37 |
| Minimum diaphyseal circumference | 932.8 | 932.8 | 915.9 |
| Distal condyles circumference | 1,772.86+ | 1,937.46 | 1,757.5+ |
| (a)/(b) | | 0.24 | 0.21 |
| **EMF105** *Australotitan cooperensis* (referred) | | | |
| Maximum proximodistal length (b) | 1,412.32 | 1,412.32 | |
| Maximum medial, proximodistal length | 1,310.42 | 1,310.42 | |
| Maximum lateral, proximodistal length | 1,379.44 | 1,379.44 | |
| Maximum mediolateral width of proximal epiphysis | 469.77 | 469.77 | |
| Maximum anteroposterior length of proximal epiphysis | 219.82+ | 232.41 | |
| Maximum mediolateral width across distal condyles | 470.88 | 470.88 | |
| Maximum anteroposterior length of distal medial condyle | 279.32+ | 320.49 | |
| Maximum anteroposterior length of distal lateral condyle | 251.04+ | 296.94 | |
| Maximum midshaft mediolateral width (a) | 298.99 | 298.99 | |
| Minimum midshaft anteroposterior width | 143.16 | 143.16 | |
| Proximal epiphysis circumference | 1,123.53+ | 1,134.25 | |
| Midshaft circumference | 733.74 | 733.74 | |
| Minimum diaphyseal circumference | 717.91 | 717.91 | |
| Distal condyles circumference | 1,273.13+ | 1,443.93 | |
| (a)/(b) | 0.21 | | |
| **AODF604** *Diamantinasaurus matildae* | | | |
| Maximum proximodistal length (b) | 1,357.87 | | |
| Maximum medial, proximodistal length | 1,297.88 | | |
| Maximum lateral, proximodistal length | 1,336.72 | | |
| Maximum mediolateral width of proximal epiphysis | 412.5 | | |
| Maximum anteroposterior length of proximal epiphysis | 187.42 | | |
| Maximum mediolateral width across distal condyles | 488.57 | | |
| Maximum anteroposterior length of distal medial condyle | 255.32 | | |

(Continued)

| Table 7 (continued) | | | |
|---|---|---|---|
| | Preserved | Estimate 1 reconstruction | Estimate 2 EMF105 |
| Maximum anteroposterior length of distal lateral condyle | 235.43 | | |
| Maximum midshaft mediolateral width (a) | 274.21 | | |
| Minimum midshaft anteroposterior width | 104.54 | | |
| Proximal epiphysis circumference | 902.19 | | |
| Midshaft circumference | 661.92 | | |
| Distal condyles circumference | 1,366.26 | | |
| (a)/(b) | 0.20 | | |
| QMF43302 ?Wintontitan wattsi | | | |
| Maximum proximodistal length (b) | 1,505.68 | | |
| Maximum medial, proximodistal length | 1,430.59+ | | |
| Maximum lateral, proximodistal length | 1,438.89+ | | |
| Maximum mediolateral width of proximal epiphysis | 388.78+ | | |
| Maximum anteroposterior length of proximal epiphysis | 188.98+ | | |
| Maximum mediolateral width across distal condyles | 436.86 | | |
| Maximum anteroposterior length of distal medial condyle | 202.89+ | | |
| Maximum anteroposterior length of distal lateral condyle | 161.68+ | | |

**Notes:**
All measurements in mm.
+, full length not preserved.
<, less than

The fourth trochanter is positioned slightly more proximally than the medial margin of the femoral head. The lateral margin of the femur is shallowly sigmoidal in overall outline shape, made up of the abductor crest (lateral bulge) that curves laterally in a shallow convex outline, distally from the greater trochanter, encompassing approximately a third of the proximal length of the entire lateral margin of the diaphysis. Distal of this, the lateral margin of the diaphysis then curves medially in a shallow concave outline along the remaining two thirds of the shaft where it meets the lateral epicondyle. The medial margin curves laterally in a shallow concave outline from the medial margin of the femoral head position to the fourth trochanter and then curves laterally again in another shallow concave outline from the distal margin of the fourth trochanter to the tibial (medial) condyle.

The distal condylar region is mediolaterally wide with an anteroposteriorly narrow distal epiphysis with the lateral condyle mediolaterally broader than the medial condyle. The articular surface of both condyles extends onto the anterior face of the diaphysis and both condylar articular surfaces on the anterior face are dorsolateral to ventromedially directed, the medial condyle more so than the lateral condyle.

*Posterior view.* A low rounded ridge (lesser trochanter + trochanteric shelf) runs from the greater trochanter along to the lateral bulge and merges with the diaphysis approximately 1/3 the length of the shaft. The fourth trochanter is best visible in posterior view and is proximodistally ovoid in shape and positioned on the posteromedial face of the

diaphysis. The distal end of the diaphysis expands mediolaterally and houses a shallow broad fossa proximal to the distal epiphysis. The distal articular region is divided into two regions, the tibial (medial) condyle and the fibular (lateral) condyle, which includes the lateral epicondyle. The posterior origin of the fibular condyle and lateral epicondyle extends further proximally on the posterior face than the tibial condyle. The fibular condyle and lateral epicondyle are divided by a distinct and deep fossa. The lateral margin of the lateral epicondyle expands from the main articular surface creating a small shallow fossa on the distolateral corner. The tibial and fibular condyles are divided by a deep and wide intercondylar fossa.

*Proximal view*. Although poorly preserved, the femoral head is expanded anteroposteriorly and rounded medially. The greater trochanter is constricted anteroposteriorly with a mediolaterally tapered articular region. A shallow 'D'-shaped transverse cross-sectional outlines the proximal diaphysis, being broad mediolaterally and very narrow anteroposteriorly. The midshaft transverse cross-section outline is anteroposteriorly deeper forming a more distinct 'D'-shape.

*Distal view*. The long axes of the tibial and fibular condyles in distal view are oriented anterolaterally to posteromedially. The tibial (medial) condyle is anteroposteriorly longer than the fibular (lateral) condyle. The crural extensor fossa on the anterior side of the distal epiphysis is broad and similarly as deep to the intercondylar fossa of the posterior side. The anterolateral to posteromedial orientation of the condyles is similar to the distal condyles described for *Daxiatitan binglingi* (*You et al., 2008*), *Dr. schrani* (*Ullmann & Lacovara, 2016*), *L. astibiae* (*Díaz, Suberbiola & Sanz, 2013*) and cf. *L. astibiae* (*Vila et al., 2012*). In *Da. binglingi*, a combination of this feature with dorsolateral bevelling of the distal condyles was considered both unique features of this taxon (*You et al., 2008*). This feature was considered to be one of a number of features that could identify femora to *L. astibiae* (*Vila et al., 2012*). However, in *Dr. schrani* (*Ullmann & Lacovara, 2016*) the medially oriented distal condyles were considered to be oriented in this plane due to taphonomic distortion through lithostatic compression. Therefore, in some taxa this seems to be a real feature, whilst in others it is taphonomic. The anterolateral to posteromedially directed condyles in *A. cooperensis* are unlikely to be taphonomic, although there has been loss of surface bone to the condyles indicating some damage but crushing is restricted to the proximal half of the holotype femur. The same condylar feature is observed in the referred femur EMF112, which has not been crushed.

When comparing the distal condyles of specimens referred to *A. cooperensis* with other femora from the Winton Formation there are clear differences in distal epiphyseal shape (Figs. 23–26). Other than the considerable larger size, the femur of *A. cooperensis* also differs from the femur of *D. matildae*, the only described Winton Formation taxon to preserve a femur, in a number of ways. These differences are also observed when comparing several additional isolated femoral elements from the Winton Formation not currently assigned to a taxon (Figs. 25 & 26), and include: (1) A more proximomedially directed femoral head; a mediolaterally broader and anteroposteriorly narrower diaphysis

along the entire length; (2) A relatively larger and more posteriorly positioned fourth trochanter; (3) a less sigmoidal lateral margin and more convex medial margin; and (4) Anterolateral to posteromedially oriented distal condyles (in distal view). These features not only differentiate the two taxa possessing femora, but also differentiate the southern-central from the northern Winton Formation femoral specimens. Therefore, the femur may be of taxonomic value when differentiating taxa between regions. This also suggests closer morphological similarities to those taxa found within a particular region, relative to between regions. These differences do not seem to relate to overall element size because the differences are seen in specimens from Eromanga and Winton that are very different in size (Figs. 24 & 26).

Overall, the femur of *A. cooperensis* is similar to titanosauriform sauropods and more derived titanosaurians. Comparing the outline shape of the anterior and posterior views across titanosauriform sauropods similarities in overall shape are found in *Dr. schrani* (*Ullmann & Lacovara, 2016*), *Traukutitan eocaudata* (*González Riga et al., 2019*), *L. astibiae* (*Díaz, Suberbiola & Sanz, 2013*), *Aegyptosaurus baharijensis* (*Stromer, 1932*) and *Ampelosaurus atacis* (*Le Loeuff, 2005*). These similarities reflect a broad femoral shaft relative to proximal and distal condylar breadths, along with a long shallowly curved lateral bulge and less bulbous proximal femoral head. The femora are also narrow anteroposteriorly along the diaphyseal length, but possess expanded proximal and distal epiphyseal regions.

The northern Winton Formation femora, including *D. matildae*, all have narrower and deeper diaphyseal shafts, more bulbous proximal femoral heads, anteroposteriorly thicker lesser trochanter, and anteroposteriorly rotund distal epiphyses (Figs. 25 & 26). The femoral shaft is relatively narrower and dorsoventrally straightened in the northern Winton Formation sauropods compared to the southern-central specimens. Such variation in femoral shaft morphology is present in several titanosaurians, ranging from stout and robust diaphyses in taxa like *N. robustus* (*Otero, 2010*), *Sa. loricatus, Ep. sciuttoi* (*Martínez et al., 2004*) and *Bonatitan reigi* (*González Riga et al., 2019*), to straight and deep diaphyses in taxa like *P. mayorum* (*Carballido et al., 2017*), to anteroposteriorly compressed and mediolaterally broad, sinuous diaphyses in taxa like *A. cooperensis*, *L. astibiae* (*Díaz, 2013*) and *Dr. schrani* (*Ullmann & Lacovara, 2016*).

### Referred Specimens

**EMF164.** *Axial remains.* The type specimen for *A. cooperensis* does not possess associated vertebrae; however, the referred specimen EMF164 from EML010 includes isolated pieces of presacral vertebrae preserving distinctly camellate somphospondylous internal centrum bone. The internal cavities filled with matrix are large and indicate derived somphospondylous architecture similar to that seen in all other Cretaceous-aged sauropods from Australia. The camellate bone structure is very thin, reticulated, thin bone struts held within a mudstone matrix, approximating the same degree of camellate structuring seen in the holotype dorsal vertebrae of *Austrosaurus mckillopi, D. matildae, W. wattsi* and *S. elliottorum*. The thickness of the trabeculae and the size of the vacuities observed in the isolated pieces of EMF164 are larger than those from these previously
described taxa, thus indicating that the vertebrae were much larger in overall size. Large pieces of plank-like rib shafts are also present, although no proximal rib articular ends have been identified.

*Appendicular remains.* Identifiable pieces of ulna include sections of diaphysis and a fragment preserving a thick ridge that represents the prominent interosseous ridge of the radius, similar to that present in holotype (EMF102). These ulna fragments are too poorly preserved to provide additional information that the holotype provides; however, the thickness of the cortical bone seen in cross-section of EMF164 when compared to that of the holotype (EMF102) indicates that EMF164 was a larger individual.

The larger size of EMF164 is best represented by the fragments of a right femur. A large number of fragments represent diaphyseal pieces of the femur that are clearly anteroposteriorly narrow indicating a broad, but narrow diaphysis for the femur, similar to that seen in EMF102. However, these pieces have much thicker cortical bone in cross-sectional comparison. As with the ulna pieces, this thickness of cortical bone indicates an individual of larger size than that of EMF102.

The elements of the EMF164 femur do not provide any additional details of the femur from a comparative point of view, other than its larger size. Estimating the size of this larger femur provides some additional information in regards to the overall variation in the size of these elements and estimates of body-size in this taxon. Therefore, we have undertaken three different estimations of the femur length of EMF164 and will report the average and range.

We directly matched the largest fragments of the femur of EMF164 to the femora of the *A. cooperensis* holotype, EMF102 and referred EMF105. We first did this by sight and then digitally by aligning and scaling the 3-D surface meshes of the smaller femora (EMF102 & 105) to match the size of the combined 3-D surface meshes of the EMF164 femoral pieces. This was achieved in Meshlab (*Cignoni et al., 2008*) and Cloud Compare (*Girardeau-Montaut, 2016*) by point picking, rotation/translation, then isometrically scaling the 3-D surface mesh of EMF102 and EMF105 to match the size and position of the EMF164 pieces.

The resulting isometrically scaled reconstruction returned maximum total lengths of the femur when scaled to EMF102: maximum medial length of 2,117 mm; maximum central length of 2,134 mm; maximum lateral length of 2,160 mm. The reconstructed surface mesh of EMF102 does not include the proximal-most femoral head articular surface because this is missing portions of the proximal-most cortical bone, therefore, these estimates could be considered underestimations.

Second, we undertook the same process, but this time we matched the 3-D restored femur that was based on the surface mesh of EMF102. Therefore, when scaling the reconstructed model isometrically, this universal scaling was automatically applied to the associated 3-D modelled femur. The resulting isometrically scaled model returned total lengths of the femur: maximum medial length of 2125 mm; maximum central length of 2,176 mm and; maximum lateral length of 2,140 mm. The modelled surface mesh of EMF102 includes estimations of the proximal-most femoral head articular surface by
continuation of the surface bone shape, therefore, these estimates could be considered accurate.

Finally, we used the 3-D surface mesh created of the referred femur, EMF105, and aligned and scaled this mesh to match the surface meshes of EMF164 pieces. The resulting isometrically scaled model returned total lengths of the femur: maximum medial length of 2,133 mm; maximum central length of 2,187 mm and; maximum lateral length of 2,147 mm. EMF105 is a complete femur missing some of the proximal and distal condyles making this estimate likely a slight underestimate.

Together, taking all nine measurements we arrive at an average length of 2,146 mm with a range of 2,117–2,187 mm. Considered together, this provides an estimated length of the EMF164 femur of approximately 2,150 mm in length, which is approximately 200 mm longer than the reconstructed femur of the holotype (EMF102).

**EMF105 (Fig. 23 & 24, Table 7).** EMF105 is a complete right femur, measuring 1,412 mm in maximum proximodistal length. The femur conforms closely to the overall morphology of the holotype femora EMF102; however, it is better preserved and includes a well-preserved proximal femoral head. Post-depositional scouring of the distal condyles has truncated them in the anteroposterior plane. Excavator damage during removal of overburden has occurred to the distal diaphysis shaft with loss of preserved bone in a triangular wedge-shape.

**EMF165 (Fig. 27).** EMF165 is a portion of a distal humerus preserving a shallow and broad olecranon (= anconeal) fossa and a rounded anterior face. It is missing much of the distal epiphyseal articular surface, although it is broad relative to the diaphysis to a similar extent to that seen in EMF102. In distal view, the tri-lobate articular outline can be discerned, although the anterior and posterior extremities of the condyles are missing. Although not preserving considerable detail, the proportions of this distal humerus are similar to that of the holotype and not that of *D. matildae*, the only other Winton Formation sauropod to preserve a distal end of the humerus.

**Other titanosaurian specimens**

Currently, several titanosaurian specimens cannot as yet be directly referred to *A. cooperensis* due to their incompleteness or current state of preparation. These specimens are known from the northern and southern Plevna Downs sites and include isolated, associated and articulated remains.

Based on comparisons of these preserved elements with those from northern Winton Formation taxa, they share general features, but none possess features that definitively ally them with those taxa (i.e., *D. matildae*, *W. wattsi* or *S. elliottorum*). Therefore, we applied a conservative approach of provisionally allocating them to the local taxon, *A. cooperensis*, until sufficient overlap is found in skeletal remains to constitute a fully diagnostic allocation.

EMF100, from EML01 is a small, poorly preserved ulna, missing the majority of the proximal and distal ends (Fig. 28). However, comparison of the midshaft diaphyseal cross-section and proximal and distal shape comparisons are possible between EMF100,

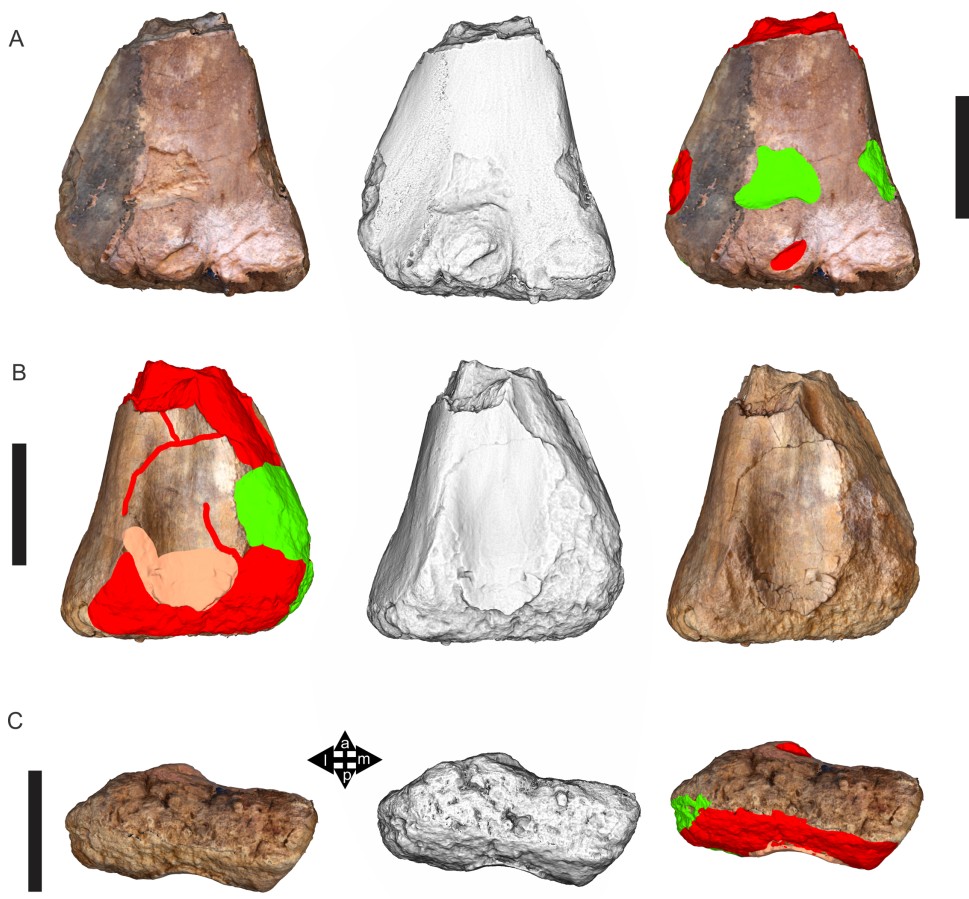

**Figure 27 EMF165, a distal humerus referred to *Australotitan cooperensis* gen. et sp. nov.**
(A) Anterior view. (B) Posterior view. (C) Distal view. 3-D image rendering methods used included, natural, A & C (left), B (right); ambient occlusion with radiance scaling A–C (middle); coloured schematic (see Fig. 8) A & C (right), B (left). Arrows indicate direction (a, anterior; l, lateral; m, medial; p, posterior). Scale bars = 20 cm.

*A. cooperensis*, *D. matildae* and *W. wattsi*. EMF100 is mediolaterally compressed as seen in *A. cooperensis* and not in *D. matildae* or *W. wattsi*. Furthermore, the shape of the shaft in distal and oblique-distal views is closer to *A. cooperensis* than it is to *D. matildae* or *W. wattsi*. In proximal view, the anteromedial process is proportionately more elongate relative to the proximolateral process, albeit missing the proximal portion of the process. However, by projecting the anteromedial and anterolateral processes proximally, the relative expansion of these processes is closer to that of *A. cooperensis* than it is to *D. matildae* or *W. wattsi*.

EMF106 occurs at EML010 and is a collection of small sauropod remains found with EMF164. Identifiable remains of EMF106 include a metapodial articular end and pieces of mid caudal centra. A portion of a caudal centrum is amphicoelous with dense non-pneumatic cancellous bone (Fig. 29G & 29H).

EMF103 occurs at EML011b and is a scattered series of cervical and dorsal vertebrae with a poorly preserved distal femur and isolated dental remains. Based on overall size

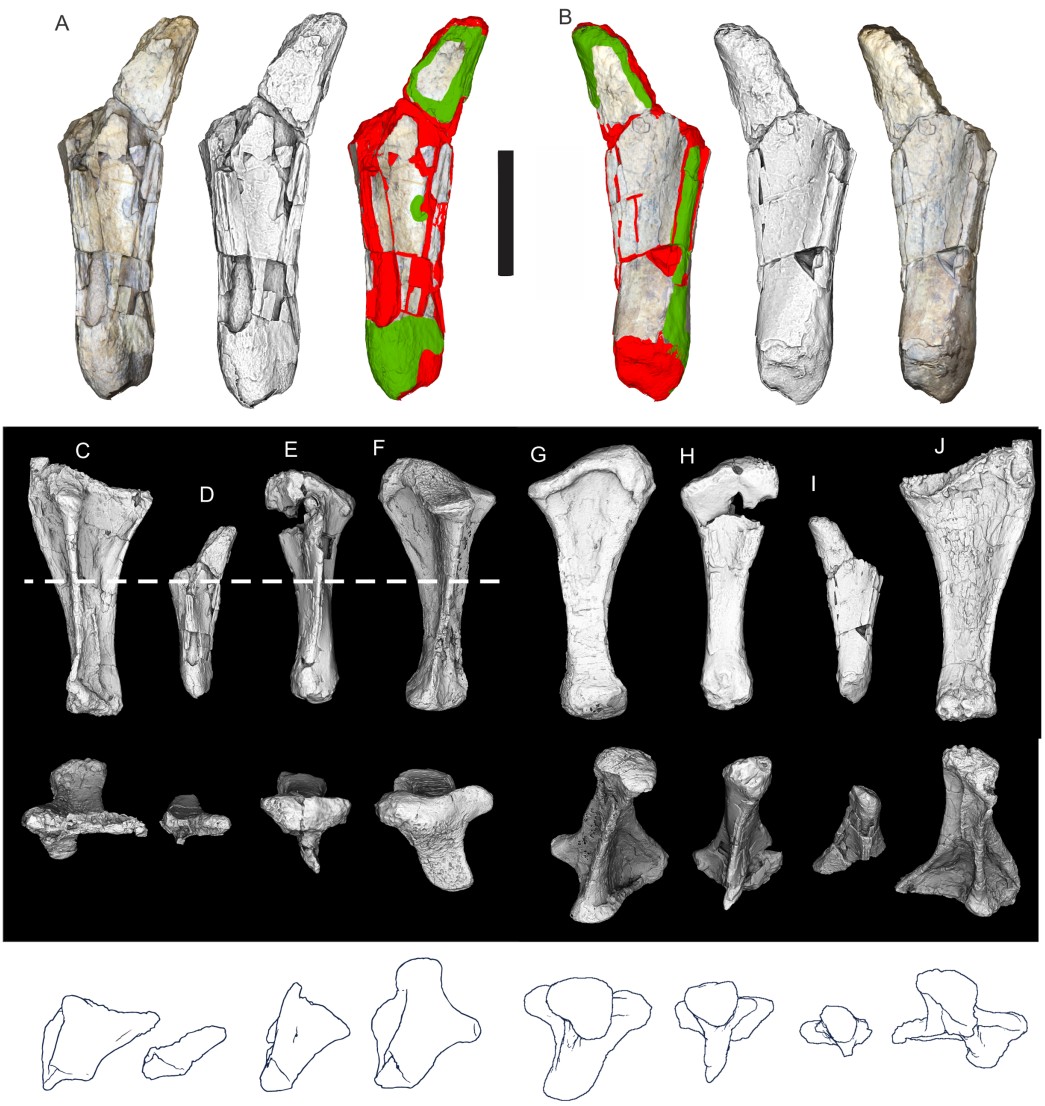

**Figure 28 EMF100 (EML01), a small partial ulna with similar morphological features to *Australotitan cooperensis* gen. et sp. nov.** (A &B) Ulna in mediolateral (A) and medial (B) views. (C–J) Comparisons between EMF100 (D) with *A. cooperensis* gen. et sp. nov. (C & J), *W. wattsi* (E & H) and *D. matildae* (F & J), scaled to minimum midshaft width. Mediolateral shape (top left), medial shape (top right), proximal shape (middle left), distal shape (middle right), midshaft cross-sectional shape (bottom left, cross-section position indicated by dotted line in top) and distal margin outline (bottom right). 3-D image rendering methods used included, natural, A (left), B (right); ambient occlusion with radiance scaling A & B (middle), C–J top and middle row; coloured schematic (see Fig. 8) A (right), B (left); and orthogonal outline edge detect (bottom row). Scale bars = 20 cm.

similarities between the cervical and dorsal vertebrae, along with the femur, it is likely that this specimen represents a single individual. However, the distribution of the skeletal elements and the post-depositional scouring and trampling makes comparing this skeleton with other individuals difficult. The femur does overlap as an appendicular element with EMF102. However, the element is not well enough preserved to ally it, or separate it, from *A. cooperensis*. The cervical and dorsal vertebrae are well preserved on the surfaces

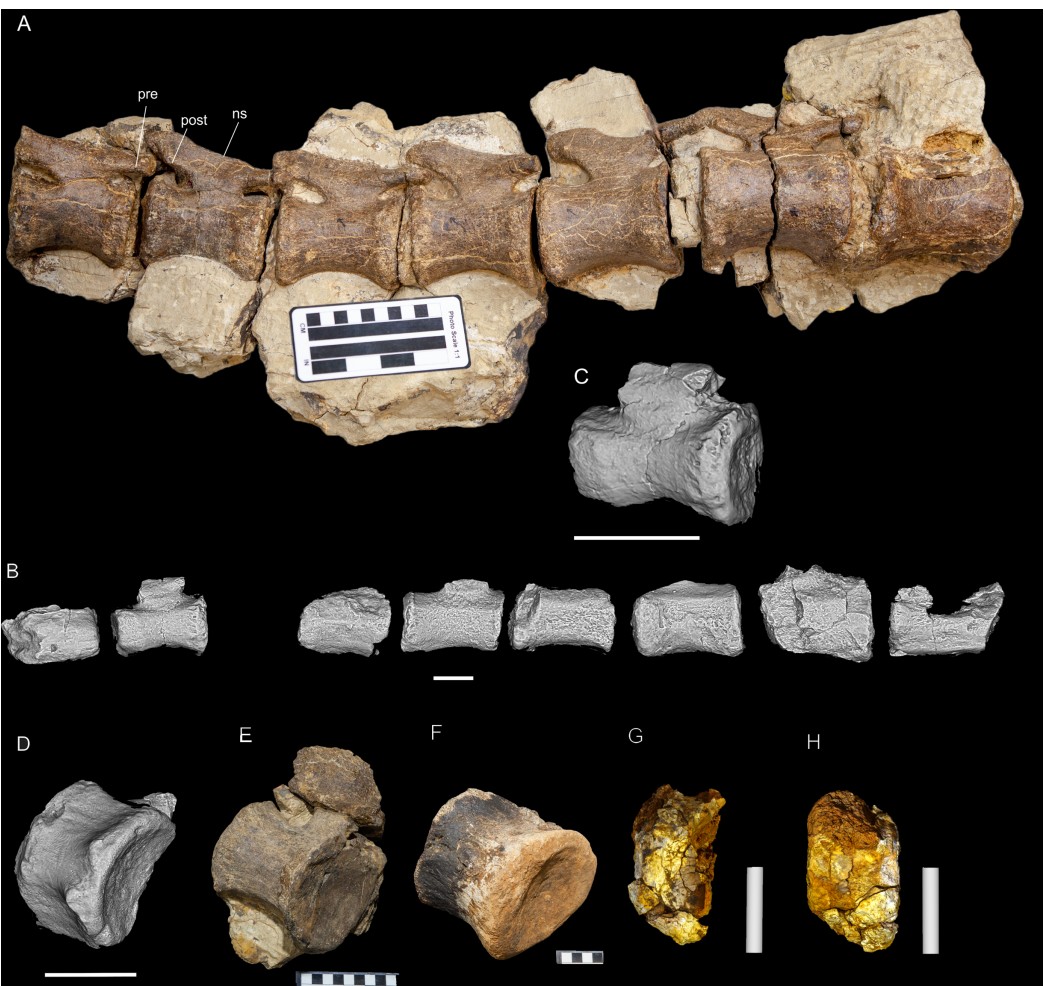

**Figure 29 Sauropod caudal vertebrae from southern-central Winton Formation sites compared to *Wintonotitan wattsi* (QMF7292).** (A) EMF109, series of articulated distal caudal vertebrae as part of an articulated skeleton (see Fig. 7K), right lateral view. (B) QMF7292, *Wintonotitan wattsi* holotype distal caudal vertebral series, right lateral view. (C) Closeup of the most complete distal caudal in the series for *W. wattsi*, in oblique craniolateral view. (D) QMF7292, *W. wattsi* holotype middle caudal vertebra, in oblique cranioventral view. (E) EMF109 (EML012) middle caudal vertebra, in oblique cranioventral view. (F) EMF171 (EML028) middle caudal vertebra, in oblique cranioventral view. (G &H) Partial proximal distal caudal, EMF106, from EML010 in anterior (G) and lateral (H) views. Abbreviations: ns, neural spine; post, postzygopophysis; pre, prezygopophysis. Scale bars = 10 cm.

that faced downward in the site. The upward projecting faces have been scoured and trampled which has dislocated and deformed the positions, and possible interpretations, of the vertebral laminae. Therefore, this precludes meaningful comparisons to the other Winton Formation taxa preserving cervical and dorsal vertebral laminae, until we can retrodeform and model the original positions of these features.

EMF166 is an isolated metacarpal found with EMF165 and EMF105. The metacarpal is relatively small in comparison to what would be expected to be from the individual femur (EMF105) or the humerus (EMF165). Based on comparisons with the metacarpals of *D. matildae*, *W. wattsi* and *S. elliottorum*, EMF166 is a metacarpal IV. The proximal and

distal ends are rounded through pre-depositional abrasion, marked by a thick layer of plant debris covering the bone prior to preparation. The proximal end describes a roughly tear-drop or rounded triangular shape with the broadest rounded margin being external and the narrowest margin constricted internally. There are remnants of distinct internal condylar processes that have been rounded off through abrasion. The distal external margin is rounded with no distinct indication of distal articular surfaces on the external face suggestive of phalanges. However, the lack of these features could be preservational. In external view, the metacarpal differs from the northern Winton Formation taxa by being more elongate without the proximally and distally expanded and robust epiphyses seen in *D. matildae*, *W. wattsi* and *S. elliottorum*.

EMF109 (EML012) (Figs. 6K & 29A) is an associated and articulated skeleton preserved within a massive siltstone concretion located 65 m to the southwest of EML013. Based on what skeletal elements were observable in the concretion this specimen preserves much of the torso and tail of the sauropod. The articulated caudal vertebrae were evident in the site, delineated by the concretion itself. However, the dorsal vertebrae, ribs and appendicular elements are mostly obscured by concretion. Until this concretion has been prepared, direct referral of it to a described taxon is precluded; however, the distal mid and distal caudal vertebrae have been prepared to a point that allows some initial comparison with the distal caudal vertebrae known from *W. wattsi* (Fig. 2B).

Of the two known occurrences of distal caudal vertebrae known from the northern Winton Formation both are incipiently bi-convex as originally described (*Hocknull et al., 2009*), possessing articular ends but do not approach the true bi-convexity seen in *Rinconsaurus* (*Calvo & González Riga, 2003*). This feature is now considered to be a local autapomorphy for *W. wattsi* because it is known across several titanosauriforms (*D'Emic, 2012*; *Poropat et al., 2015a*). Having said this, neither *D. matildae* or *S. elliottorum* have associated distal caudal vertebrae preserved, therefore, at this stage, the utility of this feature is equivocal and only useful to exclude *W. wattsi* from a possible candidate taxon for the southern-central Winton Formation specimen.

The distal caudal vertebrae of EMF109 are not incipiently bi-convex, instead being amphicoelous to amphiplatyan, possessing similar morphology to all other anterior and middle caudal vertebrae found across sites in both the northern and southern-central Winton Formation (see Discussion). Therefore, we can exclude *W. wattsi*, as a candidate taxon, however due to the ubiquitous nature of amphicoelous caudal vertebrae of sauropods in the Winton Formation we cannot exclude any of the other three described taxa. Based on what is indicated from the specimen as currently visible, EMF109 will provide significant data to understand the anatomy of these sauropods, being the most complete southern Winton Formation specimen.

## DISCUSSION

### Comparison with other Winton Formation sauropod taxa

*Australotitan cooperensis* can be differentiated from the three semi-contemporaneous northern Winton Formation sauropods, *Diamantinasaurus matildae*, *Wintonotitan wattsi*

and *Savannasaurus elliottorum*, in the following ways: *A. cooperensis* is larger than all the three taxa in the scapula, humerus, ulna, femur and pubis (Tables 2–7). The scapula differs from *D. matildae* and *W. wattsi* by possessing sub-parallel dorsal and ventral margins of the scapular blade; not possessing a medial scapular blade tuberosity and; not possessing a distinct lateral mid-ridge of the scapular blade. Instead, this ridge occurs along the ventral margin (Figs. 10 & 28). The humerus differs from *D. matildae* by possessing a distinct tri-lobate distal articular epiphysis and a deltopectoral crest that terminates more distally (Figs. 15, 16 & 30). Neither *S. elliottorum* nor *W. wattsi* preserve the proximal or distal articular ends so are not directly comparable. The humerus further differs from both *W. wattsi* and *S. elliottorum* by the later taxa bearing an ovo-rectangular midshaft cross-sectional shape (Fig. 16). The ulna differs from *D. matildae* and *W. wattsi* by possessing a relatively longer proximal anteromedial process and a distinct interosseous ridge in the radial fossa (Figs. 18, 19 & 30).

Pubes are known from *D. matildae*, *S. elliottorum* and *A. cooperensis*, but are unknown in *W. wattsi*. *A. cooperensis* differs from *D. matildae* by being larger; possessing dorsoventrally thinner pubic blades; possessing an obturator foramen closer to the proximal margin; and a slightly more mediolaterally expanded distal margin (Figs. 22 & 28). The pubes of *A. cooperensis* differ from *S. elliottorum* by being larger, more ventrally directed; not possessing a lateral proximodistal mid-ridge (autapomorphy of *S. elliottorum*); and by possessing an obturator foramen that is dorsoventrally oblong instead of dorsoventrally compressed as in *S. elliottorum* (Figs. 22 & 28). The latter feature may be due to taphonomic distortion in *S. elliottorum* where the pubis has possibly been compressed in the dorsoventral plane, but if so, the obturator foramen would then be much larger in *S. elliottorum* relative to *A. cooperensis* and *D. matildae*.

The ischia of *D. matildae*, *W. wattsi*, *S. elliottorum* and *A. cooperensis* are known and all are near complete, making this element one of the best directly comparable elements between all four taxa. All taxa are similar in overall morphology, possessing a distinct 'tear-drop' shaped iliac peduncle in dorsal view; concave acetabular articular region; long ventromedially curved pubic articular surface; and similarly ventromedially curved posterior puboischial blade margin. The ischial blade expands anteroposteriorly as it curves ventrally, then connects with its contralateral element in *D. matildae*, *S. elliottorum* and *A. cooperensis*.

The distomedial margin of the ischium in *W. wattsi* is missing and precludes a definitive mid-line connection between the contralateral ischia. However, based on the close similarity in morphology and the curvature of this element with the other taxa, it is very likely that the ischia extended to contact its contralateral at the midline (Figs. 22 & 30).

In dorsal view, the posterior-most margin of each ischial blade occurs at near to two-thirds the dorsoventral length of the posterior blade margin. This produces a double-pointed posterior margin of the ischia in dorsal view with a 'v'-shaped embayment at the posteromedial margin of the ischia. This embayment is shallowest in *A. cooperensis* and steepest in *S. elliottorum*, with *D. matildae* intermediate. Although this margin is not

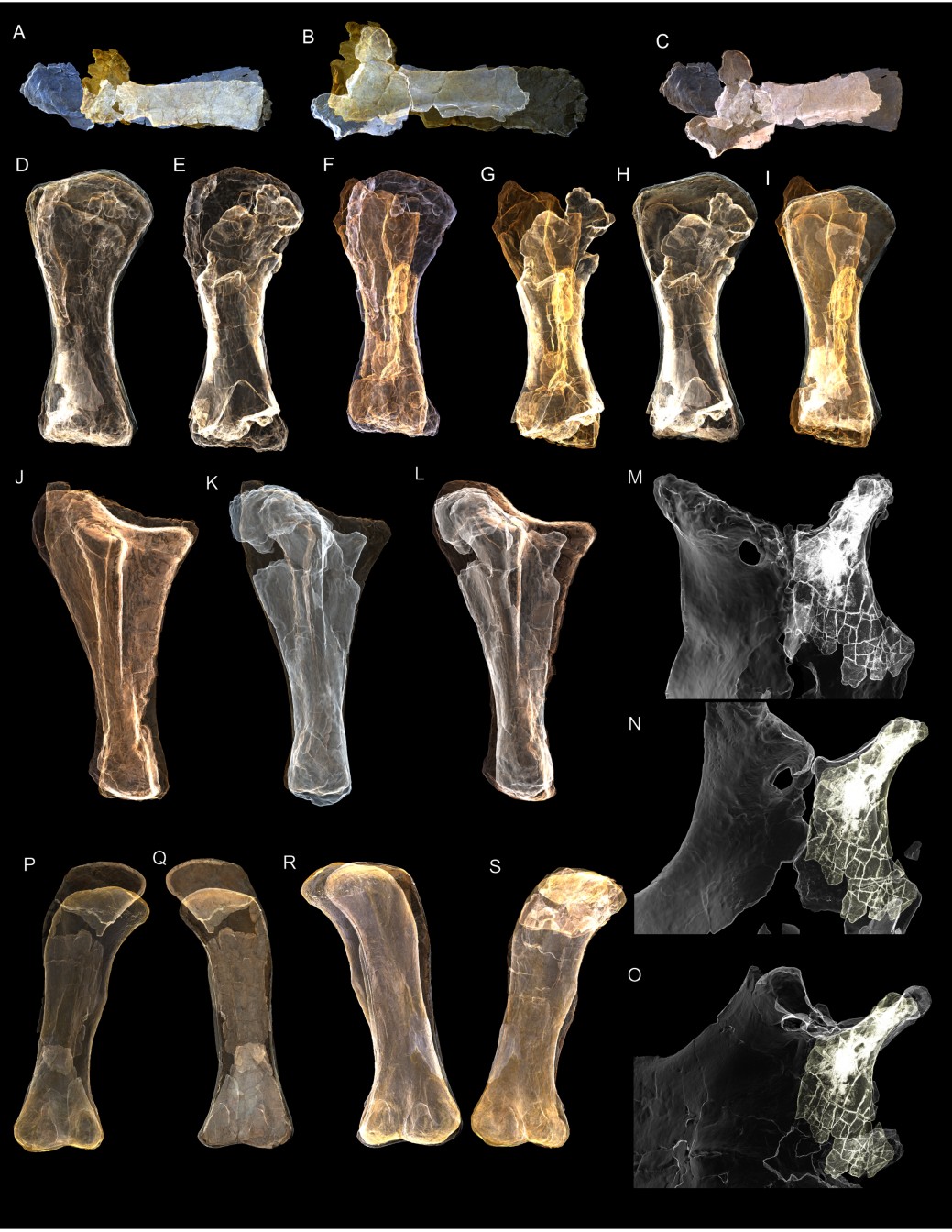

**Figure 30 Comparative Meshlab 'x-ray' renders of isometrically aligned skeletal elements shared between *Australotitan cooperensis* gen. et sp. nov. and other Winton Formation sauropods.** (A-C) Comparison of preserved scapulae in lateral view. (A) *A. cooperensis* aligned to *D. matildae*. (B) *A. cooperensis* gen. et sp. nov. aligned to *W. wattsi*. (C) *W. wattsi* aligned to *D. matildae*. (D–I) Comparison of preserved humeri in anterior view. (D) *A. cooperensis* gen. et sp. nov. aligned to *D. matildae*. (E) *A. cooperensis* gen. et sp. nov. aligned to *S. elliottorum*. (F) *A. cooperensis* gen. et sp. nov. aligned to *W. wattsi*. (G) *W. wattsi* aligned to *S. elliottorum*. (H) *D. matildae* aligned to *S. elliottorum*. (I) *D. matildae* aligned to *W. wattsi*. (J–L) Comparison of preserved ulnae in mediolateral view. (J) *A. cooperensis* gen. et sp. nov. aligned to *D. matildae*. (K) *A. cooperensis* gen. et sp. nov. aligned to *W. wattsi*. (L) *D. matildae*

**Figure 30** (continued)
aligned to *W. wattsi*. (M-O) Comparison of preserved ischium. (M) *A. cooperensis* gen. et sp. nov. aligned
to *W. wattsi*. (N) *D. matildae* aligned to *W. wattsi*. (O) *S. elliottorum* aligned to *W. wattsi*. (P–S)
Comparison of preserved femora in posterior view. (P) QMF43302 aligned to EMF105. (Q) QMF43302
aligned to *D. matildae*. (R) EMF105 aligned to *D. matildae*. (S) Reconstructed femur of *A. cooperensis*
gen. et sp. nov. (EMF102) aligned to referred femur EMF105.

completely preserved in *W. wattsi*, it is likely to have been similar based on the close
approximation of these elements to one another (Figs. 22 & 30). The posterior margin of
the ischia in *S. elliottorum* and *A. cooperensis* curve ventrally along the distal plate margin
angling the dorsal margin of this distal-most portion posteriorly. This does not occur in
*D. matildae*, where the dorsal margin of the distal plate remains dorsally oriented.
The orientation of the distal-most plate margin is unknown in *W. wattsi*, although at the
preserved distal-most margin it begins to curve ventrally. If this curvature was to continue,
it would produce a similar posteriorly directed distal plate, as seen in *S. elliottorum* and
*A. cooperensis*.

The ischium of *A. cooperensis* is larger than in both *S. elliottorum* and *D. matildae*, but
smaller than *W. wattsi*. The ischium is the only comparable element across these taxa
where *A. cooperensis* is not substantially larger. Both holotype specimens of *A. cooperensis*
(EMF102) and *W. wattsi* (QMF7292) are known from associated and semi-articulated
remains, which establishes the allocation of each ischium with other elements of each
holotype, therefore the size discrepancy is unlikely an artefact of having come from
multiple individuals.

The greater size of the ischium in *W. wattsi* is contrary to the relatively smaller sizes of
all other known appendicular elements in common with *A. cooperensis*. The preserved
scapula of *A. cooperensis* indicates that it had a much longer scapular blade relative to both
*D. matildae* and *W. wattsi*. However, this element is much more gracile in *A. cooperensis*,
having a mediolaterally thin scapular blade (Figs. 10 & 30). Although incomplete, the
reconstructed humerus of *W. wattsi* is longer than that of *D. matildae* and *S. elliottorum*,
but considerably smaller with a narrower midshaft breadth for length in comparison to
*A. cooperensis* (Figs. 15 & 30). In mid-diaphyseal cross-sectional shape, *W. wattsi* is
ovo-rectangular like *S. elliottorum*, but compared to the mediolaterally oblong and
anteroposteriorly compressed *A. cooperensis* and *D. matildae* (Fig. 16). The ulna of
*W. wattsi* has the most robust midshaft cross-sectional shape when compared to the
smaller *D. matildae* and larger *A. cooperensis* (Fig. 18). The proximal olecranon process is
robust, broad and rounded in both *D. matildae* (complete) and *W. wattsi* (incomplete)
compared with the gracile, narrow and acute process in *A. cooperensis* (Figs. 18, 19 & 30).

The femur of *A. cooperensis* differs from the femur of *D. matildae* by possessing a
relatively anteroposteriorly narrower femoral shaft, including a narrower proximal femoral
head. The distal condyles of *A. cooperensis* are beveled more medially in anterior and distal
aspects relative to that of *D. matildae*, and all other northern Winton Formation femora
compared (Figs. 24–26 & 30).

## Comparison with non-Winton Formation semi-contemporaneous members of the Titanosauria worldwide (e.g. Latest Albian-early Turonian)

Comparisons were not possible with the following semi-contemporaneous titanosaurian taxa due to a lack of overlap in preserved elements: *Austrosaurus mckillopi* (*Poropat et al., 2017*), *Sarmientosaurus musacchioi* (*Martinez et al., 2016*), *Drusilasaura deseadensis* (*Navarrete, Casal & Martínez, 2011*), *Jiutaisaurus xidensis* (*Wu et al., 2006*) and *Borealosaurus wimani* (*Hailu et al., 2004*). In addition, comparisons were not possible due to poor preservation or a lack of detailed descriptions or figures of the overlapping elements for the following taxa: *Huanghetitan liujiaxiaensis* and *Huanghetitan ruyangensis* (*Junchang et al., 2007*; *You et al., 2006*), *Quetecsaurus rusconii* (*González Riga & David, 2014*) and *Choconsaurus baileywillisi* (*Simón, Salgado & Calvo, 2017*).

Several titanosaurians are only comparable by one or two overlapping appendicular elements, and in some cases, size differences are the clearest feature that differentiates these taxa apart. *A. cooperensis* differs from *Pa. stromeri* (*Smith et al., 2001*) and *Andesaurus delgadoi* (*Mannion & Calvo, 2011*) by possessing a rounded proximal humeral epiphysis without a distinct proximolateral corner that meets at a right-angle. In addition, *Pa. stromeri* has a larger humerus with a mediolaterally narrower diaphysis. *An. delgadoi* is smaller and also has a mediolaterally narrower humeral diaphysis.

*A. cooperensis* possesses a smaller femur compared to the specimen referred to as *Argentinosaurus huinculensis* (*Bonaparte, 1996*). *A. cooperensis* differs from *Aegyptosaurus baharijensis* (*Stromer, 1932*) by being larger and possessing a mediolaterally broad midshaft for both the femur and humerus. *A. cooperensis* differs from *Dongyangosaurus sinensis* by possessing a pubis that is much longer than the ischium (*Junchang et al., 2008*). *A. cooperensis* differs from *Ruyangosaurus giganteus* by possessing a more mediolaterally broad and robust femur relative to the long and gracile femur of *R. giganteus* (*Lü et al., 2009*).

*A. cooperensis* differs from *Ep. sciuttoi* (*Martínez et al., 2004*) by being much larger in all comparative elements (i.e. humerus, ulna, femur, pubis and ischium). *A. cooperensis* possesses a less stocky and robust humerus, a distinct interosseous ridge and an accessory ridge on the distal end of the anterolateral process of the ulna. *A. cooperensis* differs from *P. mayorum* (*Carballido et al., 2017*) by being much smaller in all comparative elements except for the ulna with which it is of similar length and anterior width. *A. cooperensis* lacks the dorsoventrally deep scapular blade with distinct mid-ridge of *P. mayorum*. Both the humerus and femur are more elongate in anterior outline in *P. mayorum* than in *A. cooperensis*, which is also reflected in a narrower anteroposterior, but broader mediolateral midshaft width.

Based on the shared elements (scapula, humerus and ulna) *A. cooperensis* is larger than *Angolatitan adamastor* (*Mateus et al., 2011*). The acromion plate of the scapula of *An. adamastor* is broader dorsoventrally with a distinct posteroventral process as seen in *W. wattsi* and *D. matildae*. The dorsal and ventral margins of the scapular blade are curved to an expanded distal end, similar to that seen in *D. matildae* and *W. wattsi* but not in

*A. cooperensis*. The humerus of *An. adamastor* is an elongate element relative to
*A. cooperensis*, *D. matildae* and *S. elliottorum*; however, it is comparatively closer to the
reconstructed humerus of *W. wattsi* (Fig. 15Z). The proximolateral corner exhibits a
greater angularity relative to *D. matildae* and *A. cooperensis*. The ulna is uninformative in
*An. adamastor*, retaining the general titanosauriform shape and robust proximal
tri-radiate epiphysis. The ulna does not possess accessory distal processes or the
interosseous ridge of radial fossa seen in *A. cooperensis*. Although An. adamastor does not
compare favourably with *A. cooperensis* in scapula, humerus and ulna morphology, it does
closely resemble the elements preserved for *W. wattsi*, warranting more detailed
comparison of these two taxa in future comparative assessments.

*A. cooperensis* is much larger than *Mnyamawamtuka moyowamkia* (*Gorscak &
O'Connor, 2019*) in all preserved elements. *A. cooperensis* shares with *Mn. moyowamkia*
the scapula, humerus, ulna, pubis, ischium and femur, but all elements, except for the
scapula, are so poorly preserved that frustratingly they cannot be adequately compared.
The scapula is similar to *A. cooperensis* by being a lightly built element with a relatively
narrow acromion plate compared to the scapular blade. The blade is near straight with an
absent posteroventral process, similar to that of *A. cooperensis* and *Y. datangi*.

*A. cooperensis* is morphologically similar to *E. lilloi* (*Mannion & Otero, 2012*); however,
it is larger in all comparative elements (i.e. scapula, humerus, ulna, femur and pubis).
The distal epiphysis of the humerus approaches a similar cross-sectional shape, being
nearly tri-lobate in distal view; however, *A. cooperensis* has a much greater mediolateral
expansion of the distal epiphysis and a laterally flared ectepicondylar margin of the lateral
condyle. The proximal epiphysis of the ulna in *E. lilloi* bears a similar reduction of the
anterolateral and olecranon processes relative to the much longer anteromedial process;
however, the radial fossa does not possess the distinct interosseous ridge or the distal
anterolateral accessory ridge present in *A. cooperensis*. The pubes are similarly broadened
anteroposteriorly along the pubic blade in both taxa. However, the iliac peduncle of *E. lilloi*
is directed more anteriorly and flattened in comparison to the anterodorsally pointed
peduncle of *A. cooperensis*. The distal margin of the pubic blade is broader and truncated in
*E. lilloi* compared to the rounded distal blade margin in *A. cooperensis*.

## Comparisons with other large-bodied titanosaurians

In addition to the above comparisons between semi-contemporaneous titanosaurians,
it is also worthy to compare *A. cooperensis* with other large-bodied titanosaurians of
comparable size of preserved elements. *Futalognkosaurus dukei* possesses a similar-
sized humerus (1,510 mm) and near similar femur (1,945 mm) (*Benson et al., 2014*);
however, morphological comparisons were not possible. The pubis and ischium can be
compared (*Calvo et al., 2007b*) with the pubis having similar overall morphology, but
differing from *A. cooperensis* by possessing a anteroposteriorly longer iliac peduncle, and
by being thicker along the dorsoventral length of the pubic blade. A lateral ridge along the
mid-line of the blade is clearly visible in *F. dukei*, but not in *A. cooperensis*. A lateral ridge
along the pubic blade is also present in *S. elliottorum*, and considered an autapomorphy
(*Poropat et al., 2016*). The pubic articulation of the ischium in *F. dukei* is shorter than the

long, medially curved articulation seen in *A. cooperensis*, *D. matildae*, *S. elliottorum* and *W. wattsi*.

Both *Antarctosaurus* sp. and *T. eocaudata* (*Juárez Valieri & Calvo, 2011*) possess more elongate femora with a more bulbous and anteroposteriorly thicker greater trochanter and femoral head when compared to *A. cooperensis*. *No. gonzalezparejasi* possesses a longer humerus (1,760 mm) (*Benson et al., 2014*) and unlike *A. cooperensis* has: a proximal humeral epiphysis with a distinct proximolateral corner that meets at right angles; a flattened lateral to bulbous medial humeral head profile in anterior view; a proximodistally reduced deltopectoral crest; and a narrower midshaft diaphysis (*Gonzalez Riga et al., 2016*). *Al. sanjuanensis*'s referred humerus (*D'Emic, Wilson & Williamson, 2011*; *Gilmore, 1946*) is the same size (1,503 mm) (*Benson et al., 2014*), with a rounded proximal humeral epiphysis, similar to that of *A. cooperensis*. The referred ischia of *Al. sanjuanensis* (*Tykoski & Fiorillo, 2016*) are also similar to *A. cooperensis* including an extensive ischial contact with its contralateral element. Unlike *A. cooperensis*, the posterodistal margin of the ischial blades are directed posteriorly, past the position of the posterior margin of the iliac peduncle. The scapula of *Al. sanjuanensis* possesses a central ridge along the scapular blade that is not seen in *A. cooperensis*. *Da. binglingi* has a smaller femur (1,770 mm) (*Benson et al., 2014*), but has similarly oriented distal condyles that are bevelled in an anterolateral to posteromedial orientation when viewed distally (*You et al., 2008*). *Da. binglingi* differs from *A. cooperensis* by possessing a narrower diaphysis and dorsolaterally beveled distal condyles in posterior view.

*Dr. schrani* has a longer humerus (1,760 mm) and femur (1,910 mm) (*Benson et al., 2014*); however, is similar in overall appendicular morphology (*Ullmann & Lacovara, 2016*). The scapula shares with *A. cooperensis*: a long, straight scapular blade with subparallel dorsal and ventral margins; the absence of a central ridge of the scapular blade; and a mediolaterally thin blade. It also possesses a mediolaterally thin and gracile acromion plate. However, the acromion plate is massively expanded dorsoventrally in excess of that estimated in *A. cooperensis*. Similar to *A. cooperensis*, the humerus of *Dr. schrani* is proximally and distally broad across the epiphyses as well as being anteroposteriorly narrow and mediolaterally broad at the midshaft. *Dr. schrani* differs from *A. cooperensis* by the deltopectoral crest neither reaching as far distally nor possessing the distinctly tri-lobate distal epiphysis present in *A. cooperensis*. The ulna of *Dr. schrani* differs from *A. cooperensis* by being more robust and stocky, with near-equal anterolateral and anteromedial processes and an oblong-shaped distal epiphysis. The pubis of *Dr. schrani* differs from *A. cooperensis* by being considerably thicker along the pubic blade with a dorsoventrally short ischiadic peduncle. The ischium of *Dr. schrani* differs from *A. cooperensis* by being near-vertically oriented, with the entire dorsal surface of the ischial blade directed posteriorly. As with the pubis, the pubic peduncle is dorsoventrally short. The femur of *Dr. schrani* is similar to *A. cooperensis*, possessing an anteroposteriorly narrow and mediolaterally broad diaphyseal shaft that leads to mediolaterally expanded proximal and distal epiphyses. The distal epiphyses are bevelled in an anterolateral to posteromedial direction, a feature also seen in *Da. binglingi* (*You et al., 2008*), *L. astibiae* (*Díaz, Suberbiola & Sanz, 2013*) and cf. *L. astibiae* (*Vila et al.,*

*2012*). However, this feature has been considered to be taphonomic distortion in *Dr. schrani* created through lithostatic compression (*Ullmann & Lacovara, 2016*).

Considered together, *A. cooperensis* possesses a mosaic of features shared with titanosaurians with similar geographical (Australia) and temporal range (Latest Albian to ? Turonian), as well as similar body-size. The previously described and comparable Australian taxa (*D. matildae*, *W. wattsi* and *S. elliottorum*) share closer morphological similarities of the pubis and ischium complex with *A. cooperensis* than they do to all other taxa compared. This observation alludes to a potential shared ancestry.

Those taxa of similar geological age or similar limb size tend to share only isolated features of each element with *A. cooperensis* but this is also observed in titanosaurians from older and younger Cretaceous sites, such as the scapular similarities seen in *Y. datangi* from the Lower Cretaceous of China, or the humeral and ischial similarities of *Al. sanjuanensis* from the latest Cretaceous of North America. Such a mosaic of characteristics helps define and differentiate *A. cooperensis* from all other taxa and is especially useful in regards to those taxa found within the Winton Formation. However, the mosaic of similar and different features found in this taxon, which derive from a small number of representative appendicular elements, suggests that these characteristics will not add significantly to a phylogenetic analysis of similarly incomplete and variable taxa. The morphological similarities in titanosaurian limb morphology across multiple lineages has recently been considered, finding that potentially convergent morphologies could reflect morphofunctional similarities across lineages. However, without more detailed comparative assessments, likely by using 3-D models and geometric morphometrics this potential influence on phylogenetic signal or ecomorphology is difficult to quantify (*Páramo, Mocho & Ortega, 2020*).

## Phylogenetic position

A preliminary phylogenetic assessment, which we provide as Supplementary Information, does not resolve the phylogeny of titanosauriformes with any statistical support. However, it does allude to a possible shared relationship of the four Winton Formation taxa, as discussed below.

In the absence of such a resolved phylogeny, we can consider the phylogenetic position of *A. cooperensis* using a comparative approach, using published phylogenies and the spread of characteristics hypothesized to define particular clades. The phylogenetics of titanosaurians remains in a state of flux with multiple assessments appearing in recent years investigating the relative position of taxa in a global context, covering Late Jurassic to Late Cretaceous (*Carballido et al., 2017*; *D'Emic, 2012*; *González Riga et al., 2019*; *Gonzàlez Riga et al., 2018*; *Hechenleitner et al., 2020*; *Mannion, Allain & Moine, 2017*; *Mannion et al., 2013*; *Mannion et al., 2019a*; *Mannion et al., 2019b*).

For our comparative approach, we use a recent review of the appendicular skeletons of South American titanosaurians (*González Riga et al., 2019*) that focuses on the appendicular synapomorphies which are derived from two independent phylogenetic assessments of titanosaurians (*D'Emic, 2012*; *Mannion et al., 2013*). We find the following features present in *A. cooperensis* that are considered to be synapomorphies of

Titanosauria or clades within it: (1) The humerus length is less than 80% the femur length (= Saltasauridae) (79% for *A. cooperensis*). The length of the femur of *A. cooperensis* has been estimated in multiple different ways. Because we cannot directly confirm the length of the femur in the holotype, and with this percentage being so close to the upper limit of the expected range for saltasaurids, we treat its use as a synapomorphy for *A. cooperensis* within the Saltasauridae as dubious. (2) The humeral deltopectoral crest extends medially across the anterior face of the humerus, but this is not well developed (= Titanosauria). (3) The humeral deltopectoral crest is not expanded distally (≠ Saltasauridae). (4) Humerus with a strong posterolateral bulge around the level of the deltopectoral crest area is not well preserved or discernible in *A. cooperensis* (≠ Saltasauridae). (5) Humeral radial and ulnar condyles are undivided distally (≠ *Alamosaurus* + 'Saltasaurini'). (6) Anterior surface of the distal lateral condyle of the humerus seems to be divided by a notch in *A. cooperensis*; however, this feature is poorly defined (≠ Lithostrotia). (7) Prominent ulnar olecranon process projecting well above proximal articulation is present in *A. cooperensis* (= Lithostrotia). (8) Anteroposterior to mediolateral width ratio of iliac articular surface of pubis is ≥2.0 (= Titanosauria). (9) Acetabular margin of ischium strongly concave in lateral view such that pubic articular surface forms a proximodorsal projection (= Titanosauria or Lithostrotia). (10) No emargination of ischium distal to pubic articulation (= Titanosauria). (11) Ratio of dorsoventral width of distal end of ischial shaft to minimum shaft dorsoventral greater than 1.5 (≠ Titanosauria). (12) Femur with longitudinal ridge on anterior face of shaft (linea intermuscularis cranialis (*Otero, 2010*)) is preserved on the anterior face of the distal diaphysis in the holotype EMF102 and is well preserved along the entire anterior face of the referred femur in EMF105 (= *Alamosaurus* + 'Saltasaurini'). (13) Femoral distal condyles are bevelled 10° dorsomedially relative to shaft (= Saltasauridae) with the slightly distally projected fibular condyle that is not as exaggerated as seen in *Saltasaurus* and *Bonatitan* (*González Riga et al., 2019*).

Based on this assessment, *A. cooperensis* possesses a single synapomorphy of the Saltasauridae, being bevelled distal condyles of the femur. One character state supports and another does not support placement within the 'Saltaurini' clade (*D'Emic, 2012*). Two character states support and one does not support placement within the Lithostrotia. Finally, four character states support and two do not support placement within the Titanosauria (Table 8). Such a mosaic of synapomorphies makes any solid phylogenetic footing equivocal.

However, the distribution of the combined synapomorphic features of the appendicular skeleton recovered from two independent phylogenetic assessments of titanosauriformes (*D'Emic, 2012*; *Mannion et al., 2013*) at least supports our placement of *A. cooperensis* within Titanosauria and suggests that it could be part of the Lithostrotia. Whether or not *A. cooperensis* is a lithostrotian titanosaurian, or a non-lithostrotian titanosaurian is remarkably the same situation for two other Winton Formation taxa: *S. elliottorum* and *D. matildae* (*Mannion et al., 2013*; *Poropat et al., 2016*).

The most recent phylogenetic analyses that include the Winton Formation titanosaurians (*González Riga et al., 2019*; *Gonzàlez Riga et al., 2018*; *Mannion, Allain &*
**Table 8  Synapomorphies of Titanosauria in Australian Taxa.**

| Synapomorphy | Clade | *Australotitan cooperensis* | *Diamantinasaurus matildae* | *Wintonotitan wattsi* | *Savannasaurus elliottorum* |
|---|---|---|---|---|---|
| **Scapula** | | | | | |
| Scapula, ventral margin with well-developed ventromedial process | Titanosauria | ? | ✓ | ✓ | ? |
| **Humerus** | | | | | |
| humerus length less than 80% femur length | Saltasauridae | ✓ (~79%) | ✗ (85%) | ? | ? |
| deltopectoral crest extends medially across anterior face | Titanosauria | ✓ - less than *Saltasaurus/ Opithsocoelicaudia* | ✓ - less than *Saltasaurus/ Opithsocoelicaudia* | ? | ? |
| deltopectoral crest strongly expanded distally | Saltasauridae | ✗ | ✗ | ✗ | ✗ |
| strong posterolateral bulge around level of deltopectoral crest | Saltasauridae | ? | ✗ | ✗ | ✗ |
| radial and ulnar condyles divided distally | *Alamosaurus* + 'Saltasaurini' | ✗ | ✗ | ? | ? |
| Anterior surface of distal lateral condyle of humerus undivided | Lithostrotia | ✗ | ✗ | ? | ? |
| **Radius** | | | | | |
| radius distal end beveled ~20° proximolaterally relative to shaft | Saltasauridae | ? | ✓ | ✓ - poorly preserved | ✗ |
| **Ulna** | | | | | |
| Prominent olecranon process, projecting well above proximal articulation | Lithostrotia | ✓ | ✓ | ✓? | ? |
| **Manus** | | | | | |
| Metacarpal I:metacarpal II/III proximodistal length ratio ≥ 1.0 | Lithostrotia | ? | ✗ | ? | ✗ |
| **Pubis** | | | | | |
| Anteroposterior to mediolateral width ratio of iliac articular surface of pubis ≥2.0 | Titanosauria | ✓ | ✓ | ? | ? |
| **Ischium** | | | | | |
| Acetabular margin of ischium strongly concave in lateral view such that pubic articular surface forms proximodorsal projection | Titanosauria or Lithostrotia | ✓ | ✓ | ✓? | ✓ |
| No emargination of ischium distal to pubic articulation | Titanosauria | ✓ | ✓ | ✓? | ✓ |
| Ratio of dorsoventral width of distal end of ischial shaft: minimum shaft dorsoventral width <1.5 | Titanosauria | ✗ (1.63) | ✗ (1.73) | ✗ (~1.64) | ✗ (1.74) |
| **Femur** | | | | | |
| Femur with longitudinal ridge on anterior face of shaft | *Alamosaurus* + 'Saltasaurini' | ✓ | ✓ | ? | ? |
| Femoral distal condyles beveled 10° dorsomedially relative to shaft | Saltasauridae | ✓ - less than *Saltasaurus/ Bonatitan* | ✓ - less than *Saltasaurus/ Bonatitan* | ? | ? |
| % of known characters | | 75% | 100% | 31–50% | 43% |
| **Shared Characters** | | | | | |
| Not Titanosauria | | 1 | 1 | 1 | 1 |

(Continued)

| Table 8 (continued) | | | | | |
|---|---|---|---|---|---|
| Synapomorphy | Clade | *Australotitan cooperensis* | *Diamantinasaurus matildae* | *Wintonotitan wattsi* | *Savannasaurus elliottorum* |
| Within Titanosauria | | 8 | 9 | 2 + 3 possible | 2 |
| Within Lithostrotia or Saltasaurini/Saltasauridae | | 5 | 5 | 1 + 2 possible | 1 |
| Not within Lithostrotia or Saltasaurini/ Saltasauridae | | 4 | 8 | 3 | 5 |

**Note:**
Synapomorphies of Titanosauria from *González Riga et al. (2019)*.

*Moine, 2017*; *Mannion et al., 2019a*; *Mannion et al., 2019b*) provide context for our discussion in two important ways. Firstly, there is growing support for a nearly global distribution of most titanosaurian clades by the Early Cretaceous, and by extension, titanosaurians from Cretaceous Australia could potentially represent one or more of those clades. However, there is also growing support for clades restricted to specific regions, such as Colossosauria, Rincosauria, and Lognkosauria of South America (*González Riga et al., 2019*). Therefore, the mosaic of features that *A. cooperensis* shares with taxa from older, semi-contemporaneous and geographically distant regions could potentially place it within any of these clades unless homoplasy has played a more significant role in the evolution of sauropod appendicular elements than previously thought (*Upchurch, 1998*).

Secondly, the relative positions of the Australian taxa are unstable, changing position depending on the phylogenetic methodologies and taxa included within each assessment. The relative phylogenetic position of *W. wattsi* as basal to *D. matildae* has changed since the first phylogenetic assessment was undertaken (*Hocknull et al., 2009*) and further since the addition of *S. elliottorum* (*Poropat et al., 2016*). *W. wattsi* has been resolved as a non-titanosaurian somphospondylan (*Hocknull et al., 2009*; *Poropat et al., 2015a*), but has also been recovered outside of titanosauriformes (*Carballido et al., 2011b*; *Hechenleitner et al., 2020*); more derived than *D. matildae* (*Mannion et al., 2013*); within the titanosaurian 'Andesauroidea'; or sister taxon to the Titanosauria (*Mannion et al., 2019a*).

Over time, new phylogenetic assessments have proposed a more basal position for *D. matildae*, first falling outside of the derived Saltasauridae and then further outside Lithostrotia. *D. matildae* has variably been recovered as a derived saltasaurid (*Gonzalez Riga et al., 2016*; *Hocknull et al., 2009*; *Mannion et al., 2019a*; *Mannion et al., 2019b*; *Upchurch, Mannion & Taylor, 2015*); a non-lithostrotian titanosaurian (*González Riga et al., 2019*) with *S. elliottorum* as sister taxon (*Gonzàlez Riga et al., 2018*; *Mannion, Allain & Moine, 2017*; *Mannion et al., 2019a*; *Mannion et al., 2019b*; *Poropat et al., 2015b*); or close to *Yonglinglong* (*Li et al., 2014*).

With the addition of more taxa to these newer phylogenetic analyses, especially adding taxa from Asia, the once derived position of *D. matildae* (along with *S. elliottorum*), relative to *W. wattsi* has eroded. Therefore, with such instability in their relative positions it would be premature to add a further fragmentary taxon to derive another alternative phylogeny.

Our new taxon, along with the others from the Winton Formation, is unlikely to provide new phylogenetically useful data to these large-scale global analyses until the known better-preserved specimens such as those currently being prepared are available (*Hocknull et al., 2019*; *Poropat et al., 2019*).

All four taxa possess appendicular elements and for those elements with overlap between at least two taxa, they allow comparison between each other and to the appendicular synapomorphies found in Titanosauria (*González Riga et al., 2019*). The scapulae of *D. matildae* and *W. wattsi* both possess a well-developed ventromedial process of the ventral margin (= Titanosauria) (*Hocknull et al., 2009*; *Poropat et al., 2015a*; *Poropat et al., 2016*; *Poropat et al., 2015b*) (Figs. 9–10 & 28A–28C), although titanosaurian outgroup taxa, including *C. insignis*, also possess this feature (*Carballido et al., 2011a*; *González Riga et al., 2019*). The area where this feature would be found in *A. cooperensis* and *S. elliottorum* is missing.

The relative humerus to femur length of *A. cooperensis*, estimated at 79%, is less than 85% for *D. matildae*. However, they are either at or above the limit of this feature being a synapomorphy of Saltasauridae (i.e., less than 80%). Neither *W. wattsi* or *S. elliottorum* preserve a complete humerus and femur for comparison.

The deltopectoral crest extends medially across the anterior face of the humerus in both *A. cooperensis* and *D. matildae* (= Titanosauria) although it does not extend as far as that of derived titanosaurians like *O. skarzynskii* and *Sa. loricatus*. These features are missing from the preserved humeri of *W. watts* and *S. elliottorum*. Based on what is preserved of the humeri in the four Australian taxa, none of them possess a distally expanded deltopectoral crest or a strong posterolateral bulge level with the deltopectoral crest (≠ Saltasauridae). The distal humeral condyles of *A. cooperensis* and *D. matildae* are undivided (≠ *Alamosaurus* + 'Saltasaurini') and both possess a distal lateral condyle that has a divided anterior surface (≠ Lithostrotia). The proximal and distal condyles of the humeri of *W. wattsi* and *S. elliottorum* are unknown. The midshaft cross-sectional shape of *W. wattsi* and *S. elliottorum* approximate one another by being anteroposteriorly thick, creating a rounded (ovo-rectangular) outline, whilst in *A. cooperensis* and *D. matildae*, this outline is mediolaterally broad, creating a more oblong outline.

The distal end of the radius is bevelled ~20° proximolaterally relative to the shaft in *D. matildae* (*Poropat et al., 2015b*) and estimated in *W. wattsi* (*Poropat et al., 2015a*) although the distal ends in the *W. wattsi* holotype radii are very poorly preserved (= Saltasauridae). The radius is not bevelled in *S. elliottorum* (≠ Saltasauridae), and the radius is unknown in *A. cooperensis*. A prominent olecranon process is present in *A. cooperensis* and *D. matildae* (= Lithostrotia), but is unknown in *S. elliottorum* and not preserved in *W. wattsi*. However, this feature in *W. wattsi* is likely similar to *D. matildae* based on shape comparisons of this element (see Figs. 18B & 18C & 28L) and would then place *W. wattsi* within the Lithostrotia. The relative size of metacarpal I to metacarpal II or III is less than 1.0 in *S. elliottorum* and *D. matildae* (≠ Lithostrotia). This characteristic is not preserved in *W. wattsi* and unknown in *A. cooperensis*.

The anteroposterior length to mediolateral width of the iliac articular surface of the pubis is greater than 2.0 in both *A. cooperensis* and *D. matildae* (= Titanosauria) (Table 8).

The pubis is unknown in *W. wattsi* and the iliac articulation of the pubis is missing from both sides of the pelvis of *S. elliottorum*. In *D. matildae*, *A. cooperensis* and *S. elliottorum*, the acetabular margin of the ischium is strongly concave in lateral view such that the pubic articular surface forms a proximodorsal projection (= Titanosauria or Lithostrotia). The acetabular rim of the ischium in *W. wattsi* is broken along its entire length, exposing internal cancellous bone (Fig. 22). This indicates the loss of substantial bone around the acetabular rim. Therefore, the morphology of the acetabular rim cannot be accurately defined, or is questionable. The very similar shape of the ischium of all four Australian taxa suggests that the acetabular rim of *W. wattsi* could have been concave, changing this feature from a typically non-titanosaurian character state to a character state found in Titanosauria or Lithostrotia (Fig. 30).

There is no emargination of the ischium distal to pubic articulation in *A. cooperensis*, *D. matildae* and *S. elliottorum* (= Titanosauria). This region of the ischium in *W. wattsi* is not preserved, being broken along the pubic articulation and medial region where the contralateral elements may have met. The ischium curves ventrally at the broken medial margin suggesting a significant extension of ischium directed medioventrally, similar to that observed in *S. elliottorum*. Therefore, it is possible that the ischia did meet at a symphysis with no emargination, thus a feature synapomorphic in Titanosauria. The ratio of dorsoventral width of the distal end of ischial shaft to minimum shaft dorsoventral width is greater than 1.5 in *A. cooperensis*, *S. elliottorum*, *D. matildae* and estimated to be so in *W. wattsi* (≠ Titanosauria) (Tables 8 and 9).

The femur is only known in *A. cooperensis* and *D. matildae*. The femur of *A. cooperensis* possesses a longitudinal ridge on the anterior face of shaft (linea intermuscularis cranialis (= *Alamosaurus* + 'Saltasaurini')), but this is absent in *D. matildae* (≠ *Alamosaurus* + 'Saltasaurini'). The distal condyles are bevelled 10° dorsomedially with a slightly distally projected fibular condyle, unlike that of highly derived saltasaurids (*González Riga et al., 2019*).

Summarising the above comparative phylogenetic appraisal of the four Australian taxa by using synapomorphies derived from three independent phylogenetic character assessments (Tables 8 and 9), we find only one character-state of the sixteen, that are found in all four taxa, that is not a synapomorphy of Titanosauria. Therefore, there is support for the placement of all four Australian taxa within the Titanosauria. In the ischium, the ratio of dorsoventral (anteroposterior) width of the distal end of the shaft to the minimum shaft dorsoventral (anterior-posterior) width is greater than 1.5, which is not a synapomorphy of Titanosauria. The ratios for the four Australian taxa are very similar between *A. cooperensis* (1.63) and *W. wattsi* (1.64 est.), and between *D. matildae* (1.73) and *S. elliottorum* (1.74), which reflects the overall similar morphology of the ischium (Figs. 30M–30O). The shared similarities of the ischium, regardless of overall body-size differences and other appendicular differences, may point to a synapomorphy uniting all four Australian taxa.

Several other features are shared between the four Australian taxa and are summarised in Table 9. *W. wattsi* shares with *D. matildae* a proximal medial tuberosity of the scapular blade (Fig. 9). *W. wattsi* shares with *S. elliottorum* amphicoelous anterior caudal

**Table 9 Shared features between two or more Australian species.**

| Characteristic | *Australotitan* | *Diamantinasaurus* | *Wintontitan* | *Savannasaurus* |
|---|---|---|---|---|
| **Scapula** | | | | |
| Medial tuberosity on the proximal scapular blade | ✗ | ✓ | ✓ | ? |
| Proximoventral process | ? | ✓ | ✓ | ? |
| **Humerus** | | | | |
| Midshaft cross-sectional shape | Mediolaterally broad, anteroposteriorly narrow | Mediolaterally broad, anteroposteriorly narrow | Mediolateral breadth similar to anteroposterior length | Mediolateral breadth similar to anteroposterior length |
| **Pubis** | | | | |
| Dorsoventral thickness along pubic blade. | Thin | Thick | ? | Thin |
| **Ischium** | | | | |
| Distal ischial blade ventrally curved, dorsal margin posteriorly facing. | ✓ | ✗ | ?✓ | ✓ |
| **Anterior Caudal Vertebrae** | | | | |
| Amphicoelous | ✓* | ✓* | ✓ | ✓ |
| Pneumatic neural arch and zygopophyses | ? | ? | ✓ | ✓ |
| Centrum cancellous | ✓* | ✓* | ✓ | ✓ |

**Note:**

\* Assumed present due to ubiquitous presence within the Winton Formation (See "Discussion").

vertebrae that bear pneumatic neural arches and zygopophyses with centra possessing dense cancellous bone (Figs. 31–33). These shared features of the ischia, scapulae and caudal vertebrae have not been observed in combination with other members of the Titanosauria so could be considered synapomorphies that unite the Australian taxa. In addition, we observe that all of the known sauropod anterior and middle caudal vertebrae from the Winton Formation, both northern and southern-central sites are ubiquitously amphicoelous (Figs. 29, 31–33). Although most of the isolated caudal vertebrae are not taxonomically allocated to a known Australian taxon, it is revealing that they are among the most common of the non-appendicular elements preserved in the Winton Formation, and yet all of them are amphicoelous. This, although circumstantial, one could hypothesise that the anterior and middle caudal vertebrae of *D. matildae* and *A. cooperensis* were likely amphicoelous. Such a hypothesis is supported by the presence of amphicoelous middle and distal caudal vertebrae found at the referred localities of *A. cooperensis* (EML010 and EML012) (Figs. 29, 31–33) and *D. matildae* (QML1333/ AODL127). Of note here is the lack of sauropod proceolous caudal vertebrae from the Winton Formation. Considering the global distribution of titanosaurian clades by the mid-Cretaceous, and the presence of proceolous caudal vertebrae in taxa from most continents, it seems strikingly at odds with the observed amphicoelous-only caudal vertebrae from Australia.

One feature currently distinguishing the anterior caudal vertebrae of *S. elliottorum* from *W. wattsi* is the presence in *S. elliottorum* of pneumatic fossae (*Poropat et al., 2016*). These

 

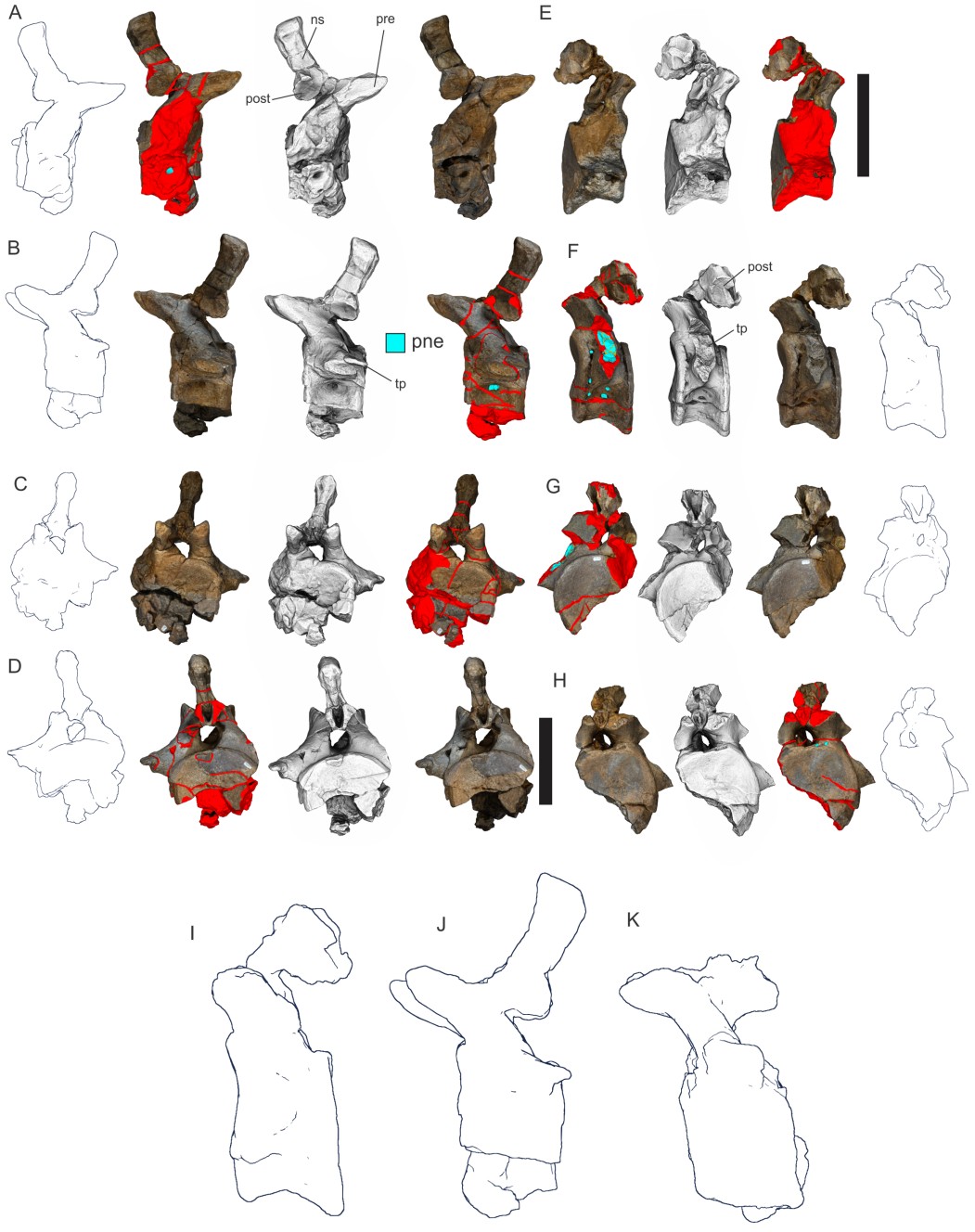

**Figure 31 Anterior caudal vertebrae from *Savannasaurus elliottorum* holotype (QMF7292).** (A) Right lateral view. (B) left lateral view. (C) Cranial view. (D) Caudal view. (E) Right lateral view. (F) Left lateral view. (G) Anterior view. (H) Posterior view. (I–K) Anterior caudal vertebra of *S. elliottorum* (I & J) compared to *W. wattsi* (K) isometrically scaled to minimum central cranial-caudal length. All in left lateral orthogonal outline view. 3-D image rendering methods used included, natural; A, D, F, G (right), B, C, E, H (left); ambient occlusion with radiance scaling, A–H (middle); coloured schematic (see Fig. 8), A, D, F, G (left) & B, C, E, H (right); Orthogonal outline edge detect, A–D (far left), F–H (far right) & I–K. Abbreviations: pne; pneumatic cavities; post, postzygopophysis; pre, prezygopophysis; ns, neural spine; tp, transverse process. Scale bars = 20 cm.

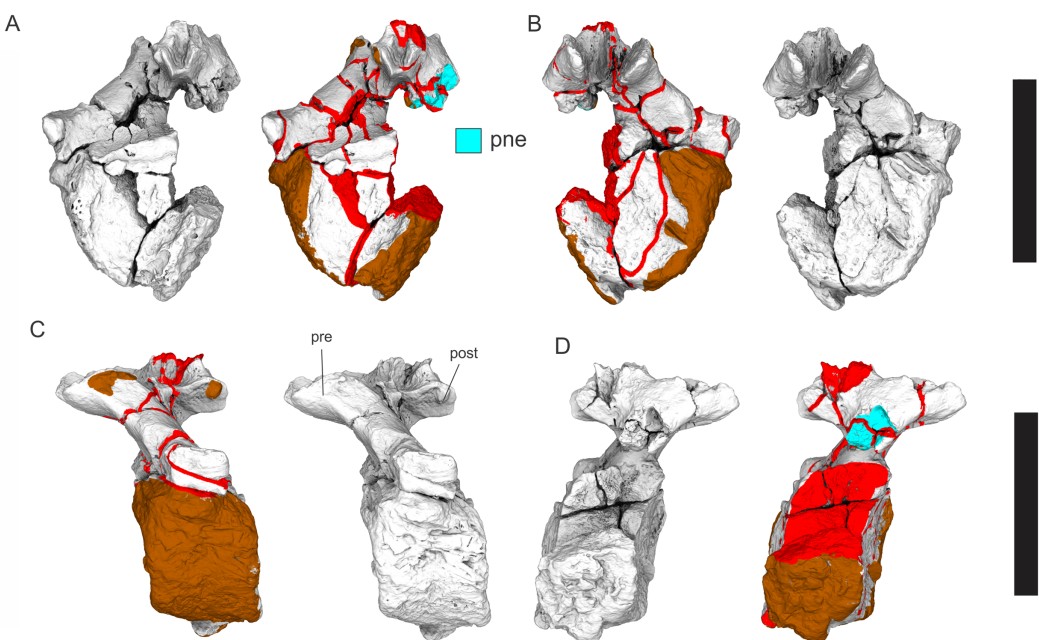

**Figure 32 Anterior caudal vertebra from *Wintonotitan wattsi* holotype (QMF7292).** (A) Cranial view. (B) Caudal view. (C) Left lateral view. (D) Right lateral view. 3-D image rendering methods used included, ambient occlusion with radiance scaling A & D (left), B & C (right); coloured schematic (see Fig. 8) A & D (right), B & C (left). Abbreviations: pne; pneumatic cavities; post, postzygopophysis; pre, prezygopophysis. Scale bars = 20 cm.               

fossae possess pneumatic pores that lead into the centrum; however, they do not enter a camellate internal structure, instead, the internal structure of the centrum is dense cancellous bone. Dorsally, large camellate internal structures are observable in cross-section, occurring within the neural arch and zygopophyses (Fig. 31). The presence in anterior amphicoelous caudal vertebrae of pneumatic fossae, pores, pneumatic neural arches and zygopophyses, but with a solid cancellous bone centrum, are symplesiomorphic characteristics of titanosauriformes and lithostrotian titanosaurians (*Mannion, Allain & Moine, 2017*; *Mannion et al., 2013*; *Wedel & Taylor, 2013*; *Whitlock, D'Emic & Wilson, 2011*). This suggests that the Australian taxa have uniquely retained symplesiomorphic features of the tail but possess derived titanosaurian to saltasaurid features of the appendicular skeleton. It would seem that all of the Australian taxa did not possess the derived proceolous caudal vertebrae of saltasaurid titanosaurians (*Zurriaguz & Cerda, 2017*).

We CT scanned the anterior caudal vertebrae of *W. wattsi* that reveal the presence of pneumatic camellate chambers in the neural arch and zygopophyses with dense cancellous bone within the amphicoelous anterior caudal vertebra of this taxon (Figs. 32 & 33). However, there is no clear indication of external pneumatic pores (Fig. 32). Thus pneumaticity of the anterior caudal neural arch and zygopophyses paired with dense cancellous bone within the centrum is a characteristic feature now shared between *W. wattsi* and *S. elliottorum*.

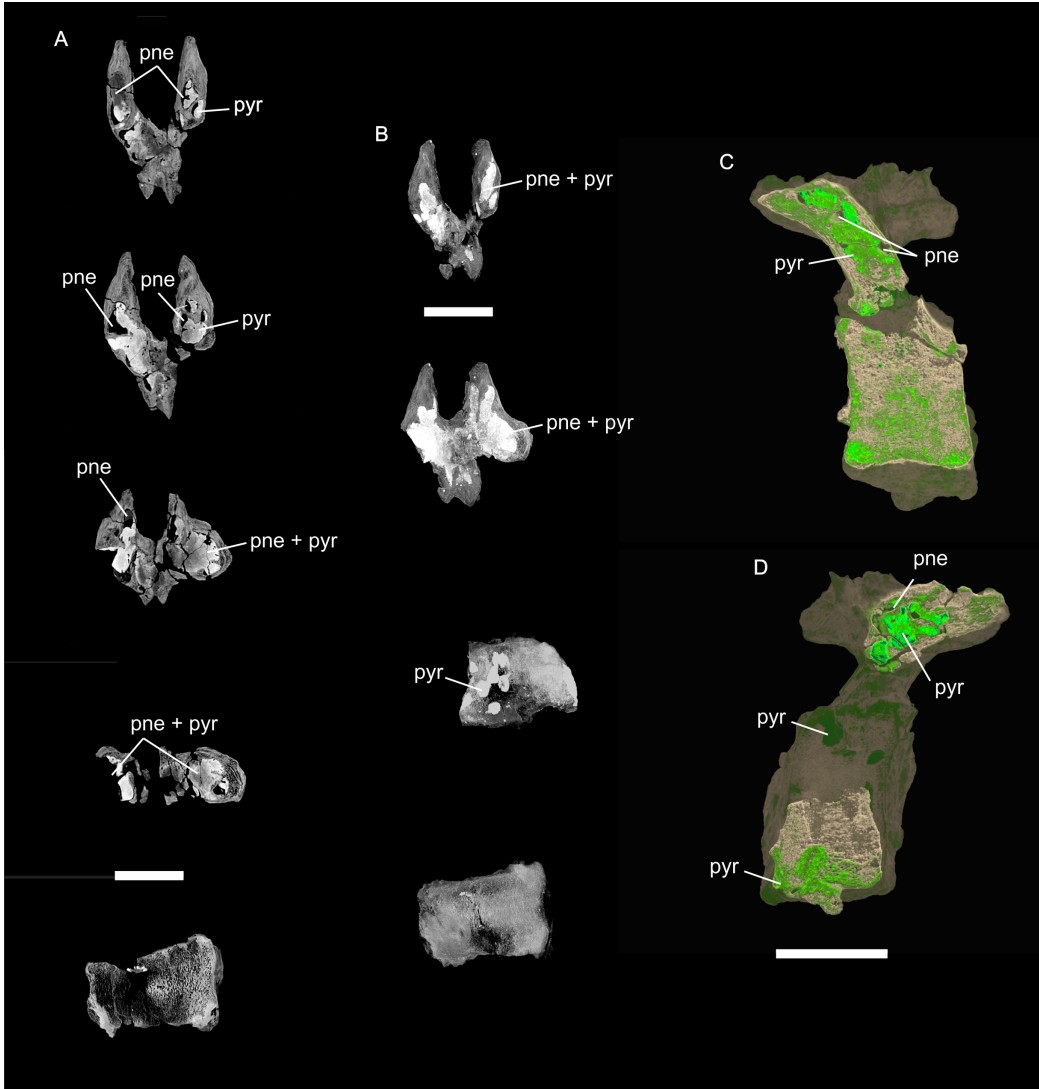

**Figure 33 Anterior caudal vertebra from *Wintonotitan wattsi* holotype (QMF7292) showing pneumatic cavities within the neural arch.** (A) A series of absorption contrast CT scan images taken from dorsal view through the prezygopophyses, neural arch and centrum. Revealing the internal cavities of the zygopophyses and neural arch that have been infilled with a dense material (iron-oxide pseudomorph of pyrite). (B) A series of maximum intensity CT scan images taken from dorsal view through the prezygopophyses, through the neural arch and into the centrum. (C & D) Coloured volume renders of the anterior caudal vertebra, clipped longitudinally through the vertebra at the position of the left prezygopophysis (C) and right prezygopophysis (D) to reveal the internal pneumatic cavities that have been partially infilled with iron-oxide pseudomorph of pyrite. Abbreviations: pne; pneumatic cavities, pyr; dense material infill (pseudomorph of pyrite). Scale bars = 5 cm (A & B); 10 cm (C & D).

The placement of *W. wattsi* within the Titanosauria is contra previous assessments that found it to be a non-titanosaurian somphospondylan (*Poropat et al., 2015a*). However, a more recent analysis has it occupying a position either within the Titanosauria, as part of the 'Andesauroidea', or as the sister-taxon to the Titanosauria clade (*Mannion et al., 2019a*). We recognise the very poor state of preservation in *W. wattsi* which likely

contributes to this unstable phylogenetic position, with less than 50% of the characters considered here available in the holotype. However, based on the similarities shared with the other Winton Formation taxa and our preliminary computational phylogenetic analysis supporting, albeit poorly, a more derived position of *W. wattsi* with the other Australian taxa (Fig. S1), we propose that *W. wattsi* should be grouped with the three other Winton Formation taxa, within Titanosauria. Refinement of the characters and scoring of the Australian taxa for each of the three separate phylogenetic assessments (*D'Emic, 2012*; *Gonzalez Riga et al., 2016*; *Mannion et al., 2013*) along with statistical testing of an Australian clade would test this proposal and will be undertaken as new better preserved specimens come to light.

Recent support for a clade containing *D. matildae* and *S. elliottorum* has been advocated (*Poropat et al., 2021*; *Poropat et al., 2020*) with the proposed clade name 'Diamantinasauria'. As we have demonstrated above, *A. cooperensis* and *W. wattsi* also show similarities to each other and with *D. matildae* and *S. elliottorum*. All four possess a mosaic of features with some possibly uniting them all in a single clade. We, therefore, expand a hypothesised Australian clade to include all four taxa.

Our preliminary computational phylogenetic assessment provides some support for a hypothesis of a common ancestry for all four Australian taxa (Supplementary Information). Our assessment of the phylogenetic position of *A. cooperensis*, using the datasets and protocols from two recent analyses (*Poropat et al., 2021*; *Royo-Torres et al., 2020*), along with various a priori exclusions of characters and taxa, supports a clade containing at least *Australotitan*, *Diamantinasaurus* and *Savannasaurus* (Figs. S1A, S1C, S1D, S1E, S1G, S1H, S1L, S1M & S1N). The position of *Wintonotitan* was variably resolved as either basal to the 'Australian' clade (Figs. S1C, S1E & S1G) or derived within the 'Australian' clade (Figs. S1D, S1H, S1I, S1L, S1M & S1N). Some assessments did not resolve the topology of the Australian taxa at all, resulting in large polytomies with no support (Figs. S1B, S1F, S1J & S1K). A priori weighting of characters resolved more clades compared to unweighted analyses; however, it did not impact the overall membership of the Australia clade, or those non-Australian (e.g., Asian or South American) clades or lineages associated with these taxa. Of the fourteen topologies produced from our assessment, only half (seven) resolved the positions of all four Australian taxa. Of these, four resolved a clade containing all four Australian taxa within Titanosauria, and three did not. Therefore, the computational phylogenetic assessment does not unequivocally resolve the positions of all of the Australian taxa, or specifically the position of *W. wattsi*.

Of note in these results is the relative placement of taxa from South America and Asia to those from Australia (Fig. S1). 'Diamantinasauria' has recently been proposed to name a clade that includes two Australian taxa (*Diamantinasaurus* and *Savannasaurus*) and one South American taxon *Sarmientosaurus*, to the exclusion of *Wintonotitan* (*Poropat et al., 2021*). The implication of this clade was to conclude that it supports biogeographic interchange between Australia and South America (*Poropat et al., 2021*). In their assessment, 'Diamantinasauria' sits nested between an Asian sister-clade comprised of *Dongyangosaurus* and *Boatianmansaurus*, and a derived Asian clade comprised of *Xianshanosaurus* and *Daxiatitan*. Our assessment used their phylogenetic dataset, but

added *A. cooperensis* and some changes to the character-state scores for *Wintontitan*, *Savannasaurus* and *Diamantinasaurus*. The resulting topology is not altered from their assessment, other than to add *Australotitan* into the proposed 'Diamantinasauria' clade. This would lend support to an Australian-South American clade to the exclusion of *Wintonotitan*.

Contrary to this, our resulting strict consensus tree, based on the phylogenetic dataset of *Royo-Torres et al. (2020)* retains a similar topology that includes the Asian taxa of *Dongyangosaurus* and *Boatianmansaurus* within a clade including *Diamantinasaurus*. This results in an 'Australian-Asian' clade to the exclusion of South American taxa, which is contrary to the assessment above. Intriguingly, this 'Australian-Asian' clade is nested between the South American sister taxon, *Rinconsaurus*, and the closest derived taxon, the South American *Muyelensaurus*. This essentially describes the mirror opposite of the result above.

In addition, our resulting strict consensus tree moved *Savannasaurus* into this 'Australian-Asian' clade, along with *Australotitan* and *Wintonotitan* (Figs. S1D, S1H, S1L, S1M, S1N). No South American taxa were recovered within this 'Australian-Asian' clade. Similar results were returned when we used only appendicular characters, or when we excluded taxa younger than Turonian in age (Figs. S1I, S1J, S1L, S1M, S1N).

In summary, our use of two different datasets, that were initially based from the same original character sets (*Mannion et al., 2013*; *Mannion et al., 2019b*) returned some support for an Australian clade, comprising either all four or at least three of the four Australian taxa. Our results retain the non-Australian membership associated with the 'Australian clade' for each of the assessments, creating two potentially opposing phylogenetic hypotheses: (1) An 'Australian-South American' clade that is nested between Asian lineages; and (2) An 'Australian-Asian' clade nested between South American lineages. The conclusions drawn from these resultant hypotheses could argue for either faunal interchange between Australia and South America, or between Australia and Asia, with ancestral and descendant lineages occuring in either South America, Asia or Australia.

The caveates of both assessments include poor within-clade and between-clade resolution, and most importantly, limited statistical support. However, these opposing phylogenetic topologies could be reconciled if dispersal between all three continents, via Australia, occurred, thus allowing the opportunity for the presence of related taxa from all three regions occuring in Australia during the Early to mid-Cretaceous.

Firstly, faunal interchange between South America and Australia, hypothesised to have occurred via Antarctica, evokes long distance terrestrial dispersal, possibly during a period of mid-Cretaceous global warming (*Poropat et al., 2016*). Faunal interchange between Asia and Australia evokes long distance oceanic dispersal, which at face value seems unlikely. However, recent analyses of terrestrial vertebrates demonstrates that long-distance dispersal over oceans is possible and can occur upwards of 100s to 1000s of kilometers between landmasses (*Blom et al., 2019*; *de Queiroz, 2005*; *Gerlach, Muir & Richmond, 2006*; *Hawlitschek, Ramirez Garrido & Glaw, 2017*). In addition to these modern examples of

faunal oceanic interchange, dinosaurs, including titanosaurians, have recently been proposed to have dispersed across oceanic barriers (*Longrich et al., 2021*).

During the Early to mid-Cretaceous, the significant distance between the Australian continental landmass and that of Asia seems an unlikely source of faunal interchange. However, recent geological evidence with tectonic and palaeogeographic modelling has advanced the presence of a number of intra-oceanic terranes and island arc provinces within the Neo and Meso-Tethys regions, occuring between Australia and Asia during the Early to mid-Cretaceous. Potential oceanic 'stepping stones' include the East Java–West Suluwesi and the Sikuleh and Natal continental fragments, the Sepik Terrane, a proto-Philippine Sea Plate oceanic island arc and the Incertus and Woyla arcs (*Deng et al., 2020*; *Dimalanta et al., 2020*; *Hall, 2012*; *Rodrigo et al., 2020*; *Zahirovic et al., 2016*).

We speculate that if such oceanic regions had associated subaerial islands, they might have provided enough terrain to allow oceanic interchange between Asia and Australia for the largest terrestrial vertebrates of that time, the titanosaurians. Speculative conclusions, such as those proposed here, look to reconcile conflicting phylogenetic hypotheses; however, such conflicts in phylogenetic results more likely reiterate the lack of refined character signals within titanosaurian phylogenetics. Therefore, until much more refined phylogenies are developed, biogeographical hypotheses will remain equivocal.

## Body-size and palaeoenvironment of sauropods in the Winton Formation

Regardless of their phylogenetic relationship, the presence of four recognized sauropod taxa within the Winton Formation is not unsurprising considering the diversity of sauropod taxa from similar ages and latitudes (*de Jesus Faria et al., 2015*). In South America, seven to nine sauropod taxa are known from the Cenomanian of Argentina, covering a geographical range of approximately 700–1,000 km, similar to that between the northern and southern-central Winton Formation. However, proposing a framework of explanations for the diversity of the sauropods from the Winton Formation is still needed.

Firstly, there is a large difference in maximal limb element size between taxa from the northern and southern-central Winton Formation (Figs. 34 & 35). Secondly, the relative proportions of these limb elements, as a proxy of body-height, differ when also considering pelvic width, as a proxy of body-width. Thirdly, each taxon possesses a combination of features of each preserved limb that seems contrary to what would be expected.

The appendicular elements of the holotype of *A. cooperensis*, in particular the humerus, ulna and femur, represent the largest appendicular bones so far recovered of any described Australian dinosaur (Figs. 34–36) (Tables 2–7 and 10). In addition, the referred fragmentary femur, EMF164, represents an even larger individual (Table 10).

An unassigned isolated large sauropod femur (QMF43302 from QML1333) represents the largest sauropod appendicular element from the northern Winton Formation (Figs. 25J–25O, 26 & 35). This femur is separated into three sections, including a proximal femoral head, a mediolaterally-crushed and fragmented diaphysis, and a partial distal epiphysis that is missing the distal condyles. Preserved plant debris cover the broken and missing pieces of the proximal and distal epiphyses indicating that this specimen

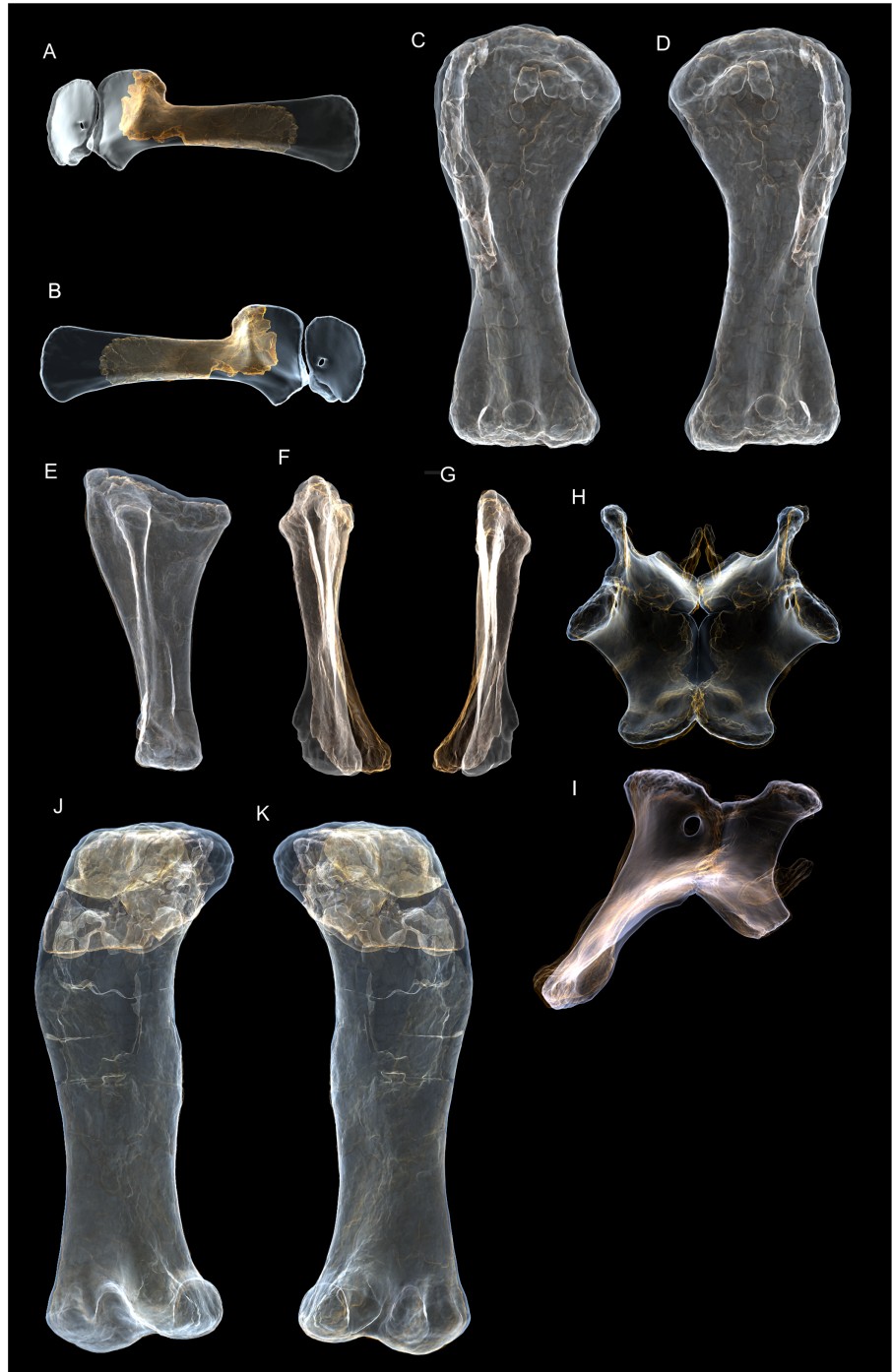

**Figure 34** 3-D digital model restorations of the appendicular elements of *Australotitan cooperensis* **gen. et sp. nov. holotype EMF102.** (A & B) Scapula in lateral (A) and medial views (B). (C & D) Humerus in anterior (C) and posterior (D) views. E–G. Ulna in anterolateral (E), posterior (F) and anteromedial (G) views. H & I. Pubes and ischia in dorsal (H) and lateral (I) views. J & K. Femur in posterior (J) and anterior (K) views. 3-D image rendering method was x-ray overlay of aligned 3-D models in orthogonal view.

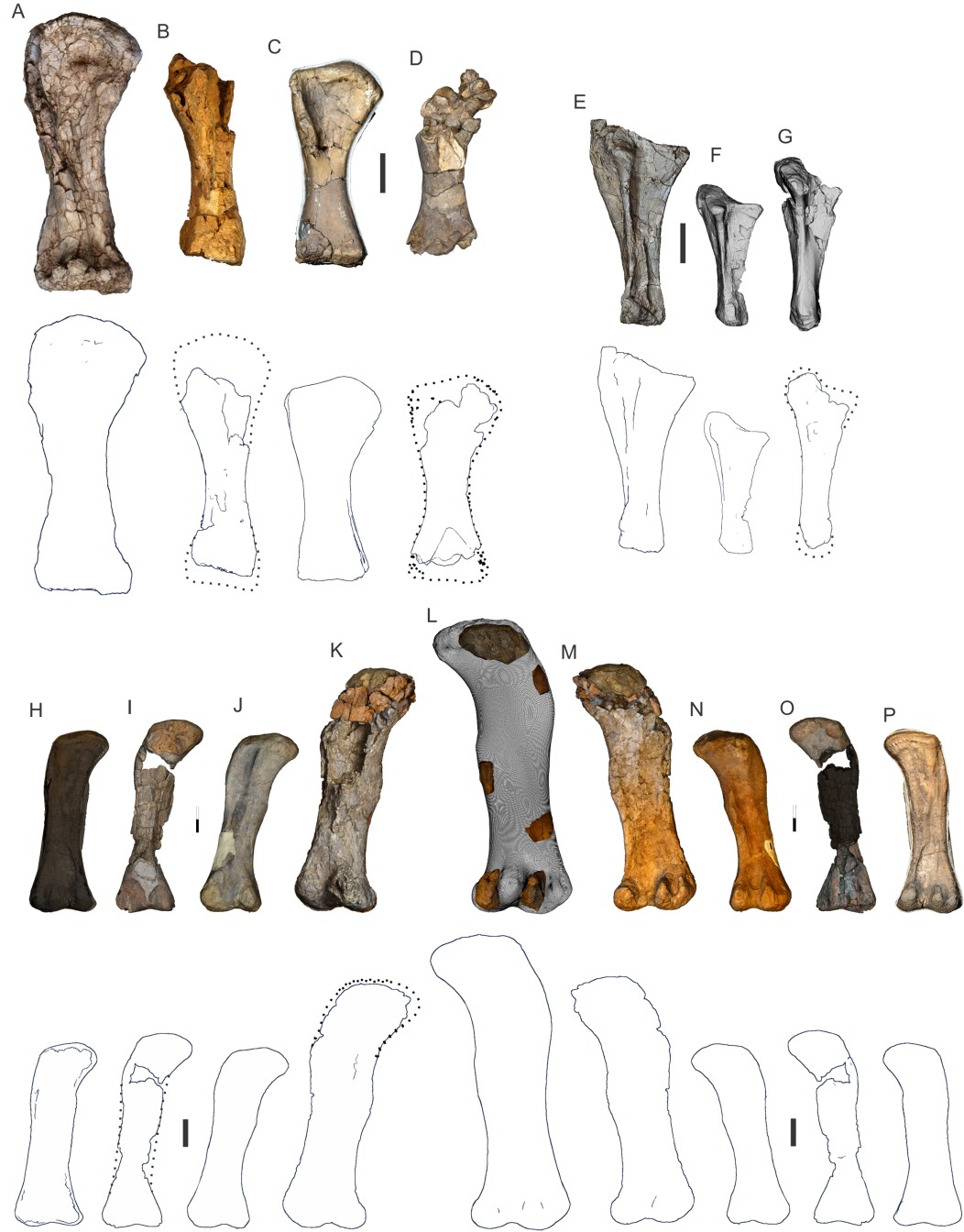

**Figure 35 Comparison of preserved size, estimated size, and shape in Winton Formation sauropod humeri, ulnae and femora (rendered as right elements).** (A–D) Humeri in anterior view; (A) *A. cooperensis* gen. et sp. nov. (B) *W. wattsi*, (C) *D. matildae* and (D) *S. elliottorum*. (E & F) Ulnae in anterolateral view; (E) *A. cooperensis* gen. et sp. nov.; (F) *D. matildae*, (G) *W. wattsi* (reconstruction). (H–K) Femora in anterior view; (H) *D. matildae*, (I) ?*W. wattsi* (QMF43302), (J) *A. cooperensis* gen. et sp. nov. (EMF105), (K) *A. cooperensis* gen. et sp. nov. (reconstructed, EMF102). (L–P) Femora in posterior view; (L) *A. cooperensis* gen. et sp. nov. (EMF164) femoral pieces set within a reconstructed outline model (transparent) (M) *A. cooperensis* gen. et sp. nov. (reconstruction, EMF102), (N) *A. cooperensis* gen. et sp. nov. (EMF105), (O) ?*W. wattsi* (QMF43302) and (P) *D. matildae*. Top rows are all natural vertex colour renders and bottom row are all orthogonal edge detected outlines. Dotted lines indicate estimated missing regions for incomplete specimens. Scale bar = 20 cm.

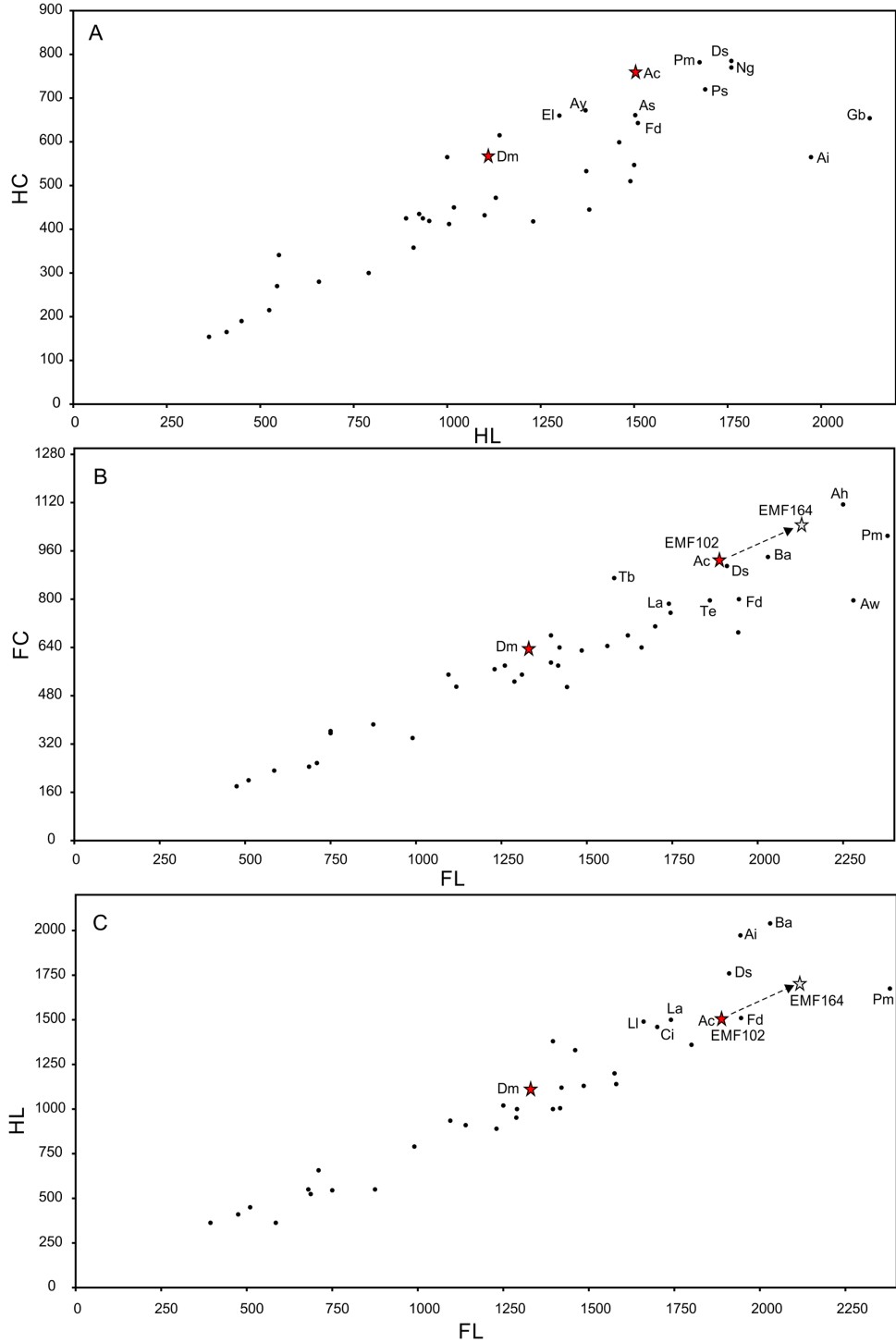

**Figure 36 Scatterplots of stylopodial measurements (mm).** (A) Humerus length (HL) plotted against humerus circumference (HC). (B) Femoral length (FL) plotted against femoral circumference (FC). (C) Femoral length (FL) plotted against humeral length (HL). Red stars indicate positions of holotype specimens of *D. matildae* (Dm) and *Australotitan cooperensis* gen. et sp. nov. (Ac), with the grey star representing the estimated position for *A. cooperensis* gen. et sp. nov. referred femur EMF164.

**Figure 36** (continued)

Abbreviations of sauropod taxa: Ah, *Argentinosaurus huiculensis*; Ai, *Atlasaurus imelakei*; As, *Alamosaurus sanjuanensis*; Ay, *Argyrosaurus superbus*; Aw, *Antarctosaurus wichmannianus*; Ba, *Brachiosaurs altithorax*; Ci, *Chubutisaurus insignis*; Ds, *Dreadnoughtus schrani*; El, *Elaltitan lilloi*; Fd, *Futalognkosaurus dukei*; Gb, *Giraffatitan brancai*; La, *Lourinhasaurus alenquerensis*; Ll, *Ligabuesaurus leanzai*; Ng, *Notocolossus gonzalezparejasi*; Ps, *Paralititan stromeri*; Pm, *Patagotitan mayorum*; Tb, *Tehuelchesaurus benitezii*; Te, *Traukutitan eocaudata*. Measurement data from *Benson et al. (2014)*

**Table 10  Maximum appendicular bone lengths for Australian sauropod taxa.**

| Taxon | Specimen | Humerus | Ulna | Femur |
|---|---|---|---|---|
| *Diamantinasaurus matildae* | AODF603 | 1,122 mm[pres] | 728 mm[pres] | 1,358 mm[pres] |
| *Wintonotitan wattsi* | QMF7292 | 924 mm[recon pres] 1,253 mm[recon est] | 897 mm[recon pres] 919 mm[recon est] | |
| *Wintonotitan wattsi?* | QMF43302 | | | 1,505 mm[pres] 1,600 mm[est] |
| *Savannasaurus elliottorum* | AODF660 | 1,020 mm[recon pres] 1,112[recon est] | | |
| *Australotitan cooperensis* | EMF102 | 1,494 mm[pres] | 1,044 mm[pres] | 1,886 mm[recon pres] 1,888 mm[recon est] |
| *Australotitan cooperensis* (referred) | EMF164 | | | 2,146 mm[est] |

**Note:**

Abbreviations; pres, as preserved; est, estimated; recon, as reconstructed.

underwent considerable transport and abrasion prior to burial and exposure. The distal condyles were broken off and lost prior to burial, whilst the proximal head was damaged, which removed 10–20 mm of cortical bone from around the proximal articular region of the greater trochanter to the femoral head. QMF43302 measures 1,505.68 mm in preserved proximodistal length, and we estimate that with the missing regions added, this would make a total length of approximately 1,600 mm (Table 7). This is approximately 250+ mm shorter than the reconstructed length of EMF102 and approximately 450+ mm shorter than the estimated length of EMF164 (Table 10).

Proximally, the femoral head is proportionately more robust than the femora seen in *A. cooperensis*, but similar to that seen in *D. matildae*. The anterior face of the diaphysis is heavily broken up into mosaic pieces, which obscures the identification of a longitudinal ridge on the anterior face of the shaft (linea intermuscularis cranialis), which would assist in referring the femur to *A. cooperensis* or *D. matildae*. Close inspection of the diaphyseal surface suggests that there is no sign of a ridge, which would then ally the femur closest to *D. matildae*, noting that the femur of *W. wattsi* and *S. elliottorum* are currently unknown. A partial sauropod skeleton (AODF836), referred to *D. matildae* (*Poropat et al., 2016*), was found 250 m to the northwest of QMF43302 (QML1333). This sauropod skeleton does not possess a femur, however, there is no evidence to demonstrate that these remains are associated with the QMF43302 femur.

When the holotype femur of *D. matildae* is compared to QMF43302 it shares the straight and narrow diaphyseal shaft and bulbous proximal head, in contrast to

*A. cooperensis* (Figs. 35H, 35I, 35O & 35P). However, when isometrically scaled to equal the minimum mediolateral width of *D. matildae*, the femoral outline of QMF43302 is proportionately taller (Fig. 30Q). Therefore, although QMF43302 is morphologically most similar to *D. matildae* in comparison to all of the southern-central Winton Formation femora described here, it remains morphologically distinct.

When considering the two other possible candidate taxa that QMF43302 could be assigned to, *W. wattsi* and *S. elliottorum*, either have preserved femora. *W. wattsi* possesses a proportionately gracile and long humerus (Figs. 15, 30, 35B) and this may reflect a much larger body-size, similar to that of the femur. *S. elliottorum* and *D. matildae* both have proportionately stocky and robust humeri than *W. wattsi*. *W. wattsi* represents the largest named sauropod taxon from the northern Winton Formation, based on limb element and ischial size. Therefore, it is conceivable that QMF43302 represents a femur of *W. wattsi*. If true, this assignment would place *W. wattsi* close to the remains of a specimen referred to *D. matildae*, albeit not directly associated with its skeleton at QML1333.

When comparing linear measurements (preserved, reconstructed and estimated) of all of the appendicular elements for all four Cretaceous Australian taxa, *A. cooperensis* has the longest scapula, humerus, ulna, pubis and femur (Table 10). Although the ischium of *A. cooperensis* is the largest ischium based on preserved length, the ischium of *W. wattsi* holotype is near its size with a thicker blade along its preserved length. *W. wattsi* is missing the proximal articular end of the iliac peduncle, acetabular rim, and the mediodistal margin of the ischial symphysis, therefore, depending on how much of the ischium is missing, *W. wattsi* could have an ischium of the same size, if not marginally larger, than *A. cooperensis*.

The humeri and ulnae of *W. wattsi* are poorly preserved, with all elements missing either both epiphyses or when preserved, missing most of the articular surfaces. This means that the longest linear proximodistal length for these elements are underestimates of the length of the bones. Using the same better-preserved elements in *D. matildae* as a guide, we were able to align and scale the 3-D model of *D. matildae* limb elements to that of *W. wattsi* to provide a prediction of length. The humerus of *W. wattsi* returned an estimated proximodistal length of 1,253 mm whilst the ulna was estimated to measure 919 mm. The longest preserved length of ulnae is 897 mm, some 22 mm shorter than the estimate; therefore, we suspect about 20–50 mm of length has been lost of the proximal and distal epiphyses.

We also estimated the length of the humerus from *S. elliottorum* by isometrically scaling the complete 3-D model of the humerus of *D. matildae* to the preserved humerus shape of *S. elliottorum*, to return an estimated maximum length of 1,112 mm. *S. elliottorum* does not preserve an ulna or femur so cannot be compared to these appendicular elements.

Considering the sizes of the best comparable elements across the four taxa in relation to columnar limb elements (i.e., humerus, ulna and femur), *A. cooperensis* represents overall the largest taxon, but more specifically the taxon with the longest limbs (Table 10). *W. wattsi* was second tallest, whilst *D. matildae* and *S. elliottorum* had the shortest limbs and most robust stature.

When comparing the overall pelvic floor between each taxon, as a proxy of body-width, it is evident that *A. cooperensis* had the deepest and widest pelvis in absolute size (Fig. 22) (Tables 5 & 6). We cannot reconstruct the pelvis of *W. wattsi* because it is missing the pubes and the medial most portion of the ischial contact. However, the ischium is so close in size and similar in morphology to both *A. cooperensis* and *D. matildae* (Figs. 30M–30O) that we would expect the pelvic floor to be proportionately as deep as both of these taxa, and impressively, as large and as wide as that of *A. cooperensis*. *S. elliottorum* shows a relatively broader and shallower pelvis (Figs. 22 & 30). Although this feature looks to be a real and unique feature of *S. elliottorum*, there are some areas at, and below, the position of the iliac peduncles of both the pubis and ischium that may reflect vertical taphonomic compression. If so, this compression would artificially reduce the pelvic floor depth creating what would seem to be a shallow appearance in anterior or posterior views (Fig. 22). Large dorsal vertebrae from the skeleton were found directly above the puboischial complex, and the humerus and ribs also show signs of directional crushing and distortion. Therefore, taphonomic alteration via trampling is possible thus altering the pelvic dorsoventral profile.

Each limb segment for the four taxa present unexpected combinations that do not intuitively correspond with one another, nor can they be easily considered part of a morphocline. *A. cooperensis* is clearly the largest taxon; however, it both possesses the most lightly built and gracile scapula, ulna and puboischial complex, but with massive and solidly built humeri and femora. *W. wattsi* is the second largest taxon with the most solidly built scapulae and ischia, and most robust ulnae in midshaft cross-section, but the least rotund humeri. *D. matildae* and *S. elliottorum* both possess equally stocky humeri and *D. matildae* the stockiest ulnae. However, *S. elliottorum* possesses a very broad, shallow and lightly built, but completely fused puboischial complex.

This somewhat contrary mosaic of characteristics for each taxon impedes explanations of adaptative ecology or as part of a morphocline. Whether or not these features represent adaptations of body-size, sexual dimorphism, locomotion, habitat (terrestrial versus semi-aquatic) and/or feeding strategies are all areas of potential explanation, but are all equally confounded by a lack of phylogenetic, temporal and environmental resolution. Simplistic explanations using modern ecological analogies cannot be argued for any of the Winton Formation sauropod taxa without a detailed understanding of the environmental context in which each taxon lived, which is severely lacking at present.

The very poor stratigraphic and temporal context of the Australian sauropod type localities as discussed above means that we cannot easily explain the taxonomic diversity in a temporal context. Based on our current understanding of the relative stratigraphic positions of the sauropod taxa within the Winton Formation we propose that *D. matildae* occurs within 100 m of the Winton Formation base, as represented by AODF836, to up to at least 350 m from the base, as represented by the type specimen AODF604. Similarly, *W. wattsi* occurs within 100 m of the Winton Formation base on the tentative identification of a single poorly preserved femur, QMF43302, up to at least 350 m, as represented by the type specimen, QMF7292. Together, this suggests that these two taxa co-occurred throughout the basal 350 m of the northern Winton Formation.

*S. elliottorum* is only known from the type specimen AODF660 which sits within 100 m of the northern Winton Formation base, whilst *A. cooperensis* is only known from sites that occur between 270–300 m of the southern-central Winton Formation base. It is therefore unlikely that all four taxa represent a single chronocline, with some tenuous evidence for three taxa co-occurring during the deposition of the basal 100 m of the northern Winton Formation. However, there is no definitive evidence demonstrating that any of these taxa were sympatric, with no single site demonstrably showing more than one taxon in a single bonebed. Therefore, we cannot definitively place these taxa together with each other at any singular place or time.

The distinctive taphonomic differences observed between sites in the northern and southern-central Winton Formation may provide some clues to palaeoenvironmental differences that could have created enough difference in habitat to select for varying types of megaherbivorous sauropods. The absence of abundant or diverse aquatic fauna, in particular, freshwater insect larvae, freshwater bivalves and snails, crustaceans, fish, lungfish and crocodilians along with the presence of scoured and highly trampled silty-muddy surfaces absent of developed palaeosols, suggests a highly labile sedimentary and turbid aquatic environment in the southern-central Winton Formation sites, compared to the northern Winton Formation sites. These observed differences could be geochronological, but note the caution we discuss above. If geochronological, the differences could represent a succession of palaeoenvironmental changes as the basin fills, with the reduction of topographic relief and development of new freshwater environments with areas likely terraformed by the largest of the sauropod taxa. If the sites are contemporaneous, then these differences could be due to regional hydroclimatic differences, perhaps relating to the distance of the southern-central Winton Formation environments from the topographically higher watershed to the east.

The greater diversity of flora and aquatic fauna in the northern Winton Formation points to a less turbid and more stable habitat with a greater diversity of vegetation both in terms of taxa and structure. The proximity of the northern Winton Formation sites to a greater diversity of older terrestrial and stable terrain provides another source of geographical diversity that would have likely been a source of biological diversity proximal to the northern Winton Formation but distal to the southern-central Winton Formation sites (*Harrington et al., 2019*).

We speculate that a spatiotemporal ecocline developed from east to west, from the eastern basin periphery and drainage topographic high to the center and topographic low. The basin rapidly filled with volcanoclastic input from the east and transitioned from low terrestrial vegetation productivity (e.g., shallow/coastal marine habitats) to highly productive habitats (e.g., paralic to fluvial and lacustrine environments). Such labile and frequently disturbed environments were likely further disturbed by the sauropods themselves, and this was set within a backdrop of variable or seasonal local climate (*Fletcher, Moss & Salisbury, 2018*) and major mid-Cretaceous global climatic fluctuations (*Hay, 2011*) associated with volcanism (*Percival et al., 2020*). Of note, a combination of frequent disturbance with climatic variability and instability has been proposed as a

mechanism that maintained megaherbivore diversity of Quaternary megafauna (*Mann et al., 2018*).

### Body-size of *Australotitan cooperensis* relative to other giant titanosaurians

It is tempting to produce an estimate of body mass for *A. cooperensis* based on the preserved and reconstructed stylopodial circumferences and using formulae previously developed; however, due to the considerable uncertainty surrounding these formulae for estimating body mass, as discussed above in the Methods, we will not undertake this estimate. Instead, we can simply use limb-size alone as a way to compare the size of *A. cooperensis* to other sauropods globally. This is useful because *A. cooperensis* represents the first osteological evidence of a very large titanosaurian in Australia, of comparable size to taxa from other parts of the Gondwanan supercontinent (*Otero et al., 2021*).

Humerus and femur lengths, along with humerus and femur circumferences from known taxa were plotted against the type specimen of *A. cooperensis* (EMF102) to see where this new Australian taxon falls in regards to the largest sauropods known from femora and humeri (Fig. 36). In a comparison of humerus length with circumference (Fig. 36A), *A. cooperensis* clusters with *Dr. schrani*, *P. mayorum*, *Pa. stromeri* and *No. gonzalezparejasi*. In comparison of femoral length with femoral circumference (Fig. 36B), *A. cooperensis* clusters with *Dr. schrani* and *Brachiosaurus altithorax*. In comparison of femoral length with humeral length (Fig. 36C), *A. cooperensis* clusters with *Futalognkosaurus dukei*. Considering the larger referred femur (EMF164), our estimated femur length of this individual 2,146 mm, which would confirm the limb element size of *A. cooperensis* close to *Dr. schrani* and *F. dukei*, but smaller than *P. mayorum*. Body mass estimates for these two titanosaurians vary considerably, from a minimum estimate for *F. dukei* of 23,601 kg to a maximum estimate for *Dr. schrani* of 74,487 kg (*Campione & Evans, 2020*). This reflects the uncertainty discussed above and thus demonstrates the issues relating to body mass estimation in extremely large tetrapods.

## CONCLUSIONS

A new dinosaurian fossil field from the southern-central Winton Formation (Eromanga Basin) has yielded a new giant titanosaurian sauropod, *Australotitan cooperensis*. It represents the largest dinosaur yet known from osteological remains in Australia and confirms the presence of gigantic titanosaurian sauropods in eastern Gondwana during the mid-Cretaceous. The currently described Winton Formation sauropod taxa share with titanosaurians from across the globe a highly fragmentary nature, which creates considerable ambiguity when searching for well-supported phylogenetic placements for each taxon, or providing useful explanations for morphological and taxonomic diversity, along with inferred palaeobiogeography.

The creation of 3-D surface models from specimens has allowed the development of a coloured schematic as a new method for annotating directly onto the bones where features are not easily distinguished. In addition, the use of a range of 3-D alignment and rendering modes offers better geometric comparison whilst allowing the identification of taphonomic

biases. These interpretations of taphonomic alteration and preservation are essential for successive morphological interpretations. Therefore, they need to be captured and communicated in 3-D on the digital models created. This will also allow these interpretations to be tested, re-interpreted, and new versions to be published in subsequent research. We see this method as providing a pathway to share all forms of interpretation undertaken on specimens within the context of a 3-D geometric cybertype of the original.

In a comparative approach, we used previously identified synapomorphic features of the appendicular skeleton and found that all four taxa could be classified as members of the Titanosauria and possibly as basal members of it, or as basal lithostrotians. Focusing on the shared preserved elements for the Winton Formation taxa, we found a mosaic of characteristics that differentiate them from each other and from taxa elsewhere. We also find a mosaic of appendicular features that are shared across titanosaurians of similar size or semi-contemporaneous age, indicating that the appendicular skeleton is useful for taxonomic differentiation, but perhaps not as useful in reconciling greater phylogenetic resolution.

Other characteristics that are shared between the Winton Formation sauropod taxa; such as the shared morphology of the ischium in *A. cooperensis*, *D. matildae* and *W. wattsi*; shared pneumatic anterior caudal vertebrae in *S. elliottorum* and *W. wattsi*; and ubiquitous presence of amphicoelous caudal vertebrae from described and undescribed specimens allude to a shared common ancestry for all of the Winton Formation taxa. We, therefore, propose a hypothesis of common ancestry for all four taxa that diversified in Australia during the mid-Cretaceous. Our preliminary phylogenetic analyses provide some support for this hypothesis by finding resulting parsimonious hypotheses that include all four taxa within a clade. Such results support a recent naming of an Australian clade, the 'Diamantinasauria'. However, our assessments find conflict as to which non-Australian taxa are also shared within 'Diamantinasauria', with separate analyses supporting either South American or Asian taxa. Therefore, whether Diamantinasauria represents a stable clade remains to be seen.

Considering that the Australian taxa might represent a single lineage or clade, we further speculate that the Australian clade could represent an adaptive response to new, rapidly changing environments developing across the Eromanga Basin during the deposition of the Winton Formation. As the basin filled, it would have transformed from an eperic epicontinental sea to complex paralic environments, through to vast, labile and frequently disturbed alluvial and lacustrine habitats. We speculate that such new and rapidly developing habitats drove the evolution of morphological diversity within the largest herbivores, the titanosaurians, as new opportunities appeared across the landscape. Alternatively, the taxa may reflect a complex morphocline or ecocline across variable environments already developed across the basin during the Cenomanian. We cannot completely rule out the presence of a species chronocline based on the current stratigraphic or chronological uncertainty of the identified sauropod taxa so far found. All explanations remain equivocal due to poor local and regional chronostratigraphic resolution we have demonstrated here. Notably, no Winton Formation sauropod taxa are verifiably sympatric.

Future research should focus on building greater detail of the local stratigraphic and palaeoenvironmental context, for both previous and new sites, because until this is achieved, phylogenetic position alone will be of limited interpretative value in the evolution of Australia's largest terrestrial vertebrates.

## INSTITUTIONAL ABBREVIATIONS

| | |
|---|---|
| **AODF** | Australian Age of Dinosaurs Museum of Natural History Fossil |
| **AODL** | Australian Age of Dinosaurs Museum of Natural History Locality |
| **EMF** | Eromanga Natural History Museum Fossil |
| **EML** | Eromanga Natural History Museum Locality |
| **QMF** | Queensland Museum Fossil |
| **QML** | Queensland Museum Locality |

## ACKNOWLEDGEMENTS

We acknowledge and pay respect to the Wangkumura and Boonthamurra People on whose traditional lands these dinosaurs were discovered. We thank Wangkumura elder, Malcolm Ebsworth for his assistance and guidance during the fieldwork on Plevna Downs Station. We would like to acknowledge those who first contributed to this work, both in the field and laboratory, including, Joanne Wilkinson, Kristen Spring, Elizabeth Cannon, Jo Pegler, Scott Turner, Alex Cook, Ralph Molnar, Paul Sereno and the Mackenzie family, in particular Sandy (jnr) Mackenzie who found the first dinosaur bone.

We thank all of the volunteers and supporters of the Queensland Museum and Eromanga Natural History Museum who have made significant contributions in the field and laboratory: Jim Macmillan, Maxine Macmillan, Stephen Tully, Annabel Tully, Tom Meakin, Janine Meakin, Scott Pegler, Denise O'Boyle, Jill Corrigan, June Gunn, Doug Miller, June Richardson, Joan Rasmussen, Angelica Wilson, Maria Zammitt, Graham Wilson, Clare Steele, Phil Wharton, Ursula Wharton, Jacki Erickson, Corey Richards, Laurie Beirne, Liz Towns, Pam Towns, Keith McGlashin, Pat Turner, Geoff Turner, Nan Mackenzie, Sandy (snr) Mackenzie, Jonathan Cramb, Susan Rigby, Noel Cannon, Wendy Groves, Kimberley Smith, Tanya Hudson, Louise McGowan, the Skinner family, and the Eromanga & Quilpie Communities. We thank Nikki Newman and Queensland X-Ray for CT scanning the specimens described here. We thank Kristen Spring (QM), Trish Sloan and David Elliott (AAOD) for access to specimens for comparative purposes. SAH thanks Adamm Yates for his assistance with Winton Formation fossils from the Northern Territory. We thank Ralph Molnar, E. Martin Hechenleitner and a third anonymous reviewer for their useful insights that have subsequently improved this work. We would like to thank Doug Boyer and Mackenzie A. Shepard of morphosource for their assistance.

### Funding

Field work, preparation, digital capture and processing was supported by Eromanga Natural History Museum, Outback Gondwana Foundation, Santos, Eromanga Earth

Moving, Bill Pegler, Eromanga Contracting, IOR and Eagle Gallery, Queensland Museum, Queensland Museum Foundation, Project DIG and ARC Linkage Grant LP100100339. The funders had no role in study design, data collection and analysis, decision to publish, or preparation of the manuscript.

### Grant Disclosures

The following grant information was disclosed by the authors:
Eromanga Natural History Museum, Outback Gondwana Foundation, Santos, Eromanga Earth Moving, Bill Pegler, Eromanga Contracting, IOR and Eagle Gallery, Queensland Museum, Queensland Museum Foundation, Project DIG and ARC Linkage: LP100100339.

### Competing Interests

The authors declare that they have no competing interests.

### Author Contributions

- Scott A Hocknull conceived and designed the experiments, performed the experiments, analyzed the data, prepared figures and/or tables, authored or reviewed drafts of the paper, and approved the final draft.
- Melville Wilkinson conceived and designed the experiments, analyzed the data, authored or reviewed drafts of the paper, and approved the final draft.
- Rochelle A. Lawrence conceived and designed the experiments, performed the experiments, analyzed the data, prepared figures and/or tables, authored or reviewed drafts of the paper, and approved the final draft.
- Vladislav Konstantinov conceived and designed the experiments, performed the experiments, prepared figures and/or tables, and approved the final draft.
- Stuart Mackenzie conceived and designed the experiments, authored or reviewed drafts of the paper, and approved the final draft.
- Robyn Mackenzie conceived and designed the experiments, performed the experiments, authored or reviewed drafts of the paper, and approved the final draft.

### Data Availability

Fossil Specimens directly studied with their accession numbers and location. The 3-D models are available at Morphosource. The 69 minted DOIs for those models are available in the Supplemental File.

Specimen List

EMF 102, Australotitan cooperensis Holotype, Eromanga Natural History Museum, Eromanga, Queensland, Australia.

EMF 105, Australotitan cooperensis (referred), Eromanga Natural History Museum, Eromanga, Queensland, Australia.

EMF 165, Australotitan cooperensis (referred), Eromanga Natural History Museum, Eromanga, Queensland, Australia.

EMF 100, ulna, Eromanga Natural History Museum, Eromanga, Queensland, Australia.

EMF 106, caudal vertebra, Eromanga Natural History Museum, Eromanga, Queensland, Australia.

EMF 109, partial sauropod skeleton with caudal vertebrae, Eromanga Natural History Museum, Eromanga, Queensland, Australia.

EMF 171, caudal vertebra, Eromanga Natural History Museum, Eromanga, Queensland, Australia.

EMF172, Pterophyte leaf, Eromanga Natural History Museum, Eromanga, Queensland, Australia.

EMF173, conifer leaf (mat), Eromanga Natural History Museum, Eromanga, Queensland, Australia.

EMF 174, Pterophyte leaf, Eromanga Natural History Museum, Eromanga, Queensland, Australia.

EMF 175, ?Bennettitalean leaf, Eromanga Natural History Museum, Eromanga, Queensland, Australia.

EMF 176, conifer twig, Eromanga Natural History Museum, Eromanga, Queensland, Australia.

EMF 177, conifer twig with leaves, Eromanga Natural History Museum, Eromanga, Queensland, Australia.

AODF 603, Diamantinasaurus matildae Holotype, Australian Age of Dinosaurs Museum of Natural History, Winton, Queensland, Australia.

AODF 660, Savannasaurus elliottorum Holotype, Australian Age of Dinosaurs Museum of Natural History, Winton, Queensland, Australia.

QMF7292, Wintonotitan wattsi Holotype, Queensland Museum, Brisbane, Queensland, Australia.

QMF43302, Wintonotitan wattsi?, Queensland Museum, Brisbane, Queensland, Australia.

QMF 3390, femur, Queensland Museum, Brisbane, Queensland, Australia

QMF 7291, femur, Queensland Museum, Brisbane, Queensland, Australia

## New Species Registration

The following information was supplied regarding the registration of a newly described species:

Publication LSID: urn:lsid:zoobank.org:pub:AF1FA65A-5351-45B1-B0CB-EC1225590A0F.

*Australotitan*:urn:lsid:zoobank.org:act:B91317FD-04ED-49C4-946E-6757BFBF6CA3.

*Australotitan* cooperensis: urn:lsid:zoobank.org:act:766BAD1A-3184-4486-B8AF-49551785091E.

## Supplemental Information

Supplemental information for this article can be found online at http://dx.doi.org/10.7717/peerj.11317#supplemental-information.

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
