# Peer review of "A new giant sauropod, Australotitan cooperensis gen. et sp. nov., from the mid-Cretaceous of Australia"

_PeerJ, doi:10.7717/peerj.11317_

## Round 0.1 · original submission · Major Revisions

Although the reviewers are positive, they raise a number of issues, which should be addressed.

Please, together with your unmarked revised manuscript, provide a marked-up copy as well as a document explaining how you have addressed each of the points raised by the reviewers.

·

Basic reporting

The language is clear, unambiguous & professional throughout. I have however, made some comments regarding issues that I feel could be further clarified, at the authors (or editors') discretion.
Literature references, background and context are are more than adequate.
The structure of the text, figures and tables is fine. There are minor issues with the figures (see appended document) that need to be addressed. These should be able to be rectified in a few hours, if that.
Raw data is presented in the tables; I do not know if it is also to ve=be deposited in some online source.
The ms is self-contained.
I think the text clearly meets your standards, although I have suggested minor improvements in the appended document. Even so the text could be published as it stands.

Experimental design

Not applicable: this is a ms based on observation, not experiment.

Validity of the findings

I think the potential impact & novelty are important here, since thesis probably the most significant ms in this field that I have seen, at least in several decades.
Replication in the strict sense is not feasible in vertebrate paleontology, although further field work is recommended.
The data have been provided, and since these are observational data, statistical soundness, etc. are not applicable here.
The conclusions are well-stated, clearly appropriate to the research aims & field discoveries, and well-supported.
Such speculation as is included is clearly indicated.
I do not see that this ms fails to meet your standards in any area. Which, again, doesn't prevent me from suggesting some alterations.

Additional comments

I think this is an exemplary ms, & that the authors have done a very good job. This has not prevented me from suggesting minor alterations, however. The only mandatory ones are those regarding altering the figures.

·

Basic reporting

The study of Scott Hocknull and colleagues is an exciting contribution about a new sauropod dinosaur from the Cretaceous Winton Formation, Australia. The new sauropod species is the largest yet recorded for this continent. Based on morphological comparisons, the authors suggest a common ancestry for all the Winton Formation sauropods. In addition to valuable information on the new species, the study includes a massive data set of 3D models of the new taxon as well as Wintonotitan wattsi, Diamantinasaurus matildae and Savannasaurus elliottorum. The study also includes a new labelling method with colour reference for the 3D models that helps the observation of specimens with severe taphonomic alterations.
Overall, the manuscript is well-written, in professional language. The Introduction section provides an adequate context, and the Geological Setting shows reliable data on sedimentology, stratigraphy and taphonomy of the principal localities of the Winton Formation. The structure of the main text is correct, although some aspects should be reviewed (see General Comments). The figures are nice looking and have good quality. However, the large number of bone images per plate sometimes threatens clarity, and one must permanently resort to the figure captions to check. It would be much appreciated if the authors increase the labelling of figures, pointing out anatomical characters that are recurrent in the description and comparison. In the Results section, the reference to the figures is scarce. Specifying which part of the plates to see when referring to a particular feature in a taxon (e.g., Fig. 9D, instead of just Fig. 9) would significantly speed up the reading. There are some suggestions about the figures in the General Comments. Finally, the authors' work regarding the 3D models of the sauropods from Winton Formation is remarkable.

Experimental design

The manuscript is consistent with the scope of this journal. The main goal, which is to describe a new species of sauropod dinosaurs, is clearly stated at the end of the introduction. Although not explicit, it is easy to deduce that the new species increases the diversity of sauropods in the Winton Formation. The methodological section is rigorous and contains a large amount of information, which guarantees the reproducibility of the 3D modelling, in addition to the labelling and retrodeformation. I found some paragraphs in the Results and Discussion that could be relocated in the Methods section (see General Comments).

Validity of the findings

The discovery of a new giant sauropod is transcendent for the dinosaur record of the Winton Formation and Australia. The 3D models and their associated new labelling method are interesting contributions that improve the visualization of fossils which undergone severe cracking and deformation. As the authors well point out, the sauropods of the Winton Formation are distributed in several museums, distant from each other, which can make direct observation of the specimens difficult. Although 3D models are not a substitute for direct observation, they will be a handy tool for colleagues around the world who are interested in the taxonomy and systematics of sauropod dinosaurs.
Speculations, such as including all the Winton Formation sauropods in a single and independent clade, are interesting. However, not all of them find much support in the data presented. Perhaps some rework in the Discussion section will improve this aspect (see General Comments).

Additional comments

In general, the information in this manuscript is solid. However, there are some questions and concerns in regards to its structure that should be addressed before it may be considered for publication.

1. Length of the manuscript. Some aspects that could help with the length of the document are a) a stratigraphic column including the positions of each sauropod taxon could summarize much of the text in the Geological Setting; b) avoid repeating descriptions of the taphonomic alterations already mentioned in the Geological Setting; c) the anatomical descriptions in results could be shorter and more direct by de-emphasizing missing characters, the figures and 3D models are very clear regarding the preservation of the material; c) several comparisons in the Results section appear repeated in the Discussion. In the attached pdf file, I marked some paragraphs that could be reduced and/or relocated.

3. Referred material. Some of the referred material is fragmentary and does not present diagnostic characteristics. In particular, the section "Australotitan cooperensis?" mentions specimens that could belong to this new taxon. The specimen EMF109 is represented exclusively by remains of the axial skeleton and, according to the authors, it is still in preparation. As the holotype specimen only preserves appendicular elements, Figs. 31-34, as well as the text referring to them, would be much more useful in the context of an independent investigation based on the specimen EMF109. Something similar occurs with EMF106. The specimen EMF164 seems to be the only possible link between EMF106 and EMF109, and the holotype of A. cooperensis, as it preserves both axial and appendicular remains. However, the axial elements of EMF164 are supposed to be presacral vertebrae. Thus, as long as the authors cannot assign them to A. cooperensis, the discussion on possible similarities of EMF106 and EMF109 with other sauropods of the Winton Formation seems not relevant in the context of the present investigation.

2. Labelling in figures and reference in the main text. Figures showing skeletal elements appear in a generalized way in the main text. I suggest referring to specific sectors of each plate. Some figures are completely devoid of labels (e.g. Fig. 11, 12, 13, 14, 15, 17, 20, 21, 22, 23, 25, 29, 30). They also do not specify the orientation of the material in proximal, distal and dorsal views. Both the labelling and the orientation of the skeletal parts facilitate understanding, especially for the public who is not familiar with the anatomy of sauropods.

4. Body size. The authors emphasize throughout the text the large size of the holotype and the referred materials. As this topic is repeatedly highlighted, it would be essential to have a section in both the methods and the results section, explaining the scatterplots of stylopodial measurements in Fig. 36. Condensing size comparisons into a single section of results would facilitate the discussion. Also, the body size is evaluated at least in three times within the Discussion, in "Comparisons with other large-bodied titanosaurs", "Body size and palaeoecology of sauropods in the Winton Formation", and "Body size of Australotitan cooperensis". The authors could avoid the overlap of information by organizing several of the paragraphs in a single section.

5. Table 11. Authors should make explicit the anatomical or functional significance for a ratio of humerus length to distance between iliac peduncles. A similar case occurs with the "Minimum mediolateral width of humerus/minimum anteroposterior (dorsoventral) width of the ischial distal blade", in Table 9. This type of relationship usually has a morphofunctional basis. For instance, the ratio length of humerus vs length of the femur accounts for a change in limb proportions and, thus, a change in posture. Perhaps the features should be compared individually.

6. Format. Some minor concerns about format include:
a. Use of scientific names. The authors should use a single convention for mentioning the taxa throughout the whole manuscript. The name Australotitan is similar to Austrosaurus. Being abbreviated, these can be confused between each other or even with Andesaurus. Something similar occurs with other species whose generic name begins with the same letter, e.g. D. matildae (Diamantinasaurus), D. sinensis (Dongyangosaurus), D. schrani (Dreadgnouthus). As alternatives, the authors could use the full names or even exclusively use the generic names, something frequent in papers about sauropods, since the vast majority of their genera are monotypic. The only exception here would be Neuquensaurus.
b. References. The literature cited is up to date. I suggested a few, hopefully useful, references in the attached pdf. The citations in the word document are not in plain text. Also, there are some spelling mistakes in the reference list.
c. Figure captions. As I above said, indicating the orientation of some views in the figures or the figure captions would be helpful. Be sure to use the same criteria when mentioning consecutive letters in the figure captions. For instance, C-D in Fig. 13 appears as C & D, and C, D in the caption of Fig, 11.
c. Tables. Flags in Tables 8 and 9 are not all the same. Some like "?+flag" should be explained in the footnotes. Some features do not have data in any columns, so they should be deleted (e.g., last five rows in Table 4).
d. Other specific comments can be found in the attached pdf file.

Reviewer 3 ·

Basic reporting

The manuscript is thoroughly written and meets these criteria. I'm not sure if it was a formatting thing but my review copy had two copies of the tables and figure captions. The second set of figure captions had the accompanying figures. However, the one thing I would say about the writing is that the first half of the manuscript seemed a bit too overwritten with the details and could be more concise.

Experimental design

The manuscript fits within these areas for the most part. It does a decent job addressing the recently growing sauropod record from the Winton Formation of Australia with the description of "Cooper" and comparisons (and well-documented figures with 3D models) with the other Australian sauropods. The one thing I wish the authors did do, although they addressed and justified their reasoning, was a phylogenetic analysis of the new sauropod. There are some interesting morphologies they noted that are shared amongst the Australian sauropods and some widespread traits within Titanosauria. However, the only real test of these features (are they really a monophyletic group? where do they fit within sauropoda) is to run an analysis. Now, I do tend to agree on erring on conservative assessments with incomplete specimens, but even incomplete specimens can hold value depending on what traits are preserved and if those traits have good phylogenetic value. The authors do a good job in their comparisons, noting similarities, differences, and checking proposed clade synapomorphies from other studies with the Australian sauropods, but running a phylogenetic analysis would help assess these proposed hypotheses further—and the authors can still argue for preliminary results until better preserved specimens are found. Otherwise, the descriptions and comparison work is decently done, although it could use a few more comparisons in some sections with non-Australian dinosaurs (African sauropods such as Angolatitan and Mnyamawamtuka may be of use since these two are from roughly the same time and part of Gondwana and have comparable elements).

Validity of the findings

I tend to agree with the authors findings. Like I said in the Experimental Design part, I think a phylogenetic analysis would help further their assessment on their proposed hypotheses/conclusions. Otherwise, the figures with the fossils and models help a lot with the comparisons and the measurement tables are robust. I ask, will the 3D models be available for those interested upon publication as supplemental material?

Additional comments

Overall, this is a very thorough manuscript with some suggestions for improvements (also see attached PDF with additional comments and noted errors). I, for one, am excited for an additional Australian Cretaceous sauropod and this manuscript also delivers much detail and plenty of comparative figures with all four Australian titanosaurian taxa. To get to it, here are some of my major comments:

1. The first half of the paper is a bit overwritten. Although the geology and the context is quite important in assessing the four sauropod taxa form the Winton Formation as well as the methodology with the model creation and comparative shortcomings, I feel these sections prior to the actual description can be a bit more concise and less "in the weeds". Otherwise, the attention to detail is appreciated it can just be more direct.

2. Some of the anatomical descriptions (e.g., femur) could use a few more taxa comparisons, some of these are noted in the PDF. There are a couple features mentioned that are present in other titanosaurs so check those couple of comments in the PDF. I would suggest including the African taxa of Angolatitan and Mnyamawamtuka from the middle Cretaceous of African for comparisons outside of Australia since they preserve some elements of interest and roughly the same time period.

3. Some general points with using Mannion 2013 and D'Emic 2012 synapomorphy lists. First, the Mannion study has since gone through updates and expansion the past couple of years, so check out those for any character updates. The D'Emic study is also limited in the number of titanosaurian taxa used, so some caution in interpretation there (that study lacks many titanosaurian subclades that could be of interest here). I would check out other phylogenetic studies such as Carballido, Gorscak, and Gonzalez-Riga lines of studies too for additional synapomorphies since they include more taxa and are more recent.

4. Speaking of phylogenetics, I do tend to agree on erring on conservative assessments on incomplete specimens for phylogenetic analyses but even incomplete specimens can be useful depending on what traits are preserved and their relative phylogenetic utility. I would strongly suggest attempting a phylogenetic analysis and see what the results would be (I think one of the recent Mannion data sets include the three other Australian titanosaurs, so that would be a good one to use). You can still argue for preliminary results due to the incompleteness, but an analysis would further help assess hypotheses on monophyly and placement within titanosauriforms for these four Australian titanosaurs.

5. Finally, so many comparative figures with the Australian titanosaurs! One thing that would help out a lot would be labeling the features discussed in the manuscript onto the figures (traits of phylogenetic importance, especially autapomorphies, any similar/dissimilar traits, etc). The ulna figure has a couple of labels, but the others are lacking.

Otherwise, a thoroughly written manuscript with great (although slightly busy) comparative figures. It may be useful to take some parts of the manuscript and figures and put them into supplemental material, but not necessary... I'll leave that up to you. Best of luck, and I look forward to seeing this published in the future!

Annotated reviews are not available for download in order to protect the identity of reviewers who chose to remain anonymous.

---

## Round 0.2 · Minor Revisions

Please copy edit and proofread your text carefully (there're a number of typos: Melboune, ournal etc). There's no need to add systematically "gen. et sp. nov." to the new taxon in the "Geological Setting", but do add it in every figure and caption in which the taxon is mentioned.

Homogenise the style of your scale bars in all your figures (do not use black and white scale bars).

You might want to cite recently published papers such as that of Otero et al. (2021) in Cretaceous Research (104754).

·

Basic reporting

See general comments.

Experimental design

See general comments.

Validity of the findings

See general comments.

Additional comments

This is the second time I review the manuscript, and now it is publishable almost as is. I'm happy to see that the authors addressed most of the concerns raised in the previous round. The new labellings greatly improved the quality of the figures. I included some minor comments and suggestions in the attached pdf.

Reviewer 3 ·

Basic reporting

Overall, I see the authors met most of what was requested of the reviewers and I have no additional remarks on the basic reporting aspect of the manuscript.

Line 2363... "unit" should be "unite".

Experimental design

(1) I am pleased that the authors included phylogenetic analyses and even did a bit more than requested by testing out various scenarios with the data sets. Although, the surrounding tone about conducting the phylogenetic analysis is a bit... dismissive? Reluctant? Pessimistic? These lines stood out to me:

"Undertaking a computational phylogenetic assessment using parsimony methods for the new taxon was not considered useful due to the lack of preserved elements across diverse portions of the skeleton" (lines 1024-1026). (This could be said about a lot of titanosaurian specimens—even specimens in other extinct groups!)

"This poor support reflects our initial reluctance to undertake a computational phylogenetic assessment; however, the resulting strict consensus trees provided some areas for discussion." (lines 1062-1063).

"As evident in the above comparative assessment, phylogenetic analysis of Australotitan cooperensis would be premature until better representative skeletal remains of this taxon are available" (lines 2192-2193). (This could be said about any new fossil specimen!)

I would argue that the analyses were useful overall. In almost every analysis from the Poropat et al (2021) and several of the Royo Torres data sets, Australotitan was recovered as a member of Diamantinasauria despite the low support values of the overall tree (it is a large data set and Poropat et al 2021 reported a very high number of MPTs... >1 million, 171,072, 9621, depending on the analysis they ran). So, it's not so much the limited skeleton of Australotitan, it's just the general nature of this massive dataset and missing data within it that yields the initial ambiguity. Still, running various analyses protocols help alleviate this problem (as seen in those studies and the current study). Furthermore, the resultant topologies from the analyses help bring insight on the characters used in the comparative analysis/synapomorphy-assessment, these approaches are complementary to each other. Again, even limited skeletons can be useful depending on what traits are preserved, and here it appears that Australaotitan preserve enough key traits to suggest it was part of or closely related to Diamantinasauria based on both phylogenetic and comparative analyses in the manuscript.

Also, this line in the synapomorphy assessment discussion:

"However, based on the similarities shared with the other Winton Formation taxa, we
propose that W. wattsi should be grouped with the three other Winton Formation taxa, within Titanosauria." Lines 2403-204.

You propose this, yes, but importantly you also tested this with the phylogenetic analyses... resulting in only a handful of topologies supporting this proposal. Phylogenetic analyses not only produce hypothetical tree(s), they also produce a hypothesis on what traits are basal/derived and synapomorphic/homoplastic which would have some bearing on the comparative analysis interpretations.

Validity of the findings

I have no additional comments about their findings or conclusions.

Additional comments

I am happy that most of what was requested from the reviewers were addressed in the manuscript. I honestly think the addition of the phylogenetic analyses overall helps the study of the new taxon by the authors. The tone surrounding these analyses should not be so dismissive, it is just another tool that we paleontologists have in our tool kit to help build our understanding of these extinct animals.

---

## Round 0.3 · accepted · Accept

I confirm your MS is accepted for publication.